# Partial Gromov-Wasserstein Metric

Yikun Bai[1], Rocio Diaz Martin[*2], Abihith Kothapalli[*1],
Hengrong Du[3], Xinran Liu[1], and Soheil Kolouri[1]

[1]Department of Computer Science, Vanderbilt University
[2]Department of Mathematics, Tufts University
[3]Department of Mathematics, University of California, Irvine
[1]yikun.bai, abi.kothapalli, xinran.liu, soheil.kolouri@vanderbilt.edu
[2]rocio.diaz_martin@tufts.edu
[3]hengrond@uci.edu

## Abstract

The Gromov-Wasserstein (GW) distance has gained increasing interest in the machine learning community in recent years, as it allows for the comparison of measures in different metric spaces. To overcome the limitations imposed by the equal mass requirements of the classical GW problem, researchers have begun exploring its application in unbalanced settings. However, Unbalanced GW (UGW) can only be regarded as a discrepancy rather than a rigorous metric/distance between two metric measure spaces (mm-spaces). In this paper, we propose a particular case of the UGW problem, termed Partial Gromov-Wasserstein (PGW). We establish that PGW is a well-defined metric between mm-spaces and discuss its theoretical properties, including the existence of a minimizer for the PGW problem and the relationship between PGW and GW, among others. We then propose two variants of the Frank-Wolfe algorithm for solving the PGW problem and show that they are mathematically and computationally equivalent. Moreover, based on our PGW metric, we introduce the analogous concept of barycenters for mm-spaces. Finally, we validate the effectiveness of our PGW metric and related solvers in applications such as shape matching, shape retrieval, and shape interpolation, comparing them against existing baselines. Our code is available at https://github.com/mint-vu/PGW_Metric.

## 1 Introduction

The classical optimal transport (OT) problem (Villani, 2009) seeks to match two probability measures while minimizing the expected transportation cost. At the heart of classical OT theory lies the principle of mass conservation, which aims to optimize the transfer between two probability measures, assuming they have the same total mass and strictly preserving it. Statistical distances that arise from OT, such as Wasserstein distances, have been widely applied across various machine learning domains, ranging from generative modeling (Arjovsky et al., 2017; Gulrajani et al., 2017) to domain adaptation (Courty et al., 2017) and representation learning (Kolouri et al., 2020). Recent advancements have extended the OT problem to address certain limitations within machine learning applications. These advancements include: 1) facilitating the comparison of non-negative measures that possess different total masses via unbalanced (Chizat et al., 2018c) and partial OT (Figalli, 2010), and 2) enabling the comparison of probability measures across distinct metric spaces through Gromov-Wasserstein distances (Mémoli, 2011), with applications spanning from quantum chemistry (Gilmer et al., 2017) to natural language processing (Alvarez-Melis & Jaakkola, 2018).

Regarding the first aspect, many applications in machine learning involve comparing non-negative measures (often empirical measures) with varying total amounts of mass, e.g.,

---

[*]These authors contributed equally to this work.

domain adaptation (Fatras et al., 2021). Moreover, OT distances (or dissimilarity measures) are often not robust against outliers and noise, resulting in potentially high transportation costs for outliers. Many recent publications have focused on variants of the OT problem that allow for comparing non-negative measures with unequal mass. For instance, the optimal partial transport problem (Figalli, 2010; Caffarelli & McCann, 2010; Figalli & Gigli, 2010; Nguyen et al., 2024; Georgiou et al., 2008; Piccoli & Rossi, 2014), Kantorovich–Rubinstein norm (Guittet, 2002; Heinemann et al., 2023; Lellmann et al., 2014), and the Hellinger–Kantorovich distance (Chizat et al., 2018a; Liero et al., 2018). Recent works formulating the metric properties of partial OT with total variation constraints include (Raghvendra et al., 2024; Nietert et al., 2023). These methods fall under the broad category of "unbalanced optimal transport." In this regard, we also highlight (Balaji et al., 2020; Nguyen et al., 2023; Le et al., 2021), which enhance OT's robustness in the presence of outliers.

Regarding the second aspect, comparing probability measures across different metric spaces is essential in many machine learning applications, ranging from computer graphics, where shapes and surfaces are compared (Bronstein et al., 2006; Mémoli, 2009), to graph partitioning and matching problems (Xu et al., 2019a). Source and target distributions often arise from varied conditions, such as different times, contexts, or measurement techniques, creating substantial differences in intrinsic distances among data points. The conventional OT framework necessitates a meaningful distance across diverse domains, a requirement that is not always achievable. To circumvent this issue, the Gromov-Wasserstein (GW) distances were proposed in (Mémoli, 2011; 2009) as an adaptation of the Gromov-Hausdorff distance, which measures the discrepancy between two metric spaces (Edwards, 1975; Gromov, 1981b;a; Burago et al., 2001). The GW distance (Mémoli, 2011; Sturm, 2023) extends OT-based distances to metric measure spaces (mm-spaces) up to isometries. Its invariance across isomorphic mm-spaces makes the GW distance particularly valuable for applications like shape comparison and matching, where invariance to rigid motion transformations is crucial.

The main computational challenge of the GW metric is the non-convexity of its formulation (Mémoli, 2011). The conventional computational approach relies on the Frank-Wolfe (FW) algorithm (Frank et al., 1956; Lacoste-Julien, 2016). Optimal transport (OT) computational methods (Guittet, 2002; Cuturi, 2013; Papadakis et al., 2014; Benamou et al., 2014; 2015; Peyré et al., 2019; Chizat et al., 2018b; Bonneel & Coeurjolly, 2019; Bai et al., 2023), such as the Sinkhorn algorithm, can be incorporated into FW iterations, which yields the classical GW solvers (Peyré et al., 2016; Xu et al., 2019b; Titouan et al., 2019a).

Given that the GW distance is limited to the comparison of probability mm-spaces, recent works have introduced unbalanced and partial variations (Séjourné et al., 2021; Chapel et al., 2020; De Ponti & Mondino, 2022). These variations have been applied in diverse contexts, including partial graph matching for social network analysis (Liu et al., 2020) and the alignment of brain images (Thual et al., 2022). Although solving these unbalanced variants of the GW problem yields notions of *discrepancies* between mm-spaces, their *metric* properties remain unclear in the literature.

Motivated by the emerging applications of the GW problem in unbalanced settings, this paper focuses on developing a metric between general (not necessarily probability) mm-spaces and providing efficient solvers for its computation. Our proposed metric arises from formulating a variant of the GW problem for unbalanced contexts, rooted in the framework provided by (Séjourné et al., 2021), which we named the *Partial Gromov-Wasserstein* (PGW) problem. In contrast to (Séjourné et al., 2021), which introduces a KL-divergence penalty and a Sinkhorn solver, we employ a total variation penalty, demonstrate the resulting metric properties, and provide novel, efficient solvers for this problem. To the best of our knowledge, this paper presents the first metric for non-probability mm-spaces based on the GW distance.

**Contributions.** Our specific contributions to this paper are:

- **GW metric in unbalanced settings.** We propose the Partial Gromov-Wasserstein (PGW) problem and prove that it gives rise to a metric between arbitrary mm-spaces.

- **PGW solver.** [1] Analogous to the technique presented in (Caffarelli & McCann, 2010), we show that the PGW problem can be turned into a variant of the GW problem. Based on this relation, we propose two mathematically equivalent, but distinct in numerical implementation, Frank-Wolfe solvers for the discrete PGW problem. Inspired by the results of (Lacoste-Julien, 2016), we prove that similar to the Frank-Wolfe solver presented in (Chapel et al., 2020), our proposed solvers for the PGW problem converge linearly to a stationary point.

- **Numerical experiments.** We demonstrate the performance of our proposed algorithms in terms of computation time and efficacy on a series of tasks: shape-matching with outliers between 2D and 3D objects, shape retrieval between 2D shapes, and shape interpolation using the concept of PGW barycenters. We compare the performance of our proposed algorithms against existing baselines for each task.

## 2 BACKGROUND

In this section, we review the basics of OT theory, one of its variants in unbalanced contexts called Partial OT (POT), and their connection as established in (Caffarelli & McCann, 2010). We then introduce the GW distance.

### 2.1 OPTIMAL TRANSPORT AND PARTIAL OPTIMAL TRANSPORT

Let $\Omega \subseteq \mathbb{R}^d$ be, for simplicity, a compact subset of $\mathbb{R}^d$, and $\mathcal{P}(\Omega)$ be the space of probability measures defined on the Borel $\sigma$-algebra of $\Omega$.

**The Optimal Transport (OT) problem** for $\mu, \nu \in \mathcal{P}(\Omega)$, with transportation cost $c(x, y) : \Omega \times \Omega \to \mathbb{R}_+$ being a lower-semi continuous function is defined as:

$$OT(\mu, \nu) := \min_{\gamma \in \Gamma(\mu, \nu)} \gamma(c), \qquad \text{where} \quad \gamma(c) := \int_{\Omega^2} c(x, y) \, d\gamma(x, y) \tag{1}$$

and where $\Gamma(\mu, \nu)$ denotes the set of all joint probability measures on $\Omega^2 := \Omega \times \Omega$ with marginals $\mu, \nu$, i.e., $\gamma_1 := \pi_{1\#}\gamma = \mu, \gamma_2 := \pi_{2\#}\gamma = \nu$, where $\pi_1, \pi_2 : \Omega^2 \to \Omega$ are the canonical projections $\pi_1(x, y) := x, \pi_2(x, y) := y$. A minimizer for (1) always exists (Villani, 2009; 2021) and when $c(x, y) = \|x - y\|^p$, for $p \geq 1$, it defines a metric on $\mathcal{P}(\Omega)$, which is referred to as the "$p$-Wasserstein distance":

$$W_p^p(\mu, \nu) := \min_{\gamma \in \Gamma(\mu, \nu)} \int_{\Omega^2} \|x - y\|^p d\gamma(x, y). \tag{2}$$

**The Partial Optimal Transport (POT) problem** (Chizat et al., 2018c; Figalli & Gigli, 2010; Piccoli & Rossi, 2014) extends the OT problem to the set of Radon measures $\mathcal{M}_+(\Omega)$, i.e., non-negative and finite measures. For $\lambda > 0$ and $\mu, \nu \in \mathcal{M}_+(\Omega)$, the POT problem is defined as:

$$POT(\mu, \nu; \lambda) := \inf_{\gamma \in \mathcal{M}_+(\Omega^2)} \gamma(c) + \lambda(|\mu - \gamma_1| + |\nu - \gamma_2|), \tag{3}$$

where, in general, $|\sigma|$ denotes the total variation norm of a measure $\sigma$, i.e., $|\sigma| := \sigma(\Omega)$. The constraint $\gamma \in \mathcal{M}_+(\Omega^2)$ in (3) can be further restricted to $\gamma \in \Gamma_{\leq}(\mu, \nu)$:

$$\Gamma_{\leq}(\mu, \nu) := \{\gamma \in \mathcal{M}_+(\Omega^2) : \gamma_1 \leq \mu, \gamma_2 \leq \nu\},$$

denoting $\gamma_1 \leq \mu$ if for any Borel set $B \subseteq \Omega$, $\gamma_1(B) \leq \mu(B)$ (respectively, for $\gamma_2 \leq \nu$) (Figalli, 2010). Roughly speaking, the linear penalization indicates that if the classical transportation cost exceeds $2\lambda$, it is better to create/destroy' mass (see (Bai et al., 2023) for further details).

**The relationship between POT and OT.** By using the techniques in (Caffarelli & McCann, 2010), the POT problem can be transferred into an OT problem, and thus, OT solvers (e.g., network simplex) can be employed to solve the POT problem.

---

[1] Rigorously speaking, due to the non-convexity of the GW problem and its variants, current methods achieve only local minima rather than global minima. We use the term "solver" following the convention of previous related works (Séjourné et al., 2021; Chapel et al., 2020). However, it should be emphasized that the proposed methods aim to find local minima rather than global minima, similar to related and classical computational GW works.

**Proposition 2.1.** *(Caffarelli & McCann, 2010; Bai et al., 2023) Given $\mu, \nu \in \mathcal{M}_+(\Omega)$, construct the following measures on $\hat{\Omega} := \Omega \cup \{\hat{\infty}\}$, for an auxiliary point $\hat{\infty}$:*

$$\hat{\mu} = \mu + |\nu|\delta_{\hat{\infty}} \quad and \quad \hat{\nu} = \nu + |\mu|\delta_{\hat{\infty}}. \tag{4}$$

*Consider the following OT problem*

$$OT(\hat{\mu}, \hat{\nu}) = \min_{\hat{\gamma} \in \Gamma(\hat{\mu}, \hat{\nu})} \hat{\gamma}(\hat{c}), \quad where \quad \hat{c}(x, y) := \begin{cases} c(x, y) - 2\lambda & if \ x, y \in \Omega, \\ 0 & elsewhere. \end{cases} \tag{5}$$

*Then, there exists a bijection $F : \Gamma_{\leq}(\mu, \nu) \to \Gamma(\hat{\mu}, \hat{\nu})$ given by*

$$F(\gamma) := \gamma + (\mu - \gamma_1) \otimes \delta_{\hat{\infty}} + \delta_{\hat{\infty}} \otimes (\nu - \gamma_2) + |\gamma|\delta_{\hat{\infty},\hat{\infty}}. \tag{6}$$

*such that $\gamma$ is optimal for the POT problem (3) if and only if $F(\gamma)$ is optimal for the OT problem (5).*

It is worth noting that instead of considering the same underlying space $\Omega$ for both measures $\mu$ and $\nu$, the OT and POT problems can be formulated in the scenario where $\mu$ and $\nu$ are defined on different metric spaces $X$ and $Y$, respectively. In this setting, one needs a cost function $c : X \times Y \to \mathbb{R}_+$ to formulate the OT and POT problems. However, in practice, it is usually difficult to define reasonable 'distance' or *ground cost* $c(\cdot, \cdot)$ between the two spaces $X$ and $Y$. In particular, the $p$-Wasserstein distance cannot be adopted if $\mu, \nu$ are defined on different spaces. To relax this requirement, in the next section, we will review the fundamentals of the *Gromov-Wasserstein* problem (Mémoli, 2011).

## 2.2 THE GROMOV-WASSERSTEIN (GW) PROBLEM

A metric measure space (mm-space) consists of a set $X$ endowed with a metric structure, that is, a notion of distance $d_X$ between its elements, and equipped with a Borel measure $\mu$. As in Mémoli (2011, Ch. 5), we will assume that $X$ is compact and that $\text{supp}(\mu) = X$. Given two probability mm-spaces $\mathbb{X} = (X, d_X, \mu)$, $\mathbb{Y} = (Y, d_Y, \nu)$, with $\mu \in \mathcal{P}(X)$ and $\nu \in \mathcal{P}(Y)$, and a non-negative lower semi-continuous cost function $L : \mathbb{R}^2 \to \mathbb{R}_+$ (e.g., the Euclidean distance or the KL-loss), the Gromov-Wasserstein (GW) matching problem is defined as:

$$GW^L(\mathbb{X}, \mathbb{Y}) := \inf_{\gamma \in \Gamma(\mu, \nu)} \gamma^{\otimes 2}(L(d_X(\cdot, \cdot), d_Y(\cdot, \cdot))), \tag{7}$$

where, for brevity, we employ the notation $\gamma^{\otimes 2}$ for the product measure $d\gamma^{\otimes 2}((x, y), (x', y')) = d\gamma(x, y)d\gamma(x', y')$. If $L(a, b) = |a - b|^p$, for $1 \leq p < \infty$, we denote $GW^L(\cdot, \cdot)$ simply by $GW^p(\cdot, \cdot)$. In this case, the expression (7) defines an equivalence relation $\sim$ among probability mm-spaces, i.e., $\mathbb{X} \sim \mathbb{Y}$ if and only if $GW^p(\mathbb{X}, \mathbb{Y}) = 0$[2]. A minimizer of the GW problem (7) always exists, and thus, we can replace inf by min. Moreover, similar to OT, the above GW problem defines a distance for probability mm-spaces after taking the quotient under $\sim$. For details, we refer to Mémoli (2011, Ch. 5 and 10).

## 3 THE PARTIAL GROMOV-WASSERSTEIN (PGW) PROBLEM

The Unbalanced Gromov-Wasserstein (UGW) problem for general (compact) mm-spaces $\mathbb{X} = (X, d_X, \mu), \mathbb{Y} = (Y, d_Y, \nu)$, with $\mu \in \mathcal{M}_+(X), \nu \in \mathcal{M}_+(Y)$, studied in (Séjourné et al., 2021; Kong et al., 2024) is defined as:

$$UGW_\lambda^L(\mathbb{X}, \mathbb{Y}) := \inf_{\gamma \in \mathcal{M}_+(X \times Y)} \gamma^{\otimes 2}(L(d_X, d_Y)) + \lambda(D_\phi(\gamma_1^{\otimes 2} \| \mu^{\otimes 2}) + D_\phi(\gamma_2^{\otimes 2} \| \nu^{\otimes 2})), \tag{8}$$

where $\lambda > 0$ is a fixed linear penalization parameter, and $D_\phi$ is a Csiszár or $\phi$-divergence. The above formulation extends the classical GW problem (7) into the unbalanced setting ($\mu$ and $\nu$ are no longer necessarily probability measures but general Radon measures).

We underline two points: First, as discussed in (Séjourné et al., 2021), while the above quantity allows us to 'compare' the mm-spaces $\mathbb{X}$ and $\mathbb{Y}$, its *metric* property is unclear.

---

[2]Moreover, given two probability mm-spaces $\mathbb{X}$ and $\mathbb{Y}$, $GW(\mathbb{X}, \mathbb{Y}) = 0$ if and only if there exists a bijective isometry $\phi : X \to Y$ such that $\phi_\# \mu = \nu$. In particular, the GW distance is invariant under rigid transformations (translations and rotations) of a given probability mm-space.

Secondly, when $D_\phi$ is the KL divergence, a Sinkhorn solver has been proposed in (Séjourné et al., 2021). However, a solver for general $\phi$-divergences has not yet been proposed.

In this paper, we will analyze the case when $D_\phi$ is the total variation norm. Specifically, for $q \geq 1$, we consider the following problem, which we refer to as the *Partial Gromov-Wasserstein* (PGW) problem:

$$PGW_{\lambda,q}^L(\mathbb{X}, \mathbb{Y}) := \inf_{\gamma \in \mathcal{M}_+(X \times Y)} \gamma^{\otimes 2}(L(d_X^q, d_Y^q)) + \lambda(|\mu^{\otimes 2} - \gamma_1^{\otimes 2}| + |\nu^{\otimes 2} - \gamma_2^{\otimes 2}|). \quad (9)$$

**Remark 3.1.** *If $\gamma \in \Gamma \leq (\mu, \nu)$, the above cost functional can be rewritten as*

$$\gamma^{\otimes 2}(L(d_X^q, d_Y^q)) + \lambda(|\mu^{\otimes 2} - \gamma_1^{\otimes 2}| + |\nu^{\otimes 2} - \gamma_2^{\otimes 2}|) = \gamma^{\otimes 2}\left(L(d_X^q, d_Y^q) - 2\lambda\right) + \underbrace{\lambda\left(|\mu|^2 + |\nu|^2\right)}_{does \ not \ depend \ on \ \gamma} .$$

**Proposition 3.2.** *Given mm-spaces $\mathbb{X} = (X, d_X, \mu)$, $\mathbb{Y} = (Y, d_Y, \nu)$, the minimization problem (9) can be restricted to the set $\Gamma_\leq(\mu, \nu) = \{\gamma \in \mathcal{M}_+(X \times Y) : \gamma_1 \leq \mu, \gamma_2 \leq \nu\}$. That is,*

$$PGW_{\lambda,q}^L(\mathbb{X}, \mathbb{Y}) = \inf_{\gamma \in \Gamma_\leq(\mu,\nu)} \gamma^{\otimes 2}\left(L(d_X^q, d_Y^q) - 2\lambda\right) + \lambda(|\mu|^2 + |\nu|^2). \quad (10)$$

For the proof inspired by (Piccoli & Rossi, 2014), we direct the reader to Appendix B.

We notice that a similar Partial Gromov-Wasserstein problem (and its solver) has been studied (Chapel et al., 2020). Indeed, in (Chapel et al., 2020), the $\lambda$-penalization in the optimization problem (10) is avoided, but the constraint set is replaced by the subset of all $\gamma \in \Gamma_\leq(\mu, \nu)$ such that $|\gamma| = \rho$ for a fixed $\rho \in [0, \min\{|\mu|, |\nu|\}]$. We will call this formulation the *Mass-Constrained Partial Gromov-Wasserstein* (MPGW) problem. In Appendix L, we explore the relations between PGW and MPGW, and in Section 5 and Appendices O, Q, we analyze the performance of the different solvers through different experiments.

**Proposition 3.3.** *If $L(r_1, r_2) = |r_1 - r_2|^p$, for $p \in [1, \infty)$, we use $PGW_{\lambda,q}^p$ to denote $PGW_{\lambda,q}^L$. In this case, (9) and (10) admit a minimizer.*

The proof is given in Appendix C: Its idea extends results from (Mémoli, 2011) from probability mm-spaces to arbitrary mm-spaces.

Next, we state one of our main results: The PGW problem gives rise to a metric between mm-spaces. The rigorous statement, as well as its proof, is given in Appendix D: The formal statement is based on the definition of *equivalence classes* among mm-spaces (see Remark D.1). The most difficult part of the proof is the triangle inequality, and the main technique used relies on the relation between PGW and GW (see Appendix D.3).

**Proposition 3.4.** *Let $\lambda > 0$, $1 \leq q, p < \infty$ and $L(r_1, r_2) = |r_1 - r_2|^p$. Then $(PGW_{\lambda,q}^p(\cdot, \cdot))^{1/p}$ defines a metric between mm-spaces.*

Finally, for consistency, we provide the following result when the penalization tends to infinity. Its proof is given in Appendix E.

**Proposition 3.5.** *Consider probability mm-spaces $\mathbb{X} = (X, d_X, \mu)$, $\mathbb{Y} = (Y, d_Y, \nu)$, that is, $|\mu| = |\nu| = 1$. Assume that $L$ is a continuous function. Then $\lim_{\lambda \to \infty} PGW_{\lambda,q}^L(\mathbb{X}, \mathbb{Y}) = GW^L(\mathbb{X}, \mathbb{Y})$.*

## 4 COMPUTATION OF THE PARTIAL GW DISTANCE

In the discrete setting, consider mm-spaces $\mathbb{X} = (X, d_X, \sum_{i=1}^n p_i^X \delta_{x_i})$, $\mathbb{Y} = (Y, d_Y, \sum_{j=1}^m q_j^Y \delta_{y_j})$, where $X = \{x_1, \ldots, x_n\}$, $Y = \{y_1, \ldots, y_m\}$, the weights $p_i^X$, $q_j^Y$ are non-negative numbers, and the distances $d_X$, $d_Y$ are determined by the matrices $C^X \in \mathbb{R}^{n \times n}$, $C^Y \in \mathbb{R}^{m \times m}$ defined by

$$C_{i,i'}^X := d_X^q(x_i, x_{i'}) \quad \forall i, i' \in [1:n] \quad \text{and} \quad C_{j,j'}^Y := d_Y^q(y_j, y_{j'}) \quad \forall j, j' \in [1:m]. \quad (11)$$

Let $\mathrm{p} := [q_1^X, \ldots, q_n^X]^\top$ and $\mathrm{q} := [q_1^Y, \ldots, q_m^Y]^\top$ denote the weight vectors corresponding to the given discrete measures. We view the sets of transportation plans $\Gamma(\mathrm{p}, \mathrm{q})$ and $\Gamma_\leq(\mathrm{p}, \mathrm{q})$

for the GW and PGW problems, respectively, as the subsets of $n \times m$ matrices

$$\Gamma(p, q) := \{\gamma \in \mathbb{R}_+^{n \times m} : \gamma 1_m = p, \gamma^\top 1_n = q\}, \quad \text{if } |p| = \sum_{i=1}^n p_i^X = 1 = \sum_{j=1}^m q_j^Y = |q|; \quad (12)$$

$$\Gamma_\le(p, q) := \{\gamma \in \mathbb{R}_+^{n \times m} : \gamma 1_m \le p, \gamma^\top 1_n \le q\}, \quad (13)$$

for any pair of non-negative vectors $p \in \mathbb{R}_+^n$, $q \in \mathbb{R}_+^m$, where $1_n$ is the vector with all ones in $\mathbb{R}^n$ (resp. $1_m$), and $\gamma 1_m \le p$ means that component-wise the $\le$ relation holds.

Given by a non-negative function $L : \mathbb{R}^{n \times n} \times \mathbb{R}^{m \times m} \to \mathbb{R}_+$, the transportation cost $M$ and the 'partial' transportation cost $\tilde{M}$ are represented by the $n \times m \times n \times m$ tensors:

$$M_{i,j,i',j'} = L(C_{i,i'}^X, C_{j,j'}^Y) \qquad \text{and} \qquad \tilde{M} := M - 2\lambda := M - 2\lambda 1_{n,m,n,m}, \quad (14)$$

where $1_{n,m,n,m}$ is the tensor with ones in all its entries. For each $n \times m \times n \times m$ tensor $M$ and each $n \times m$ matrix $\gamma$, we define tensor-matrix multiplication $M \circ \gamma \in \mathbb{R}^{n \times m}$ by

$$(M \circ \gamma)_{ij} = \sum_{i',j'} (M_{i,j,i',j'}) \gamma_{i',j'}.$$

Then, the Partial GW problem in (10) can be written as

$$PGW_\lambda^L(\mathbb{X}, \mathbb{Y}) = \min_{\gamma \in \Gamma_\le(p,q)} \mathcal{L}_{\tilde{M}}(\gamma) + \lambda(|p|^2 + |q|^2), \quad \text{where} \quad (15)$$

$$\mathcal{L}_{\tilde{M}}(\gamma) := \tilde{M}\gamma^{\otimes 2} := \sum_{i,j,i',j'} \tilde{M}_{i,j,i',j'} \gamma_{i,j} \gamma_{i',j'} = \sum_{ij} (\tilde{M} \circ \gamma)_{ij} \gamma_{ij} =: \langle \tilde{M} \circ \gamma, \gamma \rangle_F, \quad (16)$$

and $\langle \cdot, \cdot \rangle_F$ stands for the Frobenius dot product. The constant term $\lambda(|p|^2 + |q|^2)$ will be ignored in the rest of this paper since it does not depend on $\gamma$.

### 4.1 Frank-Wolfe for the PGW Problem – Solver 1

In this section, we discuss the Frank-Wolfe (FW) algorithm for the PGW problem (15). A second variant of the FW solver is provided in the Appendix G.

As a summary, in our proposed method, we address the discrete PGW problem (15), highlighting that the *direction-finding subproblem* in the Frank-Wolfe (FW) algorithm is a POT problem for (15). Specifically, (15) is treated as a discrete POT problem in our Solver 1, where we apply Proposition 2.1 to solve a discrete OT problem.

For each iteration $k$, the procedure is summarized in three steps detailed below.

The convergence analysis, detailed in Appendix K, applies the results from (Lacoste-Julien, 2016) to our context, showing that the FW algorithm achieves a stationary point at a rate of $\mathcal{O}(1/\sqrt{k})$ for non-convex objectives with a Lipschitz continuous gradient in a convex and compact domain.

Step 1. Computation of gradient and optimal direction.

It is straightforward to verify that the gradient of the objective function (16) in (15) is given by

$$\nabla \mathcal{L}_{\tilde{M}}(\gamma) = 2\tilde{M} \circ \gamma. \quad (17)$$

The classical method to compute $M \circ \gamma$ is the following: First, convert $M$ into an $(n \times m) \times (n \times m)$ matrix, denoted as $v(M)$, and convert $\gamma$ into an $(n \times m) \times 1$ vector $v(\gamma)$. Then, the computation of $M \circ \gamma$ is equivalent to the matrix multiplication $v(M)v(\gamma)$. The computational cost and the required storage space are $\mathcal{O}(n^2 m^2)$. In certain conditions, the above computation can be reduced to $\mathcal{O}(n^2 + m^2)$. We refer to Appendices F and H for details.

Next, we aim to solve the following problem:

$$\gamma^{(k)'} \leftarrow \arg \min_{\gamma \in \Gamma_\le(p,q)} \langle \nabla \mathcal{L}_{\tilde{M}}(\gamma^{(k)}), \gamma \rangle_F,$$

which is a discrete POT problem since it is equivalent to

$$\min_{\gamma \in \Gamma_\le(p,q)} \langle 2M \circ \gamma^{(k)}, \gamma \rangle_F + \lambda |\gamma^{(k)}| (|p| + |q| - 2|\gamma|).$$

The solver can be obtained by firstly converting the POT problem into an OT problem via Proposition 2.1 and then solving the proposed OT problem.

---

**Algorithm 1:** Frank-Wolfe Algorithm for PGW, ver 1

---

**Input:** $\mu = \sum_{i=1}^{n} p_i^X \delta_{x_i}, \nu = \sum_{j=1}^{m} q_j^Y \delta_{y_j}, \gamma^{(1)}$
**Output:** $\gamma^{(final)}$
Compute $C^X, C^Y$
**for** $k = 1, 2, \ldots$ **do**
$\quad G^{(k)} \leftarrow 2\tilde{M} \circ \gamma^{(k)}$ // Compute gradient
$\quad \gamma^{(k)'} \leftarrow \arg\min_{\gamma \in \Gamma_{\leq}(\mathrm{p},\mathrm{q})} \langle G^{(k)}, \gamma \rangle_F$ // Solve the POT problem.
$\quad$ Compute $\alpha^{(k)} \in [0, 1]$ via (18) // Line Search
$\quad \gamma^{(k+1)} \leftarrow (1 - \alpha^{(k)})\gamma^{(k)} + \alpha^{(k)}\gamma^{(k)'}$ // Update $\gamma$
$\quad$ if convergence, break
**end for**
$\gamma^{(final)} \leftarrow \gamma^{(k)}$

---

STEP 2: LINE SEARCH METHOD.

In this step, at the $k$-th iteration, we need to determine the optimal step size:

$$\alpha^{(k)} = \arg\min_{\alpha \in [0,1]} \{\mathcal{L}_{\tilde{M}}((1-\alpha)\gamma^{(k)} + \alpha\gamma^{(k)'})\}.$$

The optimal $\alpha^{(k)}$ takes the following values (see Appendix I for details):

$$\text{Let } \alpha^{(k)} = \begin{cases} 0 & \text{if } a \leq 0, a + b > 0, \\ 1 & \text{if } a \leq 0, a + b \leq 0, \\ \text{clip}(\frac{-b}{2a}, [0,1]) & \text{if } a > 0, \end{cases} \text{ where } \begin{cases} \delta\gamma^{(k)} = \gamma^{(k)'} - \gamma^{(k)}, \\ a = \langle \tilde{M} \circ \delta\gamma^{(k)}, \delta\gamma^{(k)} \rangle_F, \\ b = 2\langle \tilde{M} \circ \gamma^{(k)}, \delta\gamma^{(k)} \rangle_F. \end{cases}$$

(18)

and $\text{clip}(\frac{-b}{2a}, [0,1]) = \min\{\max\{-\frac{b}{2a}, 0\}, 1\}$.

STEP 3: UPDATE $\gamma^{(k+1)} \leftarrow (1-\alpha^{(k)})\gamma^{(k)} + \alpha^{(k)}\gamma^{(k)'}$.

### 4.2 NUMERICAL IMPLEMENTATION DETAILS

**The initial guess, $\gamma^{(1)}$.** In the GW problem, if there is no prior knowledge, the initial guess is set to $\gamma^{(1)} = \mathrm{p}\,\mathrm{q}^{\top}$.

In PGW, however, as $\mu, \nu$ may not necessarily be probability measures (i.e., $\sum_i p_i^X, \sum_j q_j^Y \neq 1$ in general), we set $\gamma^{(1)} = \frac{\mathrm{pq}^{\top}}{\max(|\mathrm{p}|,|\mathrm{q}|)}$. It is straightforward to verify that $\gamma^{(1)} \in \Gamma_{\leq}(\mathrm{p}, \mathrm{q})$ as

$$\gamma^{(1)}1_m = \frac{|\mathrm{q}|\mathrm{p}}{\max(|\mathrm{p}|, |\mathrm{q}|)} \leq \mathrm{p}, \quad \gamma^{(1)\top}1_n = \frac{|\mathrm{p}|\mathrm{q}}{\max(|\mathrm{p}|, |\mathrm{q}|)} \leq \mathrm{q}.$$

**Column/Row-Reduction.** According to the interpretation of the penalty weight parameter in the Partial OT problem (e.g. see Lemma 3.2 in (Bai et al., 2023)), during the POT solving step, for each $i \in [1:n]$ (or $j \in [1:m]$), if the $i^{th}$ row ($j^{th}$ column) of $\tilde{M} \circ \gamma^{(k)}$ contains a non-negative entry, all the mass of $p_i^X$ ($q_j^Y$) will be destroyed (created). Thus, we can remove the corresponding row (column) to improve the computational efficiency.

## 5 EXPERIMENTS

### 5.1 TOY EXAMPLE: SHAPE MATCHING WITH OUTLIERS

We use the moon dataset and synthetic 2D/3D spherical data in this experiment. Let $\{x_i\}_{i=1}^{n}, \{y_j\}_{j=1}^{n}$ denote the source and target point clouds. In addition, we add $\eta n$ (where $\eta = 20\%$) outliers to the target point cloud. See Figure 1 for visualization.

We visualize the transportation plans given by the GW (Mémoli, 2011), MPGW (Chapel et al., 2020), UGW (Séjourné et al., 2021), and our proposed PGW problems. For MPGW, UGW, and PGW, we set the mass to be 1 for each point in the source and target point clouds. For GW, we normalize the mass of these points so that the source and target have

the same total mass. From Figure 1, we observe that PGW and MPGW induce a one-by-one relation in both cases, and no outlier points are matched to the source point cloud. Meanwhile, GW matches all of the outliers. For UGW, as it applies the Sinkhorn algorithm, we observe mass-splitting transportation plans in both cases. Moreover, we observe that some mass from the outliers has been matched, which is not desired.

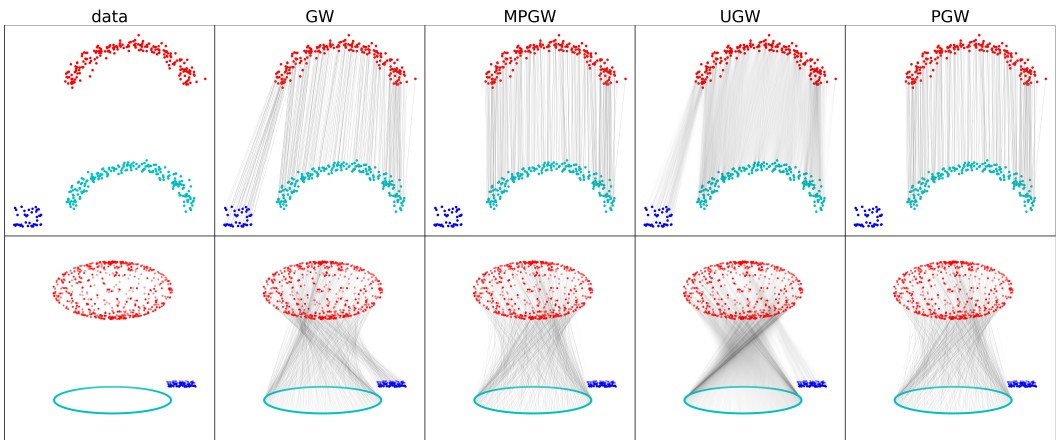

Figure 1: The set of red points comprises the source point cloud. The union of the dark blue (outliers) and light blue points comprises the target point cloud. For UGW, MPGW, and PGW, we set the mass for each point to be the same. For GW, we normalize the mass for the balanced mass constraint setting.

## 5.2 Shape Retrieval

**Experiment setup.** We now employ the PGW distance to distinguish between 2D shapes, as done in (Beier et al., 2022), and use GW, MPGW, and UGW as baselines for comparison. Given a series of 2D shapes, we represent the shapes as mm-spaces $\mathbb{X}^i = (\mathbb{R}^2, \|\cdot\|_2, \mu^i)$, where $\mu^i = \sum_{k=1}^{n^i} \alpha^i \delta_{x_k^i}$. For the GW method, we normalize the mass for the balanced mass constraint setting (i.e., $\alpha^i = \frac{1}{n^i}$), and for the remaining methods, we let $\alpha^i = \alpha$ for all the shapes, where $\alpha > 0$ is a fixed constant. In this manner, we compute the pairwise distances between the shapes.

We then use the computed distances for nearest neighbor classification. We do this by choosing a representative at random from each class in the dataset and then classifying each shape according to its nearest representative. This is repeated over 10,000 iterations, and we generate a confusion matrix for each distance used. Finally, using the approach given by (Beier et al., 2022; Titouan et al., 2019b), we combine each distance with a support vector machine (SVM), applying stratified 10-fold cross-validation. In each iteration of cross-validation, we train an SVM using $\exp(-\sigma D)$ as the kernel, where $D$ is the matrix of pairwise distances (w.r.t. one of the considered distances) restricted to 9 folds, and compute the accuracy of the model on the remaining fold. We report the accuracy averaged over all 10 folds for each model.

**Dataset setup.** We test two datasets in this experiment, which we refer to as Dataset I and Dataset II. We construct Dataset I by adapting the 2D shape dataset given in (Beier et al., 2022), consisting of 20 shapes in each of the classes: bone, goblet, star, and horseshoe. For each class, we augment the dataset with an additional class by selecting either a subset of points from each shape of that class (rectangle/bone, trapezoid/goblet, disk/star) or adding additional points to each shape of that class (annulus/horseshoe). Hence, the final dataset consists of 160 shapes across 8 total classes. This dataset is visualized in Figure 9a.

For Dataset II, we generate 20 shapes for each of the classes: rectangle, house, arrow, double arrow, semicircle, and circle. These shapes were generated in pairs, such that each shape of class rectangle is a subset of the corresponding shape of class house, and similarly for arrow/double arrow and semicircle/circle. This dataset is visualized in Figure 9b.

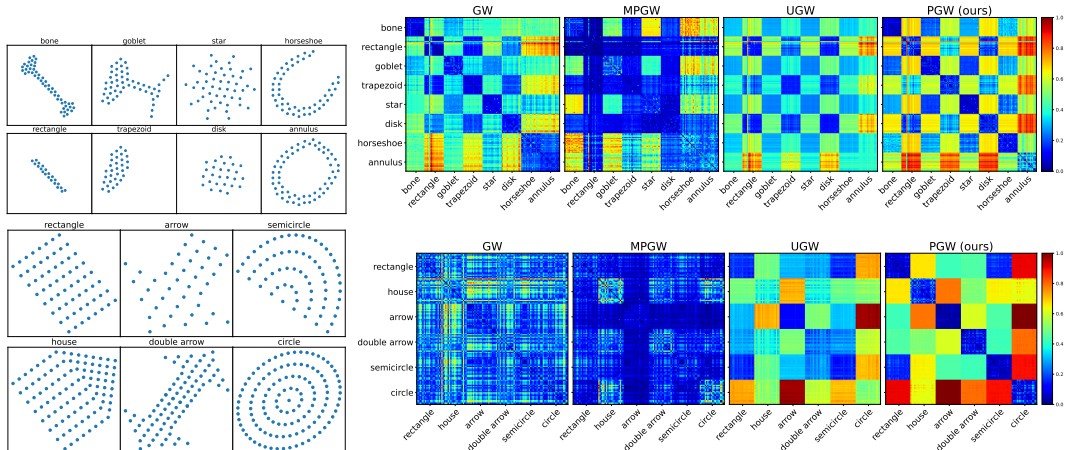

Figure 2: In each row, the first figure visualizes an example shape from each class, and the second figure visualizes the resulting pairwise distance matrices. The first row corresponds to Dataset I, and the second corresponds to Dataset II.

| Distance | Dataset I | Dataset II |
|---|---|---|
| GW | **0.9813** | 0.8083 |
| MPGW | 0.2375 | 0.2500 |
| UGW | 0.89375 | 0.9000 |
| PGW (ours) | 0.9625 | **1.0000** |

(a) Mean accuracy of SVM using each distance in the kernel.

| Distance | Dataset I | Dataset II |
|---|---|---|
| GW | 49.02s | 137.12s |
| MPGW | 49.10s | 93.90s |
| UGW | 1484.49s | 519.91s |
| PGW (ours) | **35.92s** | **79.27s** |

(b) Wall-clock time comparison.

Table 1: Accuracy and wall-clock time comparison for shape retrieval experiment.

**Performance analysis**. We refer to Appendix O for full numerical details, parameter settings, and the visualization of the resulting confusion matrices. We visualize the two considered datasets and the resulting pairwise distance matrices in Figure 2. For the SVM experiments, GW achieves the highest accuracy on Dataset I, 98.13%, while the second best method is PGW, 96.25%. For Dataset II, PGW achieves the highest accuracy, correctly classifying 100% of the samples. The complete set of accuracies for all considered distances on each dataset is reported in Table 1a.

In addition, we report the wall-clock time required to compute all pairwise distances for each distance in Table 1b. We observe that GW, MPGW, and PGW have similar wall-clock times across both experiments (30-50 seconds for Dataset I, 80-140 seconds for Dataset II), with PGW admitting a slightly faster runtime in both cases. Meanwhile, UGW requires almost 1500 seconds on the experiment with Dataset I and over 500 seconds on the experiment with Dataset II.

### 5.3 PARTIAL GROMOV-WASSERSTEIN BARYCENTER AND SHAPE INTERPOLATION

By (Peyré et al., 2016), Gromov-Wasserstein can be applied to interpolate two shapes via the concept of *Gromov-Wasserstein Barycenters*. In this paper, we introduce *Partial Gromov-Wasserstein Barycenters* by extending the GW Barycenter to the setting of PGW as follows.

Consider the discrete mm-spaces $\mathbb{X}^1, \ldots, \mathbb{X}^K$, where $\mathbb{X}^k = (X^k, \|\cdot\|_{\mathbb{R}^{d_k}}, \sum_{i=1}^{n_k} p_i^k \delta_{x_i^k})$, with $X^k = \{x_i^k\}_{i=1}^{n_k} \subset \mathbb{R}^{d_k}$. We denote $C^k = [\|x_i^k - x_{i'}^k\|^2]_{i,i'}$ and $\mathrm{p}^k = [p_1^k, \ldots, p_{n_k}^k]$. Given positive constants $\lambda_1, \ldots, \lambda_K > 0$, the PGW Barycenter is defined by:

$$\min_{C, \gamma_k} \sum_k \xi_k \langle M(C, C^k) \circ \gamma^k, \gamma^k \rangle - 2\lambda_k |\gamma^k|^2 \tag{19}$$

where each $\gamma^k \in \Gamma_{\leq}(\mathrm{p}, \mathrm{p}^k)$. We refer to Appendix M for the solver of (19) and details.

**Experiment setup.** We apply the PGW barycenter to the following problem: Given two shapes $X = \{x_i\}_{i=1}^n \subset \mathbb{R}^{d_1}$ and $Y = \{y_i\}_{i=1}^m \subset \mathbb{R}^{d_2}$, modeled as mm-spaces $\mathbb{X} = (X, \|\cdot\|_{\mathbb{R}^{d_1}}, \sum_{i=1}^n \delta_{x_i})$ and $\mathbb{Y} = (Y, \|\cdot\|_{\mathbb{R}^{d_2}}, \sum_{i=1}^m \delta_{y_i})$, we wish to find interpolations between them. In addition, we assume $\mathbb{Y}$ is corrupted by noise, i.e., $\mathbb{Y}$ is redefined as $\mathbb{Y} = (\tilde{Y}, \|\cdot\|_{\mathbb{R}^{d_2}}, \sum_{i=1}^m \delta_{y_i} + \sum_{i=1}^{m\eta} \delta_{\tilde{y}_i})$ with $\tilde{Y} = Y \cup \{\tilde{y}_i\}_{i=1}^m$, where $\eta \in [0, 1]$ is the noise level and each $\tilde{y}_i$ is randomly selected from a particular region $\mathcal{R} \subset \mathbb{R}^{d_2}$.

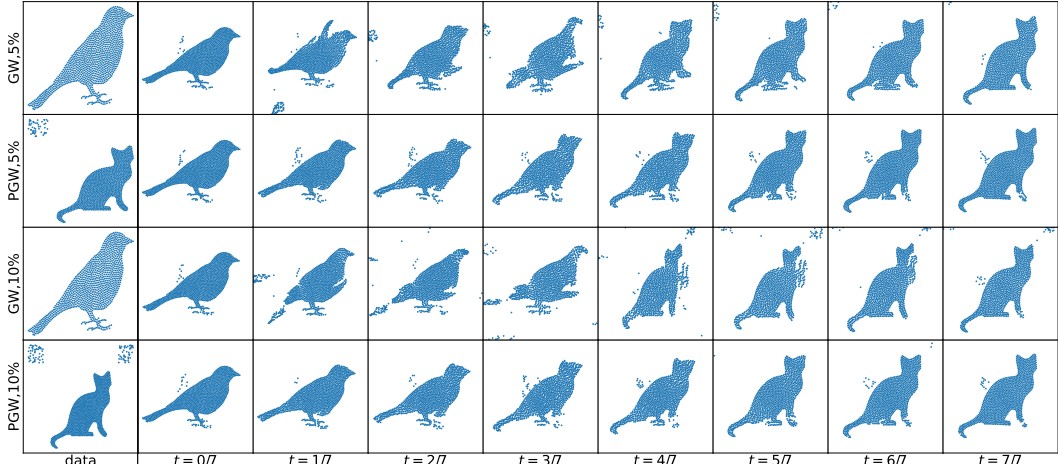

Figure 3: In the first column, the first and second figures are the source and target point clouds in the first experiment ($\eta = 5\%$); the third and fourth figures are the source and target point clouds in the second experiment ($\eta = 10\%$).

**Dataset setup.** We adapt the dataset given in (Peyré et al., 2016). See Appendix M.1 for further details on the dataset. In this experiment, we test $\eta = 5\%, 10\%$. We visualize the barycenter interpolation from $t = 0/7$ to $t = 7/7$, where $(1-t), t$ are the weight of the source $\mathbb{X}$ and the target $\mathbb{Y}$, respectively, in the barycenter (19). The visualization given in Figure 3 is obtained by applying SMACOF MDS (multidimensional scaling) of the minimizer $C$.

**Performance analysis.** From Figure 3, we observe that in these two scenarios, the interpolation derived from GW is clearly disturbed by the noise data points. For example, in rows $1, 3$, columns $t = 1/7, 2/7, 3/7$, we see that the point clouds reconstructed by MDS have significantly different width-height ratios from those of the source and target point clouds. In contrast, PGW is significantly less disturbed, and the interpolation is more natural. The width-height ratio of the point clouds generated by the PGW barycenter is consistent with that of the source/target point clouds.

## 6 SUMMARY

In this paper, we propose the Partial Gromov-Wasserstein (PGW) problem and introduce two Frank-Wolfe solvers for it. As a byproduct, we provide pertinent theoretical results, including the relation between PGW and GW, the metric property of PGW, and the PGW barycenter formulation. Furthermore, we demonstrate the efficacy of the PGW solver in solving shape matching, shape retrieval, and shape interpolation tasks. For the shape retrieval experiment, we observe that due to their metric property, PGW and GW have similar accuracy and outperform the other methods evaluated. In the shape matching and shape interpolation experiments, we demonstrate that PGW admits a more robust result when the data are corrupted by outliers/noisy data.

## ACKNOWLEDGMENTS

This research was partially supported by the NSF CAREER Award No. 2339898.

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

## A  NOTATION AND ABBREVIATIONS

- OT: Optimal Transport.
- POT: Partial Optimal Transport.
- GW: Gromov-Wasserstein.
- PGW: Partial Gromov-Wasserstein.
- FW: Frank-Wolfe.
- MPGW: Mass-Constrained Partial Gromov-Wasserstein.
- $\|\cdot\|$: Euclidean norm.
- $X^2 = X \times X$.
- $\mathcal{M}_+(X)$: set of all positive (non-negative) Randon (finite) measures defined on $X$.
- $\mathcal{P}_2(X)$: set of all probability measures defined on $X$, whose second moment is finite.
- $\mathbb{R}_+$: set of all non-negative real numbers.
- $\mathbb{R}^{n \times m}$: set of all $n \times m$ matrices with real coefficients.
- $\mathbb{R}^{n \times m}_+$ (resp. $\mathbb{R}^n_+$): set of all $n \times m$ matrices (resp., $n$-vectors) with non-negative coefficients.
- $\mathbb{R}^{n \times m \times n \times m}$: set of all $n \times m \times n \times m$ tensors with real coefficients.
- $1_n, 1_{n \times m}, 1_{n \times m \times n \times m}$: vector, matrix, and tensor of all ones.
- $\mathbb{1}_E$: characteristic function of a measurable set $E$

$$\mathbb{1}_E(z) = \begin{cases} 1 & \text{if } z \in E, \\ 0 & \text{otherwise.} \end{cases}$$

- $\mathbb{X}, \mathbb{Y}$: metric measure spaces (mm-spaces): $\mathbb{X} = (X, d_X, \mu)$, $\mathbb{Y} = (Y, d_Y, \nu)$.
- $C^X$: given a discrete mm-space $\mathbb{X} = (X, d_X, \mu)$, where $X = \{x_1, \ldots, x_n\}$, the symmetric matrix $C^X \in \mathbb{R}^{n \times n}$ is defined as $C^X_{i,i'} = d^q_X(x_i, x'_i)$.
- $\mu^{\otimes 2}$: product measure $\mu \otimes \mu$.
- $T_\# \sigma$: $T : X \to Y$ is a measurable function and $\sigma$ is a measure on $X$. $T_\# \sigma$ is the push-forward measure of $\sigma$, i.e., its is the measure on $Y$ such that for all Borel set $A \subset Y$, $T_\# \sigma(A) = \sigma(T^{-1}(A))$.
- $\gamma, \gamma_1, \gamma_2$: $\gamma$ is a joint measure defined in a product space having $\gamma_1, \gamma_2$ as its first and second marginals, respectively. In the discrete setting, they are viewed as matrices and vectors, i.e., $\gamma \in \mathbb{R}^{n \times m}_+$, and $\gamma_1 = \gamma 1_m \in \mathbb{R}^n_+$, $\gamma_2 = \gamma^\top 1_n \in \mathbb{R}^m_+$.
- $\pi_1 : X \times Y \to X$, canonical projection mapping, with $(x, y) \mapsto x$. Similarly, $\pi_2 : X \times Y \to Y$ is canonical projection mapping, with $(x, y) \mapsto y$.
- $\pi_{1,2} : S \times X \times Y \to X \times Y$, canonical projection mapping, with $(s, x, y) \to (x, y)$. Similarly, $\pi_{0,1}$ maps $(s, x, y)$ to $(s, x)$; $\pi_{0,2}$ maps $(s, x, y)$ to $(s, y)$.
- $\Gamma(\mu, \nu)$, where $\mu \in \mathcal{P}_2(X), \nu \in \mathcal{P}_2(Y)$ (where $X, Y$ may not necessarily be the same set): it is the set of all the couplings (transportation plans) between $\mu$ and $\nu$, i.e., $\Gamma(\mu, \nu) := \{\gamma \in \mathcal{P}_2(X \times Y) : \gamma_1 = \mu, \gamma_2 = \nu\}$.
- $\Gamma(\mathrm{p}, \mathrm{q})$: set of all the couplings between the discrete probability measures $\mu = \sum_{i=1}^n p_i^X \delta_{x_i}$ and $\nu = \sum_{j=1}^m q_j^Y \delta_{y_j}$ with weight vectors

$$\mathrm{p} = [p_1^X, \ldots, p_n^X]^\top \qquad \text{and} \qquad \mathrm{q} = [q_1^Y, \ldots, q_m^Y]^\top. \tag{20}$$

That is, $\Gamma(\mathrm{p}, \mathrm{q})$ coincides with $\Gamma(\mu, \nu)$, but it is viewed as a subset of $n \times m$ matrices defined in (12).

- $p, q$: real numbers $1 \le p, q < \infty$.
- $\mathrm{p}, \mathrm{q}$: vectors of weights as in (20).
- $\mathrm{p} = [p_1, \ldots, p_n] \le \mathrm{p}' = [p'_1, \ldots, p'_n]$ if $p_j \le p'_j$ for all $1 \le j \le n$.

- $|\mathrm{p}| = \sum_{i=1}^{n} p_i$ for $\mathrm{p} = [p_1, \ldots, p_n]$.
- $c(x,y) : X \times Y \to \mathbb{R}_+$ denotes the cost function used for classical and partial optimal transport problems. lower-semi continuous function.
- $OT(\mu, \nu)$: it is the classical optimal transport (OT) problem between the probability measures $\mu$ and $\nu$ defined in (1).
- $W_p(\mu, \nu)$: it is the $p$-Wasserstein distance between the probability measures $\mu$ and $\nu$ defined in (2), for $1 \leq p < \infty$.
- $POT(\mu, \nu; \lambda)$: the Partial Optimal Transport (OPT) problem defined in (3).
- $|\mu|$: total variation norm of the positive Randon (finite) measure $\mu$ defined on a measurable space $X$, i.e., $|\mu| = \mu(X)$.
- $\mu \leq \sigma$: denotes that for all Borel set $B \subseteq X$ we have that the measures $\mu, \sigma \in \mathcal{M}_+(X)$ satisfy $\mu(B) \leq \sigma(B)$.
- $\Gamma_{\leq}(\mu, \nu)$, where $\mu \in \mathcal{M}_+(X), \nu \in \mathcal{M}_+(Y)$: set of all "partial transportation plans"
$$\Gamma_{\leq}(\mu, \nu) := \{\gamma \in \mathcal{M}_+(X \times Y) : \gamma_1 \leq \mu, \gamma_2 \leq \nu\}.$$
- $\Gamma_{\leq}(\mathrm{p}, \mathrm{q})$: set of all the "partial transportation plans" between the discrete probability measures $\mu = \sum_{i=1}^{n} p_i^X \delta_{x_i}$ and $\nu = \sum_{j=1}^{m} q_j^Y \delta_{y_j}$ with weight vectors $\mathrm{p} = [p_1^X, \ldots, p_n^X]$ and $\mathrm{q} = [q_1^Y, \ldots, q_m^Y]$. That is, $\Gamma_{\leq}(\mathrm{p}, \mathrm{q})$ coincides with $\Gamma_{\leq}(\mu, \nu)$, but it is viewed as a subset of $n \times m$ matrices defined in (13).
- $\lambda > 0$: positive real number.
- $\hat{\infty}$: auxiliary point.
- $\hat{X} = X \cup \{\hat{\infty}\}$.
- $\hat{\mu}, \hat{\nu}$: given in (4).
- $\hat{\mathrm{p}}, \hat{\mathrm{q}}$: given in (53).
- $\hat{\gamma}$: given in (6).
- $\hat{c}(\cdot, \cdot) : \hat{X} \times \hat{Y} \to \mathbb{R}_+$: cost as in (5).
- $L : \mathbb{R} \times \mathbb{R} \to \mathbb{R}$: cost function for the GW problems.
- $D : \mathbb{R} \times \mathbb{R} \to \mathbb{R}$: generic distance on $\mathbb{R}$ used for GW problems.
- $GW^L(\cdot, \cdot)$: GW optimization problem given in (7).
- $GW^p(\cdot, \cdot)$: GW optimization problem given in (7) when $L(a,b) = |a - b|^p$.
- $GW_q^L(\cdot, \cdot)$: general GW optimization problem for $g \geq 1$ given in (33).
- $GW_q^p(\cdot, \cdot)$: general GW optimization problem for $q \geq 1$ and $L(a,b) = |a - b|^p$ given in (34).
- $GW_{\lambda,q}^p(\cdot, \cdot)$: generalized GW problem given in (39).
- $\widehat{GW}$: GW-variant problem given in (51) for the general case, and in (55) for the discrete setting.
- $\hat{L}$: cost given in (16) for the GW-variant problem.
- $d : \hat{X} \times \hat{X} \to \mathbb{R}_+ \cup \{\infty\}$: "generalized" metric given in (50) for $\hat{X}$.
- $\mathbb{X} \sim \mathbb{Y}$: equivalence relation in for mm-spaces, $\mathbb{X} \sim \mathbb{Y}$ if and only if they have the same total mass and $GW_q^p(\mathbb{X}, \mathbb{Y}) = 0$.
- $PGW_{\lambda,q}^L(\cdot, \cdot)$: partial GW optimization problem given in (9) or, equivalently, in (10).
- $PGW_{\lambda,q}^p(\cdot, \cdot)$: partial GW optimization problem given in (10) when $L(a,b) = |a - b|^p$.
- $PGW_{\lambda}(\cdot, \cdot)$: is is the PGW problem $PGW_{\lambda,q}^p(\cdot, \cdot)$ for the case when $p = 2 = q$.
- $\mu(\phi)$: given a measure $\mu$ and a function $\phi$,
$$\mu(\phi) := \int \phi(x) d\mu(x).$$

- $C(\gamma; \lambda, \mu, \nu)$: the transportation cost induced by transportation plan $\gamma \in \Gamma_{\leq}(\mu, \nu)$ in the Partial GW problem 10,

$$C(\gamma; \lambda, \mu, \nu) := \gamma^{\otimes 2}(L(d_X^q, d_Y^q)) + \lambda(|\mu|^2 + |\nu|^2 - 2|\gamma|^2).$$

- $\mathcal{L}$: functional for the optimization problem $PGW_\lambda(\cdot, \cdot)$.

- $M$, $\tilde{M}$, and $\hat{M}$: see (14), and (54). Notice that, $(M - 2\lambda)_{i,i',j,j'} := M_{i,i',j,j'} - 2\lambda$.

- $\langle \cdot, \cdot \rangle_F$: Frobenius inner product for matrices, i.e., $\langle A, B \rangle_F = \text{trace}(A^\top B) = \sum_{i,j}^{n,m} A_{i,j} B_{i,j}$ for all $A, B \in \mathbb{R}^{n \times m}$.

- $M \circ \gamma$: product between the tensor $M$ and the matrix $\gamma$.

- $\nabla$: gradient.

- $[1:n] = \{1, \ldots, n\}$.

- $\alpha$: step size based on the line search method.

- $\gamma^{(1)}$: initialization of the algorithm.

- $\gamma^{(k)}$, $\gamma^{(k)'}$: previous and new transportation plans before and after step 1 in the $k-$th iteration of version 1 of our proposed FW algorithm.

- $\hat{\gamma}^{(k)}$, $\hat{\gamma}^{(k)'}$: previous and new transportation plans before and after step 1 in the $k-$th iteration of version 2 of our proposed FW algorithm.

- $G = 2\tilde{M} \circ \gamma$, $\hat{G} = 2\hat{M} \circ \hat{\gamma}$: Gradient of the objective function in version 1 and version 2, respectively, of our proposed FW algorithm for solving the discrete version of partial GW problem.

- $(\delta\gamma, a, b)$ and $(\delta\hat{\gamma}, a, b)$: given in (18) and (56) for versions 1 and 2 of the algorithm, respectively.

- $C^1$-function: continuous and with continuous derivatives.

- $MPGW_\rho(\cdot, \cdot)$: Mass-Constrained Partial Gromov-Wasserstein defined in (77).

- $\Gamma_{\leq}^\rho(\mu, \nu)$: set transportation plans defined in (78) for the Mass-Constrained Partial Gromov-Wasserstein problem.

## B  Proof of Proposition 3.2

The idea of the proof is inspired by the proof of Proposition 1 in (Piccoli & Rossi, 2014).

The goal is to verify that

$$PGW_{\lambda,q}^L(\mathbb{X}, \mathbb{Y})$$
$$:= \inf_{\gamma \in \mathcal{M}_+(X,Y)} \underbrace{\int_{(X \times Y)^2} L(d_X^q(x,x'), d_Y^q(y,y')) d\gamma^{\otimes 2}}_{\text{transport GW cost}} + \underbrace{\lambda \left( |\mu^{\otimes 2} - \gamma_1^{\otimes 2}| + |\nu^{\otimes 2} - \gamma_2^{\otimes 2}| \right)}_{\text{mass penalty}}$$
$$= \inf_{\gamma \in \Gamma_{\leq}(\mu,\nu)} \int_{(X \times Y)^2} L(d_X^q(x,x'), d_Y^q(y,y')) d\gamma^{\otimes 2} + \lambda \left( |\mu^{\otimes 2} - \gamma_1^{\otimes 2}| + |\nu^{\otimes 2} - \gamma_2^{\otimes 2}| \right). \quad (21)$$

Consider $\gamma \in \mathcal{M}_+(X \times Y)$ such that $\gamma_1 \leq \mu$ does not hold. Then we can write the Lebesgue decomposition of $\gamma_1$ with respect to $\mu$:

$$\gamma_1 = f\mu + \mu^\perp,$$

where $f \geq 0$ is the Radon-Nikodym derivative of $\gamma_1$ with respect to $\mu$, and $\mu^\perp, \mu$ are mutually singular, that is, there exist measurable sets $A, B$ such that $A \cap B = \emptyset$, $X = A \cup B$ and $\mu^\perp(A) = 0, \mu(B) = 0$. Without loss of generality, we can assume that the support of $f$ lies on $A$, since

$$\gamma_1(E) = \int_{E \cap A} f(x) \, d\mu(x) + \mu^\perp(E \cap B) \qquad \forall E \subseteq X \text{ measurable.}$$

Define $A_1 = \{x \in A : f(x) > 1\}, A_2 = \{x \in A : f(x) \leq 1\}$ (both are measurable, since $f$ is measurable), and define $\bar{\mu} = \min\{f, 1\}\mu$. Then,

$$\bar{\mu} \leq \mu \qquad \text{and} \qquad \bar{\mu} \leq f\mu \leq f\mu + \mu^{\perp} = \gamma_1.$$

There exists a $\bar{\gamma} \in \mathcal{M}_+(X \times Y)$ such that $\bar{\gamma}_1 = \bar{\mu}, \bar{\gamma} \leq \gamma$, and $\bar{\gamma}_2 \leq \gamma_2$. Indeed, we can construct $\bar{\gamma}$ in the following way: First, let $\{\gamma^x\}_{x \in X}$ be the set of conditional measures (disintegration) such that for every measurable (test) function $\psi : X \times Y \to \mathbb{R}$ we have

$$\int \psi(x, y) \, d\gamma(x, y) = \int_X \int_Y \psi(x, y) \, d\gamma^x(y) \, d\gamma_1(x).$$

Then, define $\bar{\gamma}$ as

$$\bar{\gamma}(U) := \int_X \int_Y \mathbb{1}_U(x, y) \, d\gamma^x(y) \, d\bar{\mu}(x) \qquad \forall U \subseteq X \times Y \text{ Borel}.$$

Then, $\bar{\gamma}$ verifies that $\bar{\gamma}_1 = \bar{\mu}$, and since $\bar{\mu} \leq \gamma_1$, we also have that $\bar{\gamma} \leq \gamma$, which implies $\bar{\gamma}_2 \leq \gamma_2$.

Since $|\gamma_1| = |\gamma_2|$ and $|\bar{\gamma}_1| = |\bar{\gamma}_2|$, then we have $|\gamma_1^{\otimes 2} - \bar{\gamma}_1^{\otimes 2}| = |\gamma_2^{\otimes 2} - \bar{\gamma}_2^{\otimes 2}|$.

We claim that

$$|\mu^{\otimes 2} - \gamma_1^{\otimes 2}| \geq |\mu^{\otimes 2} - \bar{\gamma}_1^{\otimes 2}| + |\gamma_1^{\otimes 2} - \bar{\gamma}_1^{\otimes 2}|. \tag{22}$$

- *Left-hand side of (22)*: Since $\{A, B\}$ is a partition of $X$, we first split the left-hand side of (22) as

$$|\mu^{\otimes 2} - \gamma_1^{\otimes 2}| = \underbrace{(\mu^{\otimes 2} - \gamma_1^{\otimes 2})(A \times A)}_{(I)} + \underbrace{(\mu^{\otimes 2} - \gamma_1^{\otimes 2})(A \times B) + (\mu^{\otimes 2} - \gamma_1^{\otimes 2})(B \times A)}_{(II)}$$
$$+ \underbrace{(\mu^{\otimes 2} - \gamma_1^{\otimes 2})(B \times B)}_{(III)}.$$

Then we have

$$(III) = (\mu^{\otimes 2} - \gamma_1^{\otimes 2})(B \times B) = \mu^{\perp} \otimes \mu^{\perp}(B \times B) = |\mu^{\perp}|^2,$$
$$(II) = (\mu^{\otimes 2} - \gamma_1^{\otimes 2})(A \times B) + (\mu^{\otimes 2} - \gamma_1^{\otimes 2})(B \times A) = 2|\mu^{\perp}|(\mu - \gamma_1)(A).$$

Since $\gamma_1 = f\mu$ in $A$, then $\bar{\gamma}_1 = \gamma_1$ in $A_2$ and $\bar{\gamma}_1 = \mu$ in $A_1$, so we have

$$(\mu - \gamma_1)(A) = (\mu - \gamma_1)(A_1) + (\mu - \gamma_1)(A_2) = (\gamma_1 - \bar{\gamma}_1)(A_1) + (\mu - \bar{\gamma}_1)(A_2)$$
$$= (\gamma_1 - \bar{\gamma}_1)(A) + (\mu - \bar{\gamma}_1)(A).$$

Thus,

$$(II) = 2|\mu^{\perp}|((\gamma_1 - \bar{\gamma}_1)(A) + (\mu - \bar{\gamma}_1)(A)),$$

and we also get that

$$(I) = (\mu^{\otimes 2} - \gamma_1^{\otimes 2})(A \times A)$$
$$= (\mu^{\otimes 2} - \gamma_1^{\otimes 2})(A_1 \times A_1) + (\mu^{\otimes 2} - \gamma_1^{\otimes 2})(A_2 \times A_2) + (\mu^{\otimes 2} - \gamma_1^{\otimes 2})(A_1 \times A_2)$$
$$\quad + (\mu^{\otimes 2} - \gamma_1^{\otimes 2})(A_2 \times A_1)$$
$$= (\gamma_1^{\otimes 2} - \bar{\gamma}_1^{\otimes 2})(A_1 \times A_1) + (\mu^{\otimes 2} - \bar{\gamma}_1^{\otimes 2})(A_2 \times A_2) +$$
$$\quad + |\bar{\gamma}_1 \otimes \mu - \gamma_1 \otimes \bar{\gamma}_1|(A_1 \times A_2) + |\mu \otimes \bar{\gamma}_1 - \bar{\gamma}_1 \otimes \gamma_1|(A_2 \times A_1)$$
$$= (\gamma_1^{\otimes 2} - \bar{\gamma}_1^{\otimes 2})(A_1 \times A_1) + (\mu^{\otimes 2} - \bar{\gamma}_1^{\otimes 2})(A_2 \times A_2) + 2(\bar{\gamma}_1 - \gamma_1)(A_1)(\mu - \bar{\gamma}_1)(A_2)$$
$$= (\gamma_1^{\otimes 2} - \bar{\gamma}_1^{\otimes 2})(A \times A) + (\mu^{\otimes 2} - \bar{\gamma}_1^{\otimes 2})(A \times A) + \underbrace{2(\bar{\gamma}_1 - \gamma_1)(A_1)(\mu - \bar{\gamma}_1)(A_2)}_{\geq 0}.$$

- *Right-hand side of (22):* First notice that

$$(\gamma_1 - \bar{\gamma}_1)(B) = (\gamma_1 - \bar{\gamma}_1)(B) \leq \gamma_1(B) = |\mu^{\perp}|,$$

and since $\bar{\gamma}_1 \leq \mu$ and $\mu(B) = 0$, we have

$$(\mu - \bar{\gamma}_1)(B) = 0.$$

Then,

$$
\begin{aligned}
|\mu^{\otimes 2} - \bar{\gamma}_1^{\otimes 2}| &+ |\gamma_1^{\otimes 2} - \bar{\gamma}_1^{\otimes 2}| = \\
&= (\mu^{\otimes 2} - \bar{\gamma}_1^{\otimes 2})(A \times A) + (\gamma_1^{\otimes 2} - \bar{\gamma}_1^{\otimes 2})(A \times A) + (\mu^{\otimes 2} - \bar{\gamma}_1^{\otimes 2})(B \times B) \\
&\quad + (\gamma_1^{\otimes 2} - \bar{\gamma}_1^{\otimes 2})(B \times B) + (\mu^{\otimes 2} - \bar{\gamma}_1^{\otimes 2})(A \times B) + (\gamma_1^{\otimes 2} - \bar{\gamma}_1^{\otimes 2})(A \times B) \\
&\quad + (\mu^{\otimes 2} - \bar{\gamma}_1^{\otimes 2})(B \times A) + (\gamma_1^{\otimes 2} - \bar{\gamma}_1^{\otimes 2})(B \times A) \\
&\leq \underbrace{(\mu^{\otimes 2} - \bar{\gamma}_1^{\otimes 2})(A \times A) + (\gamma_1^{\otimes 2} - \bar{\gamma}_1^{\otimes 2})(A \times A)}_{\leq (I)} + \underbrace{|\mu^\perp|^2}_{=(III)} + \underbrace{2|\mu^\perp|(\gamma_1 - \bar{\gamma}_1)(A)}_{=(II)}.
\end{aligned}
$$

Thus, (22) holds.

We finish the proof of the proposition by noting that

$$
\begin{aligned}
|\mu^{\otimes 2} - \bar{\gamma}_1^{\otimes 2}| + |\nu^{\otimes 2} - \bar{\gamma}_2^{\otimes 2}| &\leq |\mu^{\otimes 2} - \gamma_1^{\otimes 2}| - |\gamma_1^{\otimes 2} - \bar{\gamma}_1^{\otimes 2}| + |\nu^{\otimes 2} - \bar{\gamma}_2^{\otimes 2}| \\
&= |\mu^{\otimes 2} - \gamma_1^{\otimes 2}| - |\gamma_2^{\otimes 2} - \bar{\gamma}_2^{\otimes 2}| + |\nu^{\otimes 2} - \bar{\gamma}_2^{\otimes 2}| \\
&\leq |\mu^{\otimes 2} - \gamma_1^{\otimes 2}| + |\nu^{\otimes 2} - \gamma_2^{\otimes 2}|
\end{aligned}
$$

where the first inequality follows from (22), and the second inequality holds from the fact the total variation norm $|\cdot|$ satisfies triangular inequality. Therefore, $\bar{\gamma}$ induces a smaller transport GW cost than $\gamma$ (since $\bar{\gamma} \leq \gamma$), and also $\bar{\gamma}$ decreases the mass penalty in comparison that corresponding to $\gamma$. Thus, $\bar{\gamma}$ is a better GW transportation plan, which satisfies $\bar{\gamma}_1 \leq \mu$. Similarly, we can further construct $\bar{\gamma}'$ based on $\bar{\gamma}$ such that $\bar{\gamma}'_1 \leq \mu, \bar{\gamma}'_2 \leq \nu$. Therefore, we can restrict the minimization in (9) from $\mathcal{M}_+(X \times Y)$ to $\Gamma_\leq(\mu, \nu)$. Thus, the equality (21) is satisfied.

*Proof of Remark 3.1.* Given $\gamma \in \Gamma_\leq(\mu, \nu)$, since $\gamma_1 \leq \mu$, $\gamma_2 \leq \nu$, and $\gamma_1(X) = |\gamma_1| = |\gamma| = |\gamma_2| = \gamma_2(Y)$, we have

$$
\begin{aligned}
|\mu^{\otimes 2} - \gamma_1^{\otimes 2}| + |\nu^{\otimes 2} - \gamma_2^{\otimes 2}| &= \mu^{\otimes 2}(X^2) - \gamma_1^{\otimes 2}(X^2) + \nu^{\otimes 2}(Y^2) - \gamma_2^{\otimes 2}(Y^2) \\
&= |\mu|^2 + |\nu|^2 - 2|\gamma|^2,
\end{aligned}
$$

and so the transportation cost in partial GW problem (10) becomes

$$
\begin{aligned}
C&(\gamma; \lambda, \mu, \nu) \\
&:= \int_{(X \times Y)^2} L(d_X^q(x, x'), d_Y^q(y, y')) \, d\gamma(x, y) d\gamma(x', y') + \lambda \left( |\mu^{\otimes 2} - \gamma_1^{\otimes 2}| + |\nu^{\otimes 2} - \gamma_2^{\otimes 2}| \right) \\
&= \int_{(X \times Y)^2} L(d_X^q(x, x'), d_Y^q(y, y')) \, d\gamma(x, y) d\gamma(x', y') + \lambda \left( |\mu|^2 + |\nu|^2 - 2|\gamma|^2 \right) \\
&= \int_{(X \times Y)^2} (L(d_X^q(x, x'), d_Y^q(y, y') - 2\lambda) \, d\gamma(x, y) d\gamma(x', y') + \underbrace{\lambda \left( |\mu|^2 + |\nu|^2 \right)}_{\text{does not depend on } \gamma} .
\end{aligned}
\tag{23}
$$

$\square$

## C    PROOF OF PROPOSITION 3.3

In this section, we discuss the minimizer of the Partial GW problem (9). Trivially, $\Gamma_\leq(\mu, \nu) \subseteq \mathcal{M}_+(X \times Y)$ and by using Proposition 3.2 it is enough to show that a minimizer for problem (10) exists.

We refer the reader to Mémoli (2011, Chapters 5 and 10) for similar ideas.

## C.1 Formal Statement of Proposition 3.3

*Suppose $X, Y$ are compact sets, then exists compact set $[0, \beta] \subset \mathbb{R}$, such that*

$$d(x, x'),\, d(y, y') \in [0, \beta], \qquad \forall x, x' \in X, y,\, y' \in Y$$

*Let $A = [0, \beta^q]$. Let $L_{A^2}$ denote the restriction of $L$ on $A^2$, i.e. $L_{A^2} : A^2 \to \mathbb{R}$ with $L_{A^2}(r_1, r_2) = L(r_1, r_2), \forall r_1, r_2 \in A$. Suppose $L$ satisfies the following: there exists $0 < K < \infty$ such that for every $r_1, r_1', r_2, r_2' \in A$,*

$$|L_{A^2}(r_1, r_2) - L_{A^2}(r_1', r_2)| \le K|r_1 - r_1'|, \quad |L_{A^2}(r_1, r_2) - L_{A^2}(r_1, r_2')| \le K|r_2 - r_2'| \quad (24)$$

*(i.e., $L_{A^2}$ is Lipschitz on each variable). Then $PGW_\lambda^L(\cdot, \cdot)$ admits a minimizer.*

Note, the condition (24) contains the case $L(r_1, r_2) = |r_1 - r_2|^p$ as a special case:

**Lemma C.1.** *If $L(r_1, r_2) = |r_1 - r_2|^p$, for $1 \le p < \infty$, then $L$ satisfies the condition (24).*

*Proof.* Assume that $L$ is defined on an interval of the form $[0, M]$, for some $M > 0$. Consider $r_1, r_1', r_2, r_2' \in [0, M]$. If $p = 1$, by triangle inequality, we have

$$|L(r_1, r_2) - L(r_1', r_2)| = ||r_1 - r_2| - |r_1' - r_2|| \le |r_1 - r_1'|$$

and similarly,

$$|L(r_1, r_2) - L(r_1, r_2')| \le |r_2 - r_2'|.$$

From Mémoli (2011, page 473), since for $1 \le p < \infty$, the function $t \mapsto t^p$, for $t \in [0, M]$, is Lipschitz with constant bounded by $pM^{p-1}$, we have

$$|L(r_1, r_2) - L(r_1', r_2)| \le pM^{p-1}|r_1 - r_1'|.$$

and similarly,

$$|L(r_1, r_2) - L(r_1, r_2')| \le pM^{p-1}|r_2 - r_2'|.$$

$\square$

**Lemma C.2.** *Given $q \ge 1$, consider $\beta > 0$. Then $[0, \beta] \ni c \mapsto c^q \in [0, \beta^q]$ is a Lipschitz function.*

*Proof.* Given $c_1, c_2 \in [0, \beta]$, we have

$$|c_1^q - c_2^q| \le q\beta^{q-1}|c_1 - c_2| \qquad (25)$$

Thus, $c \mapsto c^q$ is a Lipschitz function. $\square$

## C.2 Convergence Auxiliary Result

If a sequence $\{\gamma^n\}$ converges weakly to $\gamma$, we write $\gamma^n \overset{w}{\to} \gamma$. In this setting, if $\gamma^n \overset{w}{\to} \gamma$, it does not imply that $(\gamma^n)^{\otimes 2} \overset{w}{\to} \gamma^{\otimes 2}$. Thus, the technique used in classical OT for proving the existence of a minimizer for the optimal transport optimization problem as a consequence of the Stone-Weierstrass theorem does not apply directly in the Gromov-Wasserstein context.

Inspired by (Mémoli, 2011), we introduce the following lemma.

**Lemma C.3.** *Given metric space $(Z, d_Z)$, suppose $\phi : Z^2 \to \mathbb{R}$ is a Lipschitz continuous function with respect to $(Z^2, d_Z^+)$, where*

$$d_Z^+((z_1, z_2), (z_1', z_2')) := d_Z(z_1, z_1') + d_Z(z_2, z_2'), \qquad \forall (z_1, z_2), (z_1', z_2') \in Z^2.$$

*Given $\gamma \in \mathcal{M}_+(Z)$, and a sequence $\{\gamma^n\}_{n \ge 1} \in \mathcal{M}_+(Z)$ such that converges weakly to $\gamma$,*

$$\gamma^n \overset{w}{\to} \gamma \qquad (n \to \infty).$$

*Finally, consider the mapping*

$$Z \ni z \mapsto \gamma(\phi(z, \cdot)) := \int_Z \phi(z, z') d\gamma(z') \in \mathbb{R}.$$

*Then we have the following results:*

*(1)* $\gamma^n(\phi(z,\cdot)) \to \gamma(\phi(z,\cdot))$ *uniformly (when* $n \to \infty$*).*

*(2)* $(\gamma^n)^{\otimes 2}(\phi(\cdot,\cdot)) \to \gamma^{\otimes 2}(\phi(\cdot,\cdot))$ *(when* $n \to \infty$*).*

*(3) If* $\mathcal{M} \subset \mathcal{M}_+(Z)$ *is compact for the weak convergence, then* $\inf_{\gamma \in \mathcal{M}} \gamma^{\otimes 2}(\phi(\cdot,\cdot))$ *admits a minimizer.*

*Proof.* The main idea of the proof is similar to Mémoli (2011, Lemma 10.3): we extend it from $\mathcal{P}_+(Z)$ to $\mathcal{M}_+(Z)$.

(1) Since $\gamma^n \overset{w}{\to} \gamma$, and $Z$ is compact, we have $|\gamma^n| \to |\gamma|$. Then, given $\epsilon > 0$, for $n$ sufficiently large we have $|\gamma^n| \le |\gamma| + \epsilon$.

Let us denote by $\|\phi\|_{Lip}$ the Lipschitz constant of $\phi$. For any $z_1, z_2 \in Z$, we have:

$$
\begin{aligned}
|\gamma^n(\phi(z_1,\cdot)) - \gamma^n(\phi(z_2,\cdot))| &\le \int_Z |\phi(z_1,z) - \phi(z_2,z)|\gamma^n(z) \\
&\le \max_{z \in Z} |\phi(z_1,z) - \phi(z_2,z)|(|\gamma| + \epsilon) \\
&\le (|\gamma| + \epsilon)\|\phi\|_{Lip}\, d_Z(z_1,z_2) = Kd_Z(z_1,z_2),
\end{aligned}
$$

where $K = (|\gamma| + \epsilon)\|\phi\|_{Lip}$ is a finite positive value. Note that the above inequality also holds if we replace $\gamma^n$ with $\gamma$.

Since $(Z, d_Z)$ is compact, $Z = \bigcup_{i=1}^N B(z_i, \epsilon/K)$ for some $z_1, \ldots, z_N \in Z$, where $B(z_i, \epsilon/3K) = \{z \in Z : d_Z(z, z_i) \le \epsilon/3K\}$ is the closed ball centered at $z_i$, with radius $\epsilon/K$. By definition of weak convergence, when $n$ is sufficiently large,

$$|\gamma^n(\phi(z_i,\cdot)) - \gamma(\phi(z_i,\cdot))| < \epsilon/3, \qquad \text{for each } i \in [1:N].$$

Given $z \in Z$, then $z \in B(z_i)$ for some $z_i$. For sufficiently large $n$, we have:

$$
\begin{aligned}
&|\gamma^n(\phi(z,\cdot)) - \gamma(\phi(z,\cdot))| \\
&\le |\gamma^n(\phi(z,\cdot)) - \gamma^n(\phi(z_i,\cdot))| + |\gamma^n(\phi(z_i,\cdot)) - \gamma(\phi(z_i,\cdot))| + |\gamma(\phi(z_i,\cdot)) - \gamma(\phi(z,\cdot))| \\
&\le Kd(z, z_i) + \epsilon/3 + Kd(z, z_i) = \epsilon/3 + \epsilon/3 + \epsilon/3 = \epsilon.
\end{aligned}
\tag{26}
$$

Thus, we prove the first statement.

(2) We recall that we do not have $(\gamma^n)^{\otimes 2} \overset{w}{\to} \gamma^{\otimes 2}$.

Consider an arbitrary $\epsilon > 0$. We have,

$$
0 \le \limsup_{n \to \infty} |(\gamma^n)^{\otimes 2}(\phi) - (\gamma)^{\otimes 2}(\phi)| \tag{27}
$$
$$
\le \limsup_{n \to \infty} \underbrace{|(\gamma^n \otimes \gamma^n)(\phi) - (\gamma \otimes \gamma^n)(\phi)|}_{A_n} + \limsup_{n \to \infty} \underbrace{|(\gamma^n \otimes \gamma)(\phi) - (\gamma \otimes \gamma)(\phi)|}_{B_n}.
$$

For the first term, when $n$ is sufficiently large, by statement (1), we have:

$$
\begin{aligned}
A_n &= \int (\gamma^n(\phi(z,\cdot)) - \gamma(\phi(z,\cdot))\, d\gamma^n(z) \\
&\le \max_z |\gamma^n(\phi(z,\cdot)) - \gamma(\phi(z,\cdot)||\gamma^n| \\
&\le \epsilon(|\gamma| + \epsilon)
\end{aligned}
\tag{28}
$$

Thus, $\limsup_n A = \lim_n A = 0$.

Similarly, for the second term, when $n$ is sufficiently large, we have

$$
B_n := \int (\gamma^n(\phi(z,\cdot)) - \gamma(\phi(z,\cdot)))d\gamma(z) \le \epsilon|\gamma|. \tag{29}
$$

Thus, $\limsup_n B_n = \lim_n B_n = 0$.

Therefore, from (27), (28) and (29), we obtain

$$
\limsup_{n \to \infty} |(\gamma^n)^{\otimes 2}(\phi) - (\gamma)^{\otimes 2}(\phi)| = \lim_{n \to \infty} |(\gamma^n)^{\otimes 2}(\phi) - (\gamma)^{\otimes 2}(\phi)| = 0. \tag{30}
$$

(3) Let $\gamma^n \in \mathcal{M}$ be a sequence such that $(\gamma^n)^{\otimes 2}(\phi)$ (weakly) converges to $\inf_{\gamma \in \mathcal{M}} \gamma^{\otimes 2}(\phi)$. Since $\mathcal{M}$ is compact, there exists a sub-sequence $\gamma^{n_k} \overset{w}{\to} \gamma$ for some $\gamma \in \mathcal{M}$. Then, by statement (2), we have:

$$\gamma^{\otimes 2}(\phi) = \lim_k (\gamma^{n_k})^{\otimes 2}(\phi) = \inf_{\gamma \in \mathcal{M}} \gamma^{\otimes 2}(\phi),$$

and we complete the proof.

$\square$

### C.3 Proof of the Formal Statement for Proposition 3.3

The proof follows the ideas of Mémoli (2011, Corollary 10.1).

Define $(Z, d_Z)$ as $Z := X \times Y$, with $d_Z((x,y),(x',y')) := d_X(x,x') + d_Y(y,y')$.

We claim that the following mapping

$$(X \times Y)^2 = Z^2 \to \mathbb{R}$$
$$((x,y),(x',y')) \mapsto \phi((x,y),(x',y')) := L(d_X^q(x,x'), d_Y^q(y,y')) - 2\lambda$$

is a Lipschitz function with respect to $d_Z^+$, where $L$ satisfies (24). Indeed, given $((x_1,y_1),(x_1',y_1')),((x_2,y_2),(x_2',y_2')) \in Z^2$, we have:

$$|\phi((x_1,y_1),(x_1',y_1')) - \phi((x_2,y_2),(x_2',y_2'))|$$
$$= |L(d_X(x_1,x_1'), d_Y(y_1,y_1')) - L(d_X(x_2,x_2'), d_Y(y_2,y_2'))|$$
$$\leq |L(d_X(x_1,x_1'), d_Y(y_1,y_1')) - L(d_X(x_2,x_2'), d_Y(y_1,y_1'))|$$
$$\quad + |L(d_X(x_2,x_2'), d_Y(y_1,y_1')) - L(d_X(x_2,x_2'), d_Y(y_2,y_2'))|$$
$$\leq K|d_X^q(x_1,x_1') - d_X^q(x_2,x_2')| + K|d_Y^q(y_1,y_1') - d_Y^q(y_2,y_2')|$$
$$\leq K'|d_X(x_1,x_1') - d_X(x_2,x_2')| + K'|d_Y(y_1,y_1') - d_Y(y_2,y_2')| \qquad (31)$$
$$\leq K'(d_X(x_1,x_2') + d_X(x_1',x_2')) + K'(d_Y(y_1,y_2) + d_Y(y_1',y_2')) \qquad (32)$$
$$= K'[((d_X(x_1,x_2) + d_Y(y_1,y_2)) + ((d_X(x_1',x_2') + d_Y(y_1',y_2')))]$$
$$= K'[d_Z((x_1,y_1),(x_2,y_2)) + d_Z((x_1',y_1'),(x_2',y_2'))]$$
$$= K' d_Z^+(((x_1,y_1),(x_2,y_2)),((x_1,y_1),(x_2,y_2)))$$

where in (31), $K' = q\beta^{q-1}K$; the inequality holds by lemma C.2; The inequality (32) follows from the triangle inequality:

$$d_X(x_1,x_1') - d_X(x_2,x_2') \leq d_X(x_1,x_2) + d_X(x_2,x_2') + d_X(x_2',x_1') - d_X(x_2,x_2')$$
$$= d_X(x_1,x_2) + d_X(x_1',x_2'),$$

and similarly,

$$d_X(x_2,x_2') - d_X(x_1,x_1') \leq d_X(x_1,x_2) + d_X(x_1',x_2').$$

Let $\mathcal{M} = \Gamma_{\leq}(\mu,\nu)$. From Liu et al. (2023, Proposition B.1), we have that $\Gamma_{\leq}(\mu,\nu)$ is a compact set with respect to the weak convergence topology.

By Lemma (C.3) part (3), we have the PGW problem, which can be written as

$$\inf_{\gamma \in \Gamma_{\leq}(\mu,\nu)} \gamma^{\otimes 2}(\phi) + \lambda(|\mu|^2 + |\nu|^2)$$

admits a solution, i.e., a minimizer $\gamma \in \Gamma_{\leq}(\mu,\nu)$. Therefore, we end the proof of Proposition 3.3.

## D Proof of Proposition 3.4: Metric Property of Partial GW

Let $L(r_1,r_2) = D^p(r_1,r_2)$ for a metric $D$ on $\mathbb{R}$, and since all the metrics in $\mathbb{R}$ are equivalent, for simplicity, consider $D(r_1,r_2) = |r_1 - r_2|$. (Notice that this satisfies the hypothesis of Proposition H.1 used in the experiments).

Consider the GW problem, for $q \geq 1$,

$$GW_q^L(\mathbb{X}, \mathbb{Y}) := \inf_{\gamma \in \Gamma(\mu, \nu)} \int_{(X \times Y)^2} L(d_X^q(x, x'), d_Y^q(y, y')) \, d\gamma^{\otimes 2}, \tag{33}$$

or, in particular,

$$GW_q^p(\mathbb{X}, \mathbb{Y}) := \inf_{\gamma \in \Gamma(\mu, \nu)} \int_{(X \times Y)^2} |d_X^q(x, x') - d_Y^q(y, y')|^p \, d\gamma^{\otimes 2}. \tag{34}$$

For probability mm-spaces we have the equivalence relation $\mathbb{X} \sim \mathbb{Y}$ if and only if $GW_q^p(\mathbb{X}, \mathbb{Y}) = 0$.

By Mémoli (2011, Chapter 5), $\mathbb{X} \sim \mathbb{Y}$ is equivalent to the following: there exists a bijective isometry mapping $\phi : X \to Y$, such that

$$d_X(x, x') - d_Y(\phi(x), \phi(x')) = 0, \quad \mu^{\otimes 2} - a.s.$$
$$\phi_{\#}\mu = \nu.$$

**Remark D.1.** *In the literature, the case where $q = 1$ is the most frequently considered problem. In particular, in (Mémoli, 2011) it is stated the equivalence relation $\mathbb{X} \sim \mathbb{Y}$ if and only if there exists $\phi : X \to Y$ such that $\phi_{\#}\mu = \nu$ and $d_X(x, x') = d_Y(\phi(x), \phi(x')) \, \mu^{\otimes 2} - a.s.$ if and only if $GW_1^p(\mathbb{X}, \mathbb{Y}) = 0$. Thus, $\mathbb{X} \sim \mathbb{Y}$ is also equivalent to have $\phi : X \to Y$ such that $\phi_{\#}\mu = \nu$ and $d_X(x, x') = d_Y(y, y') \, \gamma^{\otimes 2} - a.s.$ where $\gamma$ is a minimizer for $GW_1^p(\mathbb{X}, \mathbb{Y})$. So, in this situation we also have $d_X^q(x, x') = d_Y^q(y, y') \, \gamma^{\otimes 2} - a.s.$ for any given $q \geq 1$. Therefore, $\mathbb{X} \sim \mathbb{Y}$ if and only if $GW_q^p(\mathbb{X}, \mathbb{Y}) = 0$.*

### D.1 FORMAL STATEMENT OF PROPOSITION 3.4

We first introduce the formal statement of Proposition 3.4. To do so, we extend the equivalence relation $\sim$ to all mm-spaces (not only probability mm-spaces): Given arbitrary mm-spaces $\mathbb{X} = (X, d_X, \mu)$, $\mathbb{Y} = (Y, d_Y, \nu)$, where $X, Y$ are compact and $\mu \in \mathcal{M}_+(X)$, $\nu \in \mathcal{M}_+(Y)$, we write $\mathbb{X} \sim \mathbb{Y}$ if and only if they have the same total mass (i.e., $|\mu| = \mu(X) = \nu(Y) = |\nu|$) and $GW_q^p(\mathbb{X}, \mathbb{Y}) = 0$.

***Formal statement of Proposition 3.4:*** *Given $\lambda > 0$, $1 \leq p, q < \infty$, then $(PGW_{\lambda,q}^p(\cdot, \cdot))^{1/p}$ defines a metric among mm-spaces under taking quotient with respect to the equivalence relation $\sim$.*

Next, we discuss its proof.

### D.2 NON-NEGATIVITY AND SYMMETRY PROPERTIES

It is straightforward to verify $PGW_{\lambda,q}^p(\mathbb{X}, \mathbb{Y}) \geq 0$, and that $PGW_{\lambda,q}^p(\mathbb{X}, \mathbb{Y}) = PGW_{\lambda,q}^p(\mathbb{Y}, \mathbb{X})$. In what follows, we will concentrate on proving $PGW_{\lambda,q}^p(\mathbb{X}, \mathbb{Y}) = 0$ if and only if $\mathbb{X} \sim \mathbb{Y}$:

If $\mathbb{X} \sim \mathbb{Y}$, then $|\mu| = |\nu|$, and we have

$$0 \leq PGW_{\lambda,q}^p(\mathbb{X}, \mathbb{Y}) \leq GW_q^p(\mathbb{X}, \mathbb{Y}) = 0,$$

where the inequality follows from the fact $\Gamma(\mu, \nu) \subseteq \Gamma_{\leq}(\mu, \nu)$. Thus, $PGW_{\lambda,q}^p(\mathbb{X}, \mathbb{Y}) = 0$.

For the other direction, suppose that $PGW_{\lambda,q}^p(\mathbb{X}, \mathbb{Y}) = 0$. We claim that $|\mu| = |\nu|$ and that there exist an optimal plan $\gamma$ for $PGW_{\lambda,q}^p(\mathbb{X}, \mathbb{Y})$ such that $|\mu| = |\gamma| = |\nu|$. Let us prove this by contradiction. Assume $|\mu| < |\nu|$. For convenience, suppose $|\mu|^2 \leq |\nu|^2 - \epsilon$, for some $\epsilon > 0$. Then, for each $\gamma \in \Gamma_{\leq}(\mu, \nu)$, we have $|\gamma^{\otimes 2}| \leq |\mu|^2 \leq |\nu|^2 - \epsilon$, and so

$$PGW_{\lambda,q}^p(\mathbb{X}, \mathbb{Y}) \geq \lambda(|\mu|^2 + |\nu|^2 - 2|\gamma|^2) \geq \lambda(|\nu^2| - |\gamma|^2) \geq \lambda\epsilon > 0.$$

Thus, $PGW_{\lambda,q}^p(\mathbb{X}, \mathbb{Y}) > 0$, which is a contradiction. So, $|\mu| = |\nu|$. In addition, if $\gamma \in \Gamma_{\leq}(\mu, \nu)$ is optimal for $PGW_{\lambda,q}^p(\mathbb{X}, \mathbb{Y})$, we have $|\gamma| = |\mu| = |\nu|$, thus $\gamma \in \Gamma(\mu, \nu)$. Therefore, since $PGW_{\lambda,q}^p(\mathbb{X}, \mathbb{Y}) = 0$, and for such optimal $\gamma$ we have $|\gamma| = |\mu| = |\nu|$, we obtain

$$\int_{(X \times Y)^2} |d_X^q(x, x') - d_Y^q(y, y')|^p d\gamma^{\otimes 2} = 0.$$

As a result, $d_X^q(x, x') = d_Y^q(y, y')$ $\gamma^{\otimes 2} - a.s.$, which implies that $GW_q^p(\mathbb{X}, \mathbb{Y}) = 0$, and so $\mathbb{X} \sim \mathbb{Y}$.

### D.3 TRIANGLE INEQUALITY – STRATEGY: CONVERT THE PGW PROBLEM INTO A GW PROBLEM

Consider three arbitrary mm-spaces $\mathbb{S} = (S, d_S, \sigma)$, $\mathbb{X} = (X, d_X, \mu)$, $\mathbb{Y} = (Y, d_Y, \nu)$. We define $\hat{\mathbb{S}} = (\hat{S}, d_{\hat{S}}, \hat{\sigma})$, $\hat{\mathbb{X}} = (\hat{X}, d_{\hat{X}}, \hat{\mu})$, $\hat{\mathbb{Y}} = (\hat{Y}, d_{\hat{Y}}, \hat{\nu})$ in a similar way to that of Proposition G.1 but now aiming to have new spaces with equal total mass:

First, introduce auxiliary points $\hat{\infty}_0, \hat{\infty}_1, \hat{\infty}_2$ and set

$$\begin{cases} \hat{S} &= S \cup \{\hat{\infty}_0, \hat{\infty}_1, \hat{\infty}_2\}, \\ \hat{X} &= X \cup \{\hat{\infty}_0, \hat{\infty}_1, \hat{\infty}_2\}, \\ \hat{Y} &= Y \cup \{\hat{\infty}_0, \hat{\infty}_1, \hat{\infty}_2\}. \end{cases}$$

Define $\hat{\sigma}, \hat{\mu}, \hat{\nu}$ as follows:

$$\begin{cases} \hat{\sigma} &= \sigma + |\mu|\delta_{\hat{\infty}_1} + |\nu|\delta_{\hat{\infty}_2}, \\ \hat{\mu} &= \mu + |\sigma|\delta_{\hat{\infty}_0} + |\nu|\delta_{\hat{\infty}_2}, \\ \hat{\nu} &= \nu + |\sigma|\delta_{\hat{\infty}_0} + |\mu|\delta_{\hat{\infty}_1}. \end{cases} \tag{35}$$

Note that $\hat{\sigma}$ is not supported on point $\hat{\infty}_0$, similarly, $\hat{\mu}$ is not supported on $\hat{\infty}_1$, $\hat{\nu}$ is not supported on $\hat{\infty}_2$. In addition, we have $|\hat{\mu}| = |\hat{\nu}| = |\hat{\sigma}| = |\mu| + |\nu| + |\sigma|$. (For a similar idea in classical unbalanced optimal transport, see, for example, (Heinemann et al., 2023).)

Finally, define $d_{\hat{S}} : \hat{S}^2 \to \mathbb{R} \cup \{\infty\}$ as follows:

$$d_{\hat{S}}(s, s') = \begin{cases} d_S(s, s') & \text{if } (s, s') \in S^2, \\ \infty & \text{elsewhere.} \end{cases} \tag{36}$$

Note, $d_{\hat{S}}(\cdot, \cdot)$ is not a rigorous metric in $\hat{S}$ since we allow $d_{\hat{S}} = \infty$. Similarly, define $d_{\hat{X}}, d_{\hat{Y}}$. As a result, we have constructed new spaces

$$\hat{\mathbb{S}} = (\hat{S}, d_{\hat{S}}, \hat{\sigma}), \quad \hat{\mathbb{X}} = (\hat{X}, d_{\hat{X}}, \hat{\mu}), \quad \hat{\mathbb{Y}} = (\hat{Y}, d_{\hat{Y}}, \hat{\nu}). \tag{37}$$

We define the following mapping $D_\lambda : (\mathbb{R} \cup \{\infty\}) \times (\mathbb{R} \cup \{\infty\}) \to \mathbb{R}_+$:

$$D_\lambda^p(r_1, r_2) = \begin{cases} |r_1 - r_2|^p & \text{if } r_1, r_2 < \infty, \\ \lambda & \text{if } r_1 = \infty, r_2 < \infty \text{ or vice versa,} \\ 0 & \text{if } r_1 = r_2 = \infty. \end{cases} \tag{38}$$

Note that $D_\lambda$ is not a rigorous metric since it may sometimes violate triangle inequality. See the following lemma for a detailed and precise explanation.

**Lemma D.2.** *Let $D_\lambda(\cdot, \cdot)$ denote the function defined in (38). For any $r_0, r_1, r_2 \in \mathbb{R} \cup \{\infty\}$, we have the following:*

- *$D_\lambda(r_1, r_2) \geq 0$. $D_\lambda(r_1, r_2) = 0$ if and only if $r_1 = r_2$, where $r_1 = r_2$ denotes that $r_1 = r_2 \in \mathbb{R}$ or $r_1 = r_2 = \infty$.*

- *Except the case $r_1, r_2 \in \mathbb{R}, r_0 = \infty$, for all other cases, we have*

$$D_\lambda(r_1, r_2) \leq D_\lambda(r_1, r_0) + D_\lambda(r_2, r_0).$$

*Proof of Lemma D.2.* It is straightforward to verify $D_\lambda(\cdot, \cdot) \geq 0$.

Now, consider $r_0, r_1, r_2 \in \mathbb{R} \cup \{\infty\}$. If $r_1 = r_2 \in \mathbb{R}$ or $r_1 = r_2 = \infty$, we have $D_\lambda(r_1, r_2) = 0$. Otherwise, $D_\lambda(r_1, r_2) > 0$. So, $D_\lambda(r_1, r_2) = 0$ if and only if $r_1 = r_2$.

For the second item, we have the following cases:

Case 1: $r_1, r_2, r_0 \in \mathbb{R}$,

$$\begin{aligned} D_\lambda(r_1, r_2) &= |r_1 - r_2| \\ &\leq |r_1 - r_2| + |r_2 - r_0| \\ &= D_\lambda(r_0, r_1) + D_\lambda(r_0, r_2) \end{aligned}$$

Case 2: $r_1, r_2 \in \mathbb{R}, r_0 = \infty$. We do not need to verify the inequality in this case.

Case 3: $r_1 \in \mathbb{R}, r_2, r_0 = \infty$, or $r_1 = \infty, r_2 \in \mathbb{R}, r_0 = \infty$. In this case, we have

$$D_\lambda(r_1, r_2) = D_\lambda(r_1, r_0) = \sqrt{\lambda}, D_\lambda(r_2, r_0) = 0$$

and it is straightforward to verify the inequality.

Case 4: $r_1, r_2 = \infty, r_3 \in \mathbb{R}$. In this case, we have $D_\lambda(r_1, r_2) = 0 \leq D_\lambda(r_0, r_1) + D_\lambda(r_0, r_2)$.

Case 5: $r_1, r_2, r_0 = \infty$. In this case, we have

$$D_\lambda(r_1, r_2) = D_\lambda(r_1, r_0) = D_\lambda(r_2, r_0) = 0$$

and it is straightforward to verify the inequality. $\qquad\square$

We construct the following *generalized GW problem*:

$$GW_{\lambda,q}^p(\hat{\mathbb{X}}, \hat{\mathbb{Y}}) := \inf_{\hat\gamma \in \Gamma(\hat\mu, \hat\nu)} \underbrace{\int_{(\hat{X} \times \hat{Y})^2} D_\lambda^p(d_{\hat{X}}^q(x, x'), d_{\hat{Y}}^q(y, y')) \, d\hat\gamma^{\otimes 2}}_{\hat{C}(\hat\gamma; \lambda, \hat\mu, \hat\nu)}. \tag{39}$$

Similarly, we define $GW_{\lambda,q}^p(\hat{\mathbb{X}}, \hat{\mathbb{S}})$, and $GW_{\lambda,q}^p(\hat{\mathbb{S}}, \hat{\mathbb{Y}})$.

The mapping (6) is modified as:

$$\begin{aligned} \Gamma_\leq(\sigma, \mu) &\ni \gamma^{01} \mapsto \hat\gamma^{01} \in \Gamma(\hat\sigma, \hat\mu), \\ \hat\gamma^{01} &:= \gamma^{01} + (\sigma - \gamma_1^{01}) \otimes \delta_{\hat\infty_0} + \delta_{\hat\infty_1} \otimes (\mu - \gamma_2^{01}) + |\gamma|\delta_{\hat\infty_1, \hat\infty_0} + |\nu|\delta_{\hat\infty_2, \hat\infty_2}; \\ \Gamma_\leq(\sigma, \nu) &\ni \gamma^{02} \mapsto \hat\gamma^{02} \in \Gamma(\hat\sigma, \hat\nu), \\ \hat\gamma^{02} &:= \gamma^{02} + (\sigma - \gamma_1^{02}) \otimes \delta_{\hat\infty_0} + \delta_{\hat\infty_2} \otimes (\nu - \gamma_2^{02}) + |\gamma|\delta_{\hat\infty_2, \hat\infty_0} + |\mu|\delta_{\hat\infty_1, \hat\infty_1}; \\ \Gamma_\leq(\mu, \nu) &\ni \gamma^{12} \mapsto \hat\gamma^{12} \in \Gamma(\hat\mu, \hat\nu), \\ \hat\gamma^{12} &:= \gamma^{12} + (\mu - \gamma_1^{12}) \otimes \delta_{\hat\infty_1} + \delta_{\hat\infty_2} \otimes (\nu - \gamma_2^{12}) + |\gamma|\delta_{\hat\infty_2, \hat\infty_1} + |\mu|\delta_{\hat\infty_0, \hat\infty_0}. \tag{40} \end{aligned}$$

**Remark D.3.** *It is straightforward to verify the above mappings are well-defined. In addition, we can observe that, for each $\gamma^{01} \in \Gamma_\leq(\sigma, \mu), \gamma^{02} \in \Gamma_\leq(\sigma, \nu), \gamma^{12} \in \Gamma_\leq(\mu, \nu)$,*

$$\hat\gamma^{01}(\{\hat\infty_2\} \times X) = \hat\gamma^{01}(S \times \{\hat\infty_2\}) = 0, \tag{41}$$

$$\hat\gamma^{02}(\{\hat\infty_1\} \times Y) = \hat\gamma^{02}(S \times \{\hat\infty_1\}) = 0, \tag{42}$$

$$\hat\gamma^{12}(\{\hat\infty_0\} \times Y) = \hat\gamma^{12}(X \times \{\hat\infty_0\}) = 0.$$

**Proposition D.4.** *If $\gamma^{12} \in \Gamma_\leq(\mu, \nu)$ is optimal in PGW problem $PGW_{\lambda,q}^p(\mathbb{X}, \mathbb{Y})$, then $\hat\gamma^{12}$ defined in (40) is optimal in generalized GW problem $GW_{\lambda,q}^p(\hat{\mathbb{X}}, \hat{\mathbb{Y}})$. Furthermore, $\hat{C}(\hat\gamma^{12}; \lambda, \hat\mu, \hat\nu) = C(\gamma^{12}; \lambda, \mu, \nu)$, and thus,*

$$PGW_{\lambda,q}^p(\mathbb{X}, \mathbb{Y}) = GW_{\lambda,q}^p(\hat{\mathbb{X}}, \hat{\mathbb{Y}}).$$

*Proof of Proposition D.4.* For each $\gamma \in \Gamma_\leq(\mu, \nu)$, define $\hat\gamma$ by (40).

Note that if we merge the points $\hat\infty_1, \hat\infty_2, \hat\infty_3$ as $\hat\infty$, i.e.

$$\hat\infty = \hat\infty_1 = \hat\infty_2 = \hat\infty_3,$$

the value $\hat{C}(\hat\gamma; \lambda, \hat\mu, \hat\nu)$ will not change. Thus, we merge these three auxiliary points.

We have:

$$\hat{C}(\hat{\gamma}; \lambda, \hat{\mu}, \hat{\nu}) = \int_{(\hat{X} \times \hat{Y})^2} D_\lambda^p(d_{\hat{X}}^q(x, x'), d_{\hat{Y}}^q(x, x')) d\hat{\gamma}^{\otimes 2}$$

$$= \int_{(X \times Y)^2} |d_X^q(x, x') - d_Y^q(y, y')|^p d\hat{\gamma}^{\otimes 2} + \int_{(\{\hat{\infty}\} \times Y)^2} \lambda d\hat{\gamma}^{\otimes 2} + \int_{(X \times \{\hat{\infty}\})^2} \lambda \hat{\gamma}^{\otimes 2}$$

$$+ 2 \int_{(\{\hat{\infty}\} \times Y) \times (X \times Y)} \lambda d\hat{\gamma}^{\otimes 2} + 2 \int_{(X \times \{\hat{\infty}\}) \times (X \times Y)} \lambda d\hat{\gamma}^{\otimes 2} + \int_{(\{\hat{\infty}\} \times \{\hat{\infty}\})^2} D_\lambda^p(\infty, \infty) d\hat{\gamma}^{\otimes 2}$$

$$+ 2 \int_{(\{\hat{\infty}\} \times Y) \times (X \times \{\hat{\infty}\})} D_\lambda^p(\infty, \infty) d\hat{\gamma}^{\otimes 2} + 2 \int_{(\{\hat{\infty}\} \times \{\hat{\infty}\}) \times (X \times Y)} D_\lambda^p(\infty, \infty) d\hat{\gamma}^{\otimes 2}$$

$$+ 2 \int_{(\{\hat{\infty}\} \times \{Y\}) \times \{\hat{\infty}\}^2} D_\lambda^p(\infty, \infty) d\hat{\gamma}^{\otimes 2} + 2 \int_{(X \times \{\hat{\infty}\}) \times \{\hat{\infty}\}^2} D_\lambda^p(\infty, \infty) d\hat{\gamma}^{\otimes 2}$$

$$= \int_{(X \times Y)^2} |d_X^q(x, x') - d_Y^q(y, y')|^p d\gamma^{\otimes 2}$$

$$+ 2\lambda(|\nu| - |\gamma|)|\gamma| + \lambda(|\nu| - |\gamma|)^2 + 2\lambda(|\mu| - |\gamma|)|\gamma| + \lambda(|\mu| - |\gamma|)^2$$

$$= \int_{(X \times Y)^2} |d_X^q(x, y') - d_Y^q(y, y')|^p d\gamma^{\otimes 2}) + \lambda(|\nu^2| + |\mu|^2 - 2|\gamma|^2) = C(\gamma; \lambda, \mu, \nu).$$

As we merged the points $\hat{\infty}_1, \hat{\infty}_2, \hat{\infty}_3$, by Bai et al. (2023, Proposition B.1.), the mapping $\gamma \mapsto \hat{\gamma}$ defined in (40) is a bijection. Then, if $\gamma \in \Gamma_\leq(\mu, \nu)$ is optimal for the PGW problem $PGW_{\lambda,q}^p(\mathbb{X}, \mathbb{Y})$ (defined in (10)), $\hat{\gamma} \in \Gamma(\hat{\mu}, \hat{\nu})$ is optimal for generalized GW problem $GW_{\lambda,q}^p(\hat{\mathbb{X}}, \hat{\mathbb{Y}})$ (defined in (39)). Therefore,

$$GW_{\lambda,q}^p(\hat{\mathbb{X}}, \hat{\mathbb{Y}}) = PGW_{\lambda,q}^p(\mathbb{X}, \mathbb{Y}).$$

$\square$

**Proposition D.5** (Triangle inequality for $GW_{\lambda,q}^p(\cdot, \cdot)$). *Consider the generalized GW problem (39). Then, for any $p \in [1, \infty)$, we have*

$$GW_{\lambda,q}^p(\hat{\mathbb{X}}, \hat{\mathbb{Y}}) \leq GW_{\lambda,q}^p(\hat{\mathbb{S}}, \hat{\mathbb{X}}) + GW_{\lambda,q}^p(\hat{\mathbb{S}}, \hat{\mathbb{Y}}).$$

*Proof of Proposition D.5.* We prove the case $p = 2$. For general $p \geq 1$, it can be proved similarly.

Choose an optimal $\gamma^{12} \in \Gamma_\leq(\mu, \nu)$ for $PGW_{\lambda,q}^2(\mathbb{X}, \mathbb{Y})$, an optimal $\gamma^{01} \in \Gamma_\leq(\sigma, \mu)$ for $PGW_{\lambda,q}^2(\mathbb{S}, \mathbb{X})$, and an optimal $\gamma^{02} \in \Gamma_\leq(\sigma, \nu)$ for $PGW_{\lambda,q}^2(\mathbb{S}, \mathbb{Y})$. Construct $\hat{\gamma}^{12}, \hat{\gamma}^{01}, \hat{\gamma}^{02}$ by (40).

By Proposition D.4, we have that $\hat{\gamma}^{12}, \hat{\gamma}^{01}, \hat{\gamma}^{02}$ are optimal for $GW_{\lambda,q}^2(\hat{\mathbb{X}}, \hat{\mathbb{Y}})$, $GW_{\lambda,q}^2(\hat{\mathbb{S}}, \hat{\mathbb{X}})$, $GW_{\lambda,q}^2(\hat{\mathbb{S}}, \hat{\mathbb{Y}})$, respectively.

Define canonical projection mapping

$$\pi_{0,1} : (\hat{S} \times \hat{X} \times \hat{Y}) \to (\hat{S} \times \hat{X})$$
$$(s, x, y) \mapsto (s, x).$$

Similarly, we define $\pi_{0,2}, \pi_{1,2}$.

By *gluing lemma* (see Lemma 5.5 (Santambrogio, 2015)), there exists $\hat{\gamma} \in \mathcal{M}_+(\hat{S} \times \hat{X} \times \hat{Y})$, such that $(\pi_{0,1})_\# \hat{\gamma} = \hat{\gamma}^{01}, (\pi_{0,2})_\# \hat{\gamma} = \hat{\gamma}^{02}$. Thus, $(\pi_{1,2})_\# \hat{\gamma}$ is a coupling between $\hat{\mu}, \hat{\nu}$. We have

$$GW_{\lambda,q}^2(\mathbb{X}, \mathbb{Y}) = \int_{(\hat{X} \times \hat{Y})^2} D_\lambda^2(d_{\hat{X}}^q(x, x'), d_{\hat{Y}}^q(y, y')) d(\hat{\gamma}^{12})^{\otimes 2}$$

$$\leq \int_{(\hat{S} \times \hat{X} \times \hat{Y})^2} D_\lambda^2(d_{\hat{X}}^q(x, x'), d_{\hat{Y}}^q(y, y')) d\hat{\gamma}^{\otimes 2}. \tag{43}$$

The inequality holds since $(\pi_{1,2})_\# \hat\gamma, \hat\gamma^{12} \in \Gamma(\hat\mu, \hat\nu)$, and $\hat\gamma^{12}$ is optimal.

Next, we will show that

$$\int_{(\hat{S} \times \hat{X} \times \hat{Y})^2} D_\lambda^2(d_{\hat{X}}^q(x, x'), d_{\hat{Y}}^q(y, y'))d\hat\gamma^{\otimes 2}$$
$$\leq \int_{(\hat{S} \times \hat{X} \times \hat{Y})^2} (D_\lambda(d_{\hat{S}}^q(s, s'), d_{\hat{X}}^q(x, x')) + D_\lambda(d_{\hat{S}}^q(s, s'), d_{\hat{Y}}^q(y, y')))^2 d\hat\gamma^{\otimes 2}.$$

Let $((s, x, y), (s', x', y')) \in (\hat{S}, \hat{X}, \hat{Y})^2$, and assume that

$$D_\lambda(d_{\hat{X}}^2(x, x'), d_{\hat{Y}}^2(y, y')) > D_\lambda(d_{\hat{S}}^2(s, s'), d_{\hat{X}}^2(x, x')) + D_\lambda(d_{\hat{S}}^2(s, s'), d_{\hat{Y}}^2(y, y')). \tag{44}$$

By Lemma D.2, (44) implies $d_{\hat{X}}(x, x'), d_{\hat{Y}}(y, y') \in \mathbb{R}, d_{\hat{S}}(s, s') = \infty$. Thus, by definition (36), it also implies

$$(x, x') \in X^2, (y, y') \in Y^2, (s, s') \in \hat{S}^2 \setminus S^2. \tag{45}$$

Define the following sets:

$$A_\alpha = \hat{S} \times X \times Y,$$
$$A_0 = \{\hat\infty_0\} \times X \times Y,$$
$$A_1 = \{\hat\infty_1\} \times X \times Y,$$
$$A_2 = \{\hat\infty_2\} \times X \times Y.$$

Notice that, $(44) \implies (45)$ is equivalent to

$$(44) \implies ((s, x, y), (s, x', y')) \in A := \bigcup_{i=0}^{2}(A_i \times A_\alpha) \cup \bigcup_{i=0}^{2}(A_\alpha \times A_i). \tag{46}$$

Next, we will show $\hat\gamma^{\otimes 2}(A) = 0$. Indeed,

$$\hat\gamma(A_0) \leq \hat\gamma(\{\infty_0\} \times \hat{X} \times \hat{Y}) = \hat\sigma(\{\infty_0\}) = 0 \qquad \text{by definition (35) of } \hat\sigma \ ,$$
$$\hat\gamma(A_1) \leq \hat\gamma(\{\infty_1\} \times \hat{X} \times Y) = \hat\gamma^{02}(\{\hat\infty_1 \times Y\}) = 0 \qquad \text{by (42)},$$
$$\hat\gamma(A_2) \leq \hat\gamma(\{\infty_2\} \times X \times \hat{Y}) = \hat\gamma^{01}(\{\hat\infty_2 \times X\}) = 0 \qquad \text{by (41)}.$$

Thus, $\hat\gamma^{\otimes 2}(A) = 0$. By considering $B = (\hat{S} \times \hat{X} \times Y)^2 \setminus A$, we obtain

$$\int_{(\hat{S} \times \hat{X} \times \hat{Y})^2} D_\lambda^2(d_{\hat{X}}^q(x, x'), d_{\hat{Y}}^q(y, y'))d\gamma^{\otimes 2}$$
$$= \int_B D_\lambda^2(d_{\hat{X}}^q(x, x'), d_{\hat{Y}}^q(y, y'))d\gamma^{\otimes 2} \qquad \text{since } \gamma^{\otimes 2}(A) = 0$$
$$\leq \int_B \left(D_\lambda(d_{\hat{S}}^q(s, s'), d_{\hat{X}}^q(x, x') + D_\lambda(d_{\hat{S}}^q(s, s'), d_{\hat{Y}}^q(y, y'))\right)^2 d\gamma^{\otimes 2} \qquad \text{by (46)}$$
$$\leq \int_{(\hat{S} \times \hat{X} \times \hat{Y})^2} \left(D_\lambda(d_{\hat{S}}^q(s, s'), d_{\hat{X}}^q(x, x') + D_\lambda(d_{\hat{S}}^q(s, s'), d_{\hat{Y}}^q(y, y'))\right)^2 d\gamma^{\otimes 2}. \tag{47}$$

Following (43) and (47), we have

$$
\begin{aligned}
GW_{\lambda,q}^2(\hat{\mathbb{X}}, \hat{\mathbb{Y}}) &\leq \left( \int_{(\hat{S} \times \hat{X} \times \hat{Y})^2} D_\lambda^2(d_{\hat{X}}^q(x,x'), d_{\hat{Y}}^q(y,y')) d\hat{\gamma}^{\otimes 2} \right)^{1/2} \\
&\leq \left( \int_{(\hat{S} \times \hat{X} \times \hat{Y})^2} \left( D_\lambda(d_{\hat{S}}^q(s,s'), d_{\hat{X}}^q(x,x')) + D_\lambda(d_{\hat{S}}^q(s,s'), d_{\hat{Y}}^q(y,y')) \right)^2 d\gamma^{\otimes 2} \right)^{1/2} \\
&\leq \left( \int_{(\hat{S} \times \hat{X} \times \hat{Y})^2} D_\lambda^2(d_{\hat{S}}^q(s,s'), d_{\hat{X}}^q(x,x')) d\gamma^{\otimes 2} \right)^{1/2} \\
&\quad + \left( \int_{(\hat{S} \times \hat{X} \times \hat{Y})^2} D_\lambda^2(d_{\hat{S}}^q(s,s'), d_{\hat{Y}}^q(y,y')) d\gamma^{\otimes 2} \right)^{1/2} \\
&= \left( \int_{(\hat{S} \times \hat{X} \times \hat{Y})^2} D_\lambda^2(d_{\hat{S}}^q(s,s'), d_{\hat{X}}^q(x,x')) d(\gamma^{01})^{\otimes 2} \right)^{1/2} \\
&\quad + \left( \int_{(\hat{S} \times \hat{X} \times \hat{Y})^2} D_\lambda^2(d_{\hat{S}}^q(s,s'), d_{\hat{Y}}^q(y,y')) d(\gamma^{02})^{\otimes 2} \right)^{1/2} \\
&= GW_{\lambda,q}^2(\hat{\mathbb{S}}, \hat{\mathbb{X}}) + GW_{\lambda,q}^2(\hat{\mathbb{S}}, \hat{\mathbb{Y}}),
\end{aligned}
\tag{48}
$$

where in the third inequality (48) we used the Minkowski inequality in $L^2((\hat{S} \times \hat{X} \times \hat{Y})^2, \hat{\gamma}^{\otimes 2})$. $\qquad \square$

Now, we can complete the proof of Proposition 3.4: By the Propositions D.4, we have
$$
PGW_{\lambda,q}^p(\mathbb{X}, \mathbb{Y}) = GW_{\lambda,q}^p(\hat{\mathbb{X}}, \hat{\mathbb{Y}})
$$
and similarly for $PGW_{\lambda,q}^p$ and $(\mathbb{S}, \mathbb{X}), PGW_{\lambda,q}^p(\mathbb{S}, \mathbb{Y})$. By the Proposition D.5, $GW_{\lambda,q}^p(\cdot, \cdot)$ satisfies the triangle inequality, thus We complete the proof:
$$
\begin{aligned}
PGW_{\lambda,q}^p(\mathbb{X}, \mathbb{Y}) &= GW_{\lambda,q}^p(\hat{\mathbb{X}}, \hat{\mathbb{Y}}) \\
&\leq GW_{\lambda,q}^p(\hat{\mathbb{S}}, \hat{\mathbb{X}}) + GW_{\lambda,q}^p(\hat{\mathbb{S}}, \hat{\mathbb{Y}}) \\
&= PGW_{\lambda,q}^p(\mathbb{S}, \mathbb{X}) + PGW_{\lambda,q}^p(\mathbb{S}, \mathbb{Y}).
\end{aligned}
$$

## E    PROOF OF PROPOSITION 3.5: PGW CONVERGES TO GW AS $\lambda \to \infty$.

In the main text, we set $\lambda \in \mathbb{R}$. In this section, we discuss the limit case when $\lambda \to \infty$.

**Lemma E.1.** *Suppose $|\mu| \leq |\nu|$, for each $\gamma \in \Gamma_\leq(\mu, \nu)$, there exists $\gamma' \in \Gamma_\leq(\mu, \nu)$ such that $\gamma \leq \gamma'$ and $(\pi_1)_{\#} \gamma' = \mu$.*

*Proof.* Let $\gamma \in \Gamma_\leq(\mu, \nu)$.

If $|\gamma| = |\mu|$, then we have $(\pi_1)_{\#} \gamma = \mu$.

If $|\gamma| < |\mu|$, let $\mu^r = \mu - (\pi_1)_{\#} \gamma, \nu^r = \nu - (\pi_2)_{\#} \gamma$. We have that $\mu^r, \nu^r$ are non-negative measures, with $|\mu^r| = |\mu| - |\gamma| > 0$. If we define
$$
\gamma' := \gamma + \frac{1}{|\nu| - |\gamma|} \mu^r \otimes \nu^r,
$$
we obtain $\gamma \leq \gamma'$. In addition, we have:
$$
(\pi_1)_{\#} \gamma' = (\pi_1)_{\#} \gamma + \mu^r \frac{|\nu^r|}{|\nu| - |\gamma|} = (\pi_1)_{\#} \gamma + \mu^r = \mu,
$$
$$
(\pi_2)_{\#} \gamma' = (\pi_2)_{\#} \gamma + \nu^r \frac{|\mu^r|}{|\nu| - |\gamma|} \leq (\pi_2)_{\#} \gamma + \nu^r \frac{|\nu^r|}{|\nu| - |\gamma|} = \nu.
$$
Thus, $\gamma' \in \Gamma_\leq(\mu, \nu)$ and $(\pi_1)_{\#} \gamma' = \mu$. $\qquad \square$

**Lemma E.2.** *Given general mm-spaces $\mathbb{X} = (X, d_X, \mu)$, $\mathbb{Y} = (Y, d_Y, \nu)$, where $\mu, \nu$ are supported on bounded sets (in general, it is assumed that $X$ and $Y$ are compact, and that $supp(\mu) = X$, $supp(\nu) = Y$), consider the problem the problem $PGW_{\lambda,q}^L(\mathbb{X}, \mathbb{Y})$ with $L(r_1, r_2)$ a continuous functions. If $\lambda$ is sufficiently large, in particular:*

$$\lambda \geq \max_{\substack{x,x' \in supp(\mu) \\ y,y' \in supp(\nu)}} L(d_X(x,x'), d_Y(y,y')),$$

*then there exists optimal $\gamma$ for $PGW_\lambda(\mathbb{X}, \mathbb{Y})$ such that $|\gamma| = \min(|\mu|, |\nu|)$.*

*Furthermore, when*

$$\lambda > \max_{\substack{x,x' \in supp(\mu) \\ y,y' \in supp(\nu)}} L(d_X(x,x'), d_Y(y,y')),$$

*the for all optimal $\gamma \in \Gamma_\leq(\mu, \nu)$, we have $|\gamma| = \min(|\mu|, |\nu|)$.*

*Proof.* We prove it for $q = 1$. For a general $q \geq 1$, it can be proved similarly.

Without loss of generality, suppose $|\mu| \leq |\nu|$.

Since $\mu, \nu$ are supported on bounded sets, there exists $A = [0, M]$ such that $d_X(x, x'), d_Y(y, y') \in A$ for all $x, x' \in \text{supp}(\mu), y, y' \in \text{supp}(\nu)$.

Thus, the restriction of $L$ on $A^2$, denoted as $L_{A^2}$, is continuous on $A^2$, and thus it is bounded. So, consider

$$\text{m} := \max_{r_1, r_2 \in A} (L(r_1, r_2)) \geq L(d_X(x, x'), d_Y(y, y')), \quad \forall x, x' \in \text{supp}(\mu), y, y' \in \text{supp}(\nu).$$

Suppose $2\lambda \geq \text{m}+1$, and assume that there exists a optimal $\gamma \in \Gamma_\leq(\mu, \nu)$ such that $|\gamma| < |\mu|$. By Lemma E.1, there exists $\gamma'$ such that $\gamma \leq \gamma', (\pi_1)_\# \gamma' = \mu$. Thus, we have

$$C(\gamma'; \lambda, \mu, \nu) - C(\gamma; \lambda, \mu, \nu) = \int_{(X \times Y)} L(d_X(x, x'), d_Y(y, y')) - 2\lambda \, d((\gamma')^{\otimes 2} - (\gamma)^{\otimes 2})$$

$$\leq \int_{(X \times Y)} \text{m} - 2\lambda \, d((\gamma')^{\otimes 2} - (\gamma)^{\otimes 2})$$

$$= -(|\gamma'|^2 - |\gamma|^2) = -(|\mu|^2 - |\gamma|^2) < 0,$$

which is contradiction since $\gamma$ is optimal, and so we have completed the proof. $\square$

**Lemma E.3.** *Consider probability mm-spaces $\mathbb{X} = (X, d_X, \mu)$, $\mathbb{Y} = (Y, d_Y, \nu)$, that is, with $|\mu| = |\nu| = 1$. Then, for each $\lambda > 0$, we have*

$$PGW_{\lambda,q}^L(\mathbb{X}, \mathbb{Y}) \leq GW_q^L(\mathbb{X}, \mathbb{Y}).$$

*Proof.* In this setting, we have $\Gamma(\mu, \nu) \subset \Gamma_\leq(\mu, \nu)$, and thus

$$PGW_{\lambda,q}^L(\mathbb{X}, \mathbb{Y})$$

$$= \inf_{\Gamma \in \Gamma_\leq(\mu,\nu)} \int_{(X \times Y)^2} L(d_X^q(x, x'), d_Y^q(y, y')) d\gamma^{\otimes 2} + \lambda(|\mu|^2 + |\nu|^2 - 2|\gamma|^2)$$

$$\leq \inf_{\gamma \in \Gamma(\mu,\nu)} \int_{(X \times Y)^2} L(d_X^q(x, x'), d_Y^q(y, y')) + \lambda(|\mu|^2 + |\nu|^2 - 2|\gamma|^2) d\gamma^{\otimes 2}$$

$$= \inf_{\gamma \in \Gamma(\mu,\nu)} \int_{(X \times Y)^2} L(d_X^q(x, x'), d_Y^q(y, y')) d\gamma^{\otimes 2}$$

$$= GW_q^L(\mathbb{X}, \mathbb{Y}).$$

$\square$

Based on the above properties, we can now prove Proposition 3.5:

**Proposition E.4** (Generalization of Proposition 3.5)**.** *Consider general probability mm-spaces* $\mathbb{X} = (X, d_X, \mu)$, $\mathbb{Y} = (Y, d_Y, \nu)$, *that is, with* $|\mu| = |\nu| = 1$*, where* $X, Y$ *are bounded. Assume that* $L$ *is continuous. Then*

$$\lim_{\lambda \to \infty} PGW_{\lambda,q}^L(\mathbb{X}, \mathbb{Y}) = GW_q^L(\mathbb{X}, \mathbb{Y}).$$

*Proof.* When $\lambda$ is sufficiently large, by Lemma E.2, for each optimal $\gamma_\lambda \in \Gamma_{\leq}(\mu, \nu)$ of the minimization problem $PGW_{\lambda,q}^L(\mathbb{X}, \mathbb{Y})$, we have $|\gamma_\lambda| = \min(|\mu|, |\nu|) = 1$. That is, $\gamma_\lambda \in \Gamma(\mu, \nu)$. Plugging $\gamma_\lambda$ into $C(\gamma_\lambda; \lambda, \mu, \nu)$, we obtain:

$$PGW_{\lambda,q}^L(\mathbb{X}, \mathbb{Y}) = \int_{(X \times Y)^2} L(d_X^q(x, x'), d_Y^q(y, y')) d\gamma_\lambda^{\otimes 2} + \lambda(1^2 + 1^2 - 2 \cdot 1^2)$$

$$= \int_{(X \times Y)^2} L(d_X^q(x, x'), d_Y^q(y, y')) d\gamma_\lambda^{\otimes 2} \geq GW(\mathbb{X}, \mathbb{Y}).$$

By Lemma E.3, we also have $PGW_{\lambda,q}^L(\mathbb{X}, \mathbb{Y}) \leq GW_q^L(\mathbb{X}, \mathbb{Y})$ and we complete the proof. $\square$

## F  TENSOR PRODUCT COMPUTATION

**Lemma F.1.** *Given a tensor* $M \in \mathbb{R}^{n \times m \times n \times n}$ *and* $\gamma, \gamma' \in \mathbb{R}^{n \times m}$*, the tensor product operator* $M \circ \gamma$ *satisfies the following:*

(i) *The mapping* $\gamma \mapsto M \circ \gamma$ *is linear with respect to* $\gamma$*.*

(ii) *If* $M$ *is symmetric, in particular,* $M_{i,j,i',j'} = M_{i',j',i,j}, \forall i, i' \in [1:n], j, j' \in [1:m]$*, then*

$$\langle M \circ \gamma, \gamma' \rangle_F = \langle M \circ \gamma', \gamma \rangle_F.$$

*Proof.*

(i) For the first part, consider $\gamma, \gamma' \in \mathbb{R}^{n \times m}$ and $k \in \mathbb{R}$. For each $i, j \in [1:n] \times [1:m]$, we have we have

$$(M \circ (\gamma + \gamma'))_{ij} = \sum_{i',j'} M_{i,j,i',j'}(\gamma + \gamma')_{i'j'}$$

$$= \sum_{i',j'} M_{i,j,i',j'} \gamma_{i'j'} + \sum_{i',j'} M_{i,j,i',j'} \gamma'_{i'j'}$$

$$= (M \circ \gamma)_{ij} + (M \circ \gamma)_{i'j'},$$

$$(M \circ (k\gamma))_{ij} = \sum_{i',j'} M_{i,j,i',j'}(k\gamma)_{ij}$$

$$= k \sum_{i',j'} M_{i,j,i',j'} \gamma_{ij}$$

$$= k(M \circ \gamma)_{ij}.$$

Thus, $M \circ (\gamma + \gamma') = M \circ \gamma + M \circ \gamma'$ and $M \circ (k\gamma) = kM \circ \gamma$. Therefore, $\gamma \mapsto M \circ \gamma$ is linear.

(ii) For the second part, we have

$$\langle M \circ \gamma, \gamma' \rangle_F = \sum_{iji'j'} M_{i,j,i',j'} \gamma_{ij} \gamma'_{i'j'}$$

$$= \sum_{i,j,i',j'} M_{i',j',i,j} \gamma_{i',j'} \gamma_{i,j} \qquad (49)$$

$$= \langle M\gamma', \gamma \rangle$$

where (49) follows from the fact that $M$ is symmetric.

$\square$

## G  ANOTHER ALGORITHM FOR COMPUTING PGW DISTANCE – SOLVER 2

Our Algorithm 2 for solving the proposed PGW problem is based on a theoretical result that relates GW and PGW. The details of our computational method, as well as the proof of Proposition G.1 stated below, are provided in Appendix G.1. Based on such a proposition, we extend the PGW problem to a discrete *GW-variant* problem (55), leading to a solution for the original PGW problem by truncating the GW-variant solution.

**Proposition G.1.** *Let $\mathbb{X} = (X, d_X, \mu)$ be a mm-space. Consider an auxiliary point $\hat{\infty}$ and let $\hat{\mathbb{X}} = (\hat{X}, d_{\hat{X}}, \hat{\mu})$, where $\hat{X} = X \cup \{\hat{\infty}\}$, $\hat{\mu}$ is constructed by (4), and considering $\infty$ as an auxiliary point to $\mathbb{R}$ such that $x \leq \infty$ for every $x \in \mathbb{R}$, we extend $d_X$ into $d_{\hat{X}} : \hat{X}^2 \to \mathbb{R} \cup \{\infty\}$ and define $L_\lambda : \mathbb{R} \cup \{\infty\} \to \mathbb{R}$ as follows:*

$$d_{\hat{X}}(x, x') = \begin{cases} d_X(x, x') & \text{if } x, x' \in X \\ \infty & \text{otherwise} \end{cases}, L_\lambda(r_1, r_2) := \begin{cases} L(r_1, r_2) - 2\lambda & \text{if } r_1, r_2 \in \mathbb{R} \\ 0 & \text{elsewhere} \end{cases}. \quad (50)$$

*Consider the following GW-variant[3] problem:*

$$\widehat{GW}^{L_\lambda}(\hat{\mathbb{X}}, \hat{\mathbb{Y}}) = \inf_{\hat{\gamma} \in \Gamma(\hat{\mu}, \hat{\nu})} \hat{\gamma}^{\otimes 2}(L_\lambda(d_{\hat{X}}^q, d_{\hat{Y}}^q)) \quad (51)$$

*Then, when considering the bijection $\gamma \mapsto \hat{\gamma}$ defined in (6), we have that $\gamma$ is optimal for PGW problem (10) if and only if $\hat{\gamma}$ is optimal for the GW-variant problem (51).*

**Remark G.2.** *Intuitively, the above proposition states that, by introducing auxiliary points $\hat{\infty}$, we can build an equivalent relation between PGW and GW problem. This idea is firstly discussed in (Cagniart et al., 2010) in a classical optimal partial transport setting. In this paper, we extend the technique to the partial GW setting. However, this technique cannot be extended to the MPGW setting; we refer to Chapel et al. (2020, Appendix A.3) for details.*

*Proof.* The mapping $F$ defined by (6) well-defined bijection, as shown in (Bai et al., 2023; Caffarelli & McCann, 2010).

Given $\gamma \in \Gamma_{\leq}(\mu, \nu)$, we have $\hat{\gamma} = F(\gamma) \in \Gamma(\hat{\mu}, \hat{\nu})$. Let $\hat{C}(\hat{\gamma}; \mu, \nu)$ denote the transportation cost in the GW-variant problem (51), that is,

$$\hat{C}(\hat{\gamma}; \mu, \nu) := \int_{(\hat{X} \times \hat{Y})^2} L_\lambda(d_{\hat{X}}^q(x, x'), d_{\hat{Y}}^q(y, y')) \, d\hat{\gamma}(x, y) d\hat{\gamma}(x', y')$$

Then, we have

$C(\gamma; \lambda, \mu, \nu)$

$$= \int_{(X \times Y)^2} (L(d_X^q(x, x'), d_Y^q(y, y')) - 2\lambda) \, d\gamma^{\otimes 2} + \underbrace{\lambda(|\mu| + |\nu|)}_{\text{does not depend on } \gamma}$$

$$= \int_{(X \times Y)^2} (L(d_X^q(x, x'), d_Y^q(y, y')) - 2\lambda) \, d\hat{\gamma}^{\otimes 2} + \lambda(|\mu| + |\nu|) \quad (\text{since } \hat{\gamma}|_{X \times Y} = \gamma)$$

$$= \int_{(X \times Y)^2} (L(d_{\hat{X}}^q(x, x'), d_{\hat{Y}}^q(y, y')) - 2\lambda) \, d\hat{\gamma}^{\otimes 2} + \lambda(|\mu| + |\nu|) \quad (\text{as } d_{\hat{X}}|_{X \times X} = d_X, d_{\hat{Y}}|_{Y \times Y} = d_Y)$$

$$= \int_{(X \times Y)^2} L_\lambda(d_{\hat{X}}^q(x, x'), d_{\hat{Y}}^q(y, y')) \, d\hat{\gamma}^{\otimes 2} + \lambda(|\mu| + |\nu|) \quad (\text{since } \hat{L}|_{\mathbb{R} \times \mathbb{R}}(\cdot, \cdot) = (L(\cdot, \cdot) - 2\lambda))$$

$$= \int_{(\hat{X} \times \hat{Y})^2} L_\lambda(d_{\hat{X}}^q(x, x'), d_{\hat{Y}}^q(y, y')) \, d\hat{\gamma}^{\otimes 2} + \underbrace{\lambda(|\mu| + |\nu|)}_{\text{does not depend on } \hat{\gamma}} . \quad (\text{since } \hat{L} \text{ assigns 0 to } \hat{\infty})$$

---

[3] $\widehat{GW}^{L_\lambda}(\hat{\mathbb{X}}, \hat{\mathbb{Y}})$ is not a rigorous GW problem since $d_{\hat{X}} = \infty$ is possible, thus it is not a metric. Also, $\mathbb{X}$, $\mathbb{Y}$ are not necessarily *probability* mm-spaces

Combining this with the fact that $F : \gamma \mapsto \hat{\gamma}$ is a bijection, we have that $\gamma$ is optimal for (10) if and only if $\hat{\gamma}$ is optimal for (51). Under the assumptions of Proposition 3.3, there exists an optimal $\gamma \in \Gamma_{\leq}(\mu, \nu)$ for the PGW problem exists, and so we have:

$$\arg \min_{\hat{\gamma} \in \Gamma(\hat{\mu}, \hat{\nu})} \hat{C}(\hat{\gamma}; \mu, \nu) = \arg \min_{\gamma \in \Gamma_{\leq}(\mu, \nu)} C(\gamma; \lambda, \mu, \nu). \tag{52}$$

$\square$

**Remark G.3.** *Both algorithms (Algorithm 1 and 2) are mathematically and computationally equivalent, owing to the equivalence between the POT problem in Solver 1 and the OT problem in Solver 2.*

### G.1 FRANK-WOLFE FOR THE PGW PROBLEM – SOLVER 2

Similarly to the discrete PGW problem (15), consider the discrete version of (4):

$$\hat{p} = [p; |q|] \in \mathbb{R}^{n+1}, \quad \hat{q} = [q; |p|] \in \mathbb{R}^{m+1}, \tag{53}$$

and, in a similar fashion, we define $\hat{M} \in \mathbb{R}^{(n+1) \times (m+1) \times (n+1) \times (m+1)}$ as

$$\hat{M}_{i,j,i',j'} = \begin{cases} \tilde{M}_{i,j,i',j'} & \text{if } i, i' \in [1:n], j, j' \in [1:m], \\ 0 & \text{elsewhere.} \end{cases} \tag{54}$$

Then, the GW-variant problem (51) can be written as

$$\widehat{GW}(\hat{\mathbb{X}}, \hat{\mathbb{Y}}) = \min_{\hat{\gamma} \in \Gamma(\hat{p}, \hat{q})} \mathcal{L}_{\hat{M}}(\hat{\gamma}). \tag{55}$$

Based on Proposition G.1 (which relates $PGW_\lambda^L(\cdot, \cdot)$ with $\widehat{GW}(\cdot, \cdot)$), we propose two versions of the Frank-Wolfe algorithm (Frank et al., 1956) that can solve the PGW problem (15). Apart from Algorithm 1 in (Chapel et al., 2020), which solves a different formulation of partial GW, and Algorithm 1 in (Séjourné et al., 2021), which applies the Sinkhorn algorithm to solve an entropic regularized version of (8), to the best of our knowledge, a precise computational method for the discrete PGW problem (15) has not been studied.

Here, we discuss another version of the FW Algorithm for solving the PGW problem (15). The main idea relies on solving first the GW-variant problem (51), and, at the end of the iterations, by using Proposition G.1, convert the solution of the GW-variant problem to a solution for the original partial GW problem (15).

First, construct $\hat{p}, \hat{q}, \hat{M}$ as described in Proposition G.1. Then, for each iteration $k$, perform the following three steps.

**Step 1: Computation of gradient and optimal direction**. Solve the OT problem:

$$\hat{\gamma}^{(k)'} \leftarrow \arg \min_{\hat{\gamma} \in \Gamma(\hat{p}, \hat{q})} \langle \mathcal{L}_{\hat{M}}(\hat{\gamma}^{(k)}), \hat{\gamma} \rangle_F.$$

The gradient $\mathcal{L}_{\hat{M}}(\gamma^{(k)})$ can be computed in a similar way as described in Lemma H.2. We refer to Section H for details.

**Step 2: Line search method**. Find optimal step size $\alpha^{(k)}$:

$$\alpha^{(k)} = \arg \min_{\alpha \in [0,1]} \{ \mathcal{L}_{\hat{M}}((1-\alpha)\hat{\gamma}^{(k)} + \alpha \hat{\gamma}^{(k)'}) \}.$$

Similar to Solver 1, let

$$\begin{cases} \delta \hat{\gamma}^{(k)} = \hat{\gamma}^{(k)'} - \hat{\gamma}^{(k)}, \\ a = \langle \hat{M} \circ \delta \hat{\gamma}^{(k)}, \delta \hat{\gamma}^{(k)} \rangle_F, \\ b = 2 \langle \hat{M} \circ \delta \hat{\gamma}^{(k)}, \hat{\gamma}^{(k)} \rangle_F. \end{cases} \tag{56}$$

Then the optimal $\alpha^{(k)}$ is given by formula (18). See Appendix J for a detailed discussion.

**Step 3**. Update $\hat{\gamma}^{(k+1)} \leftarrow (1 - \alpha^{(k)}) \hat{\gamma}^{(k)} + \alpha^{(k)} \hat{\gamma}^{(k)'}$.

---

**Algorithm 2:** Frank-Wolfe Algorithm for partial GW, ver 2

---

**Input:** $\mu = \sum_{i=1}^{n} p_i^X \delta_{x_i}, \nu = \sum_{j=1}^{m} q_j^Y \delta_{y_j}, \gamma^{(1)}$
**Output:** $\gamma^{(final)}$
Compute $C^X, C^Y, \hat{p}, \hat{q}, \hat{\gamma}^{(1)}$
**for** $k = 1, 2, \ldots$ **do**
   $\hat{G}^{(k)} \leftarrow 2\hat{M} \circ \hat{\gamma}^{(k)}$ // Compute gradient
   $\hat{\gamma}^{(k)'} \leftarrow \arg\min_{\hat{\gamma} \in \Gamma(\hat{p},\hat{q})} \langle \hat{G}^{(k)}, \hat{\gamma} \rangle_F$ // Solve the OT problem
   Compute $\alpha^{(k)} \in [0,1]$ via (56), (18) // Line Search
   $\hat{\gamma}^{(k+1)} \leftarrow (1 - \alpha^{(k)})\hat{\gamma}^{(k)'} + \alpha\hat{\gamma}^{(k)}$// Update $\hat{\gamma}$
   if convergence, break
**end for**
$\gamma^{(final)} \leftarrow \hat{\gamma}^{(k)}[1:n, 1:m]$

---

## H GRADIENT COMPUTATION IN ALGORITHMS 1 AND 2

In this section, we discuss the computation of Gradient $\nabla\mathcal{L}_{\tilde{M}}(\gamma)$ in Algorithm 1 and $\nabla\mathcal{L}_{\hat{M}}(\hat{\gamma})$ in Algorithm 2.

**Proposition H.1** (Proposition 1 (Peyré et al., 2016)). *If the cost function can be written as*

$$L(r_1, r_2) = f_1(r_1) + f_2(r_2) - h_1(r_1)h_2(r_2) \tag{57}$$

*then*

$$M \circ \gamma = u(C^X, C^Y, \gamma) - h_1(C^X)\gamma h_2(C^Y)^\top, \tag{58}$$

*where* $u(C^X, C^Y, \gamma) := f_1(C^X)\gamma_1 1_m^\top + 1_n \gamma_2^\top f_2(C^Y)$.

Additionally, the following lemma builds the connection between $\tilde{M} \circ \gamma$ and $M \circ \gamma$.

**Lemma H.2.** *For any* $\gamma \in \mathbb{R}^{n \times m}$*, we have:*

$$\tilde{M} \circ \gamma = M \circ \gamma - 2\lambda|\gamma|1_{n,m}. \tag{59}$$

*Proof.* For any $\gamma \in \mathbb{R}^{n \times m}$, we have

$$\begin{aligned}
\tilde{M} \circ \gamma &= (M1_{n,n,m,m} - 2\lambda) \circ \gamma \\
&= (M - 2\lambda 1_{n,n,m,m}) \circ \gamma \\
&= M \circ \gamma - 2\lambda 1_{n,m,n,m} \circ \gamma \\
&= M \circ \gamma - 2(\langle 1_{n,m}, \gamma \rangle_F)1_{n,m} \\
&= M \circ \gamma - 2\lambda|\gamma|1_{n,m}
\end{aligned}$$

where the second equality follows from Lemma F.1. $\qquad\square$

Next, in the setting of Algorithm 2, for any $\hat{\gamma} \in \mathbb{R}^{(n+1) \times (m+1)}$, we have

$$\nabla\mathcal{L}_{\hat{M}}(\hat{\gamma}) = 2\hat{M} \circ \hat{\gamma} \tag{60}$$

and $\hat{M} \circ \hat{\gamma}$ can be computed by the following lemma.

**Lemma H.3.** *For each* $\hat{\gamma} \in \mathbb{R}^{(n+1) \times (m+1)}$*, we have* $\hat{M} \circ \hat{\gamma} \in \mathbb{R}^{(n+1) \times (m+1)}$ *with the following:*

$$(\hat{M} \circ \hat{\gamma})_{ij} = \begin{cases} (\tilde{M} \circ \hat{\gamma}[1:n, 1:m])_{ij} & \text{if } i \in [1:n], j \in [1:m] \\ 0 & \text{elsewhere} \end{cases}. \tag{61}$$

*Proof.* Recall the definition of $\hat{M}$ is given by (54), choose $i \in [1:n], j \in [1:m]$, we have

$$
\begin{aligned}
(\hat{M} \circ \hat{\gamma})_{ij} &= \sum_{i'=1}^{n} \sum_{j'=1}^{m} \hat{M}_{i,j,i',j'} \hat{\gamma}_{i',j'} + \sum_{j'=1}^{m} \hat{M}_{i,j,n+1,j} \hat{\gamma}_{n+1,j'} + \sum_{i'=1}^{n} \hat{M}_{i,j,i',m+1} \hat{\gamma}_{i,m+1} \\
&\quad + \hat{M}_{i,j,n+1,m+1} \hat{\gamma}_{n+1,m+1} \\
&= \sum_{i'=1}^{n} \sum_{j'=1}^{m} \hat{M}_{i,j,i',j'} \hat{\gamma}_{i',j'} + 0 + 0 + 0 = \sum_{i'=1}^{n} \sum_{j'=1}^{m} \tilde{M}_{i,j,i',j'} \hat{\gamma}_{i',j'} \\
&= (\tilde{M} \circ (\hat{\gamma}[1:n,1:m]))_{ij}
\end{aligned}
$$

If $i = n+1$, we have

$$
(\hat{M} \circ \hat{\gamma})_{n+1,j} = \sum_{i'=1}^{n+1} \sum_{j'=1}^{m+1} \hat{M}_{n+1,j,i',j'} \hat{\gamma}_{i',j'} = 0
$$

Similarly, $(\hat{M} \circ \hat{\gamma})_{i,m+1} = 0$. Thus, we complete the proof. $\qquad\square$

## I    LINE SEARCH IN ALGORITHM 1

In this section, we discuss the derivation of the line search algorithm.

We observe that in the partial GW setting, for each $\gamma \in \Gamma_{\leq}(\mu, \nu)$, the marginals of $\gamma$ are not fixed. Thus, we can not directly apply the classical algorithm (e.g., (Titouan et al., 2019a)).

In iteration $k$, let $\gamma^{(k)}, \gamma^{(k)'}$ be the previous and new transportation plans from step 1 of the algorithm. For convenience, we denote them as $\gamma$ and $\gamma'$, respectively.

The goal is to solve the following problem:

$$
\min_{\alpha \in [0,1]} \mathcal{L}(\tilde{M}, (1-\alpha)\gamma + \alpha\gamma') \tag{62}
$$

where $\mathcal{L}(\tilde{M}, \gamma) = \langle \tilde{M} \circ \gamma, \gamma \rangle_F$. By denoting $\delta\gamma = \gamma' - \gamma$, we have

$$
\mathcal{L}(\tilde{M}, (1-\alpha)\gamma + \alpha\gamma') = \mathcal{L}(\tilde{M}, \gamma + \alpha\delta\gamma).
$$

Then,

$$
\begin{aligned}
&\langle \tilde{M} \circ (\gamma + \alpha\delta\gamma), (\gamma + \alpha\delta\gamma) \rangle_F \\
&= \langle \tilde{M} \circ \gamma, \gamma \rangle_F + \alpha \left( \langle \tilde{M} \circ \gamma, \delta\gamma \rangle_F + \langle \tilde{M} \circ \delta\gamma, \gamma \rangle_F \right) + \alpha^2 \langle \tilde{M} \circ \delta\gamma, \delta\gamma \rangle_F
\end{aligned}
$$

Let

$$
\begin{aligned}
a &= \langle \tilde{M} \circ \delta\gamma, \delta\gamma \rangle_F, \\
b &= \langle \tilde{M} \circ \gamma, \delta\gamma \rangle_F + \langle \tilde{M} \circ \delta\gamma, \gamma \rangle_F = 2\langle \tilde{M} \circ \gamma, \delta\gamma \rangle_F, \\
c &= \langle \tilde{M} \circ \gamma, \gamma \rangle_F,
\end{aligned} \tag{63}
$$

where the second identity in (63) follows from Lemma F.1 and the fact that $\tilde{M} = M1_{n,n,m,m} - 2\lambda 1_{n,m,n,m}$ is symmetric.

Therefore, the above problem (62) becomes

$$
\min_{\alpha \in [0,1]} a\alpha^2 + b\alpha + c.
$$

The solution is the following:

$$
\alpha^* = \begin{cases} 1 & \text{if } a \leq 0, a+b \leq 0, \\ 0 & \text{if } a \leq 0, a+b > 0, \\ \text{clip}(\frac{-b}{2a}, [0,1]) & \text{if } a > 0, \end{cases} \tag{64}
$$

where

$$\text{clip}(\frac{-b}{2a}, [0,1]) = \min\left\{1, \max\{0, \frac{-b}{2a}\}\right\} = \begin{cases} \frac{-b}{2a} & \text{if } \frac{-b}{2a} \in [0,1], \\ 0 & \text{if } \frac{-b}{2a} < 0, \\ 1 & \text{if } \frac{-b}{2a} > 1. \end{cases}$$

We can further discuss the difference in computation of $a$ and $b$ in the PGW setting and the classical GW setting. If the assumption in Proposition H.1 holds, by (58) and (59), we have

$$
\begin{aligned}
a &= \langle \tilde{M} \circ \delta\gamma, \delta\gamma \rangle_F \\
&= \langle (M \circ \delta\gamma - 2\lambda|\delta\gamma|I_{n,m}), \delta\gamma \rangle_F \\
&= \langle M \circ \delta\gamma, \delta\gamma \rangle_F - 2\lambda|\delta\gamma|^2 \\
&= \left\langle u(C^X, C^Y, \delta\gamma) - h_1(C^X)\delta\gamma h_2(C^Y)^\top, \delta\gamma \right\rangle_F - 2\lambda|\delta\gamma|^2,
\end{aligned}
\tag{65}
$$

$$
\begin{aligned}
b &= 2\langle \tilde{M} \circ \gamma, \delta\gamma \rangle_F \\
&= 2\langle M \circ \gamma - 2\lambda|\gamma|I_{n,m}, \delta\gamma \rangle \\
&= 2(\langle M \circ \gamma, \delta\gamma \rangle_F - 2\lambda|\delta\gamma||\gamma|)
\end{aligned}
\tag{66}
$$

Note that in the classical GW setting (Titouan et al., 2019a), the term $u(C^X, C^Y, \delta\gamma) = 0_{n \times m}$ and $|\delta\gamma| = 0$. Therefore, in such line search algorithm (Algorithm 2 in (Titouan et al., 2019a)), the terms $u(C^X, C^Y, \delta\gamma), 2\lambda|\delta\gamma|1_{n \times m}$ are not required. In addition, in equation (66), $M \circ \gamma, 2\lambda|\gamma|$ have been computed in the gradient computation step. Thus, these two terms can be directly applied in this step.

## J  LINE SEARCH IN ALGORITHM 2

Similar to the previous section, in iteration $k$, let $\hat{\gamma}^{(k)}, \hat{\gamma}^{(k)'}$ denote the previous transportation plan and the updated transportation plan. For convenience, we denote them as $\hat{\gamma}, \hat{\gamma}'$, respectively.

Let $\delta\hat{\gamma} = \hat{\gamma} - \hat{\gamma}'$.

The goal is to find the following optimal $\alpha$:

$$\alpha = \arg\min_{\alpha \in [0,1]} \mathcal{L}(\hat{M}, (1-\alpha)\hat{\gamma}, \alpha\hat{\gamma}') = \arg\min_{\alpha \in [0,1]} \mathcal{L}(\hat{M}, \alpha\delta\hat{\gamma} + \hat{\gamma}), \tag{67}$$

where $\hat{M} \in \mathbb{R}^{(n+1) \times (m+1) \times (n+1) \times (m+1)}$, with $\hat{M}[1:n, 1:m, 1:n, 1:m] = \tilde{M} = M - 2\lambda 1_{n \times m \times n \times m}$.

Similar to the previous section, let

$$
\begin{aligned}
a &= \langle \hat{M} \circ \delta\hat{\gamma}, \delta\hat{\gamma} \rangle_F, \\
b &= \langle \hat{M} \circ \delta\hat{\gamma}, \hat{\gamma} \rangle_F + \langle \hat{M} \circ \hat{\gamma}, \delta\hat{\gamma} \rangle_F = 2\langle \hat{M} \circ \delta\hat{\gamma}, \hat{\gamma} \rangle_F, \\
c &= \langle \hat{M} \circ \hat{\gamma}, \hat{\gamma} \rangle_F,
\end{aligned}
\tag{68}
$$

where (68) holds since $\hat{M}$ is symmetric. Then, the optimal $\alpha$ is given by (64).

It remains to discuss the computation. By Lemma F.1, we set $\gamma = \hat{\gamma}[1:n, 1:m], \delta\gamma = \delta\hat{\gamma}[1:n, 1:m]$. Then,

$$
\begin{aligned}
a &= \langle (\hat{M} \circ \delta\hat{\gamma})[1:n, 1:m], \delta\gamma \rangle_F = \langle (\tilde{M} \circ \delta\gamma, \delta\gamma \rangle_F, \\
b &= \langle (\hat{M} \circ \delta\hat{\gamma})[1:n, 1:m], \gamma \rangle_F = \langle (\tilde{M} \circ \delta\gamma, \gamma \rangle_F.
\end{aligned}
$$

Thus, we can apply (65), (66) to compute $a, b$ in this setting by plugging in $\gamma = \hat{\gamma}[1:n, 1:m]$ and $\delta\gamma = \delta\hat{\gamma}[1:n, 1:m]$.

## K  Convergence

As in (Chapel et al., 2020), we will use the results from (Lacoste-Julien, 2016) on the convergence of the Frank-Wolfe algorithm for non-convex objective functions.

Consider the minimization problems

$$\min_{\gamma \in \Gamma_{\leq}(\mathrm{p},\mathrm{q})} \mathcal{L}_{\tilde{M}}(\gamma) \qquad \text{and} \qquad \min_{\hat{\gamma} \in \Gamma(\hat{\mathrm{p}},\hat{\mathrm{q}})} \mathcal{L}_{\hat{M}}(\hat{\gamma}) \tag{69}$$

that corresponds to the discrete partial GW problem and the discrete GW-variant problem (used in version 2), respectively.

Consider the objective functions $\gamma \mapsto \mathcal{L}_{\tilde{M}}(\gamma) = \tilde{M}\,\gamma^{\otimes 2}$ and $\hat{\gamma} \mapsto \mathcal{L}_{\hat{M}}(\hat{\gamma}) = \hat{M}\,\hat{\gamma}^{\otimes 2}$, where $\tilde{M} = M - 2\lambda\,1_{n,m}$ for a fixed matrix $M \in \mathbb{R}^{n \times m}$ with $\lambda > 0$, and $\hat{M}$ is given by (54). Although $\tilde{M}$ and $\hat{M}$ are symmetric for $\lambda > 0$, they are not positive semi-definite, so these objective functions are generally non-convex.

However, the constraint sets $\Gamma_{\leq}(\mathrm{p},\mathrm{q})$ and $\Gamma(\hat{\mathrm{p}},\hat{\mathrm{q}})$ are convex and compact in $\mathbb{R}^{n \times m}$ (see Proposition B.2 in (Liu et al., 2023)) and in $\mathbb{R}^{(n+1) \times (m+1)}$, respectively.

From now on, we will concentrate on the first minimization problem in (69), and the convergence analysis for the second one will be analogous.

Consider the *Frank-Wolfe gap* of $\mathcal{L}_{\tilde{M}}$ at the approximation $\gamma^{(k)}$ of the optimal plan $\gamma$:

$$g_k = \min_{\gamma \in \Gamma_{\leq}(\mathrm{p},\mathrm{q})} \langle \nabla \mathcal{L}_{\tilde{M}}(\gamma^{(k)}), \gamma^{(k)} - \gamma \rangle_F. \tag{70}$$

It provided a good criterion to measure the distance to a stationary point at iteration $k$. Indeed, a plan $\gamma^{(k)}$ is a stationary transportation plan for the corresponding constrained optimization problem in (69) if and only if $g_k = 0$. Moreover, $g_k$ is always non-negative ($g_k \geq 0$).

From Theorem 1 in (Lacoste-Julien, 2016), after $K$ iterations, we have the following upper bound for the minimal Frank-Wolf gap:

$$g_K := \min_{1 \leq k \leq K} g_k \leq \frac{\max\{2L_1, \mathrm{Lip} \cdot (\mathrm{diam}(\Gamma_{\leq}(\mathrm{p},\mathrm{q})))^2\}}{\sqrt{K}}, \tag{71}$$

where

$$L_1 := \mathcal{L}_{\tilde{M}}(\gamma^{(1)}) - \min_{\gamma \in \Gamma_{\leq}(\mathrm{p},\mathrm{q})} \mathcal{L}_{\tilde{M}}(\gamma)$$

is the initial global sub-optimal bound for the initialization $\gamma^{(1)}$ of the algorithm; Lip is the Lipschitz constant of function $\gamma \mapsto \nabla \mathcal{L}_{\tilde{M}}$; and

$$\mathrm{diam}(\Gamma_{\leq}(\mathrm{p},\mathrm{q})) = \sup_{\gamma,\gamma' \in \Gamma_{\leq}(\mu,\nu)} \|\gamma - \gamma'\|_F$$

is the $\|\cdot\|_F$ diameter of $\Gamma_{\leq}(\mathrm{p},\mathrm{q})$ in $\mathbb{R}^{n \times m}$.

The important thing to notice is that the constant $\max\{2L_1, D_L\}$ does not depend on the iteration step $k$. Thus, according to Theorem 1 in (Lacoste-Julien, 2016), the rate in $\tilde{g}_K$ is $\mathcal{O}(1/\sqrt{K})$. That is, the algorithm takes at most $\mathcal{O}(1/\varepsilon^2)$ iterations to find an approximate stationary point with a gap smaller than $\varepsilon$.

**Lemma K.1.** *In the discrete PGW problem, we have*

$$diam(\Gamma_{\leq}(p,q)) \leq 2s := 2\min(|p|,|q|) \tag{72}$$

*Proof.* Choose $\gamma, \gamma' \in \Gamma_{\leq}(p,q)$, we apply the property

$$(a-b)^2 \leq 2a^2 + 2b^2, \forall a, b \in \mathbb{R} \tag{73}$$

and obtain

$$
\begin{aligned}
\|\gamma - \gamma'\|_F^2 &= \sum_{i,j}^{n,m} |\gamma_{i,j} - \gamma'_{i,j}|^2 \\
&\leq \sum_{i,j}^{n,m} 2|\gamma_{i,j}|^2 + 2|\gamma'_{i,j}|^2 \\
&\leq 2 \left[ (\sum_{i,j} \gamma_{i,j})^2 + (\sum_{i,j} \gamma'_{i,j})^2 \right] \\
&= 2(|\gamma|^2 + |\gamma'|^2) \\
&\leq 2\min(|p|,|q|)^2 + 2\min(|p|,|q|)^2 \\
&= 4\min(|p|,|q|)^2
\end{aligned}
\tag{74}
$$

and thus, we complete the proof. □

**Lemma K.2.** *For the Lip term in ([71]) can be bounded as following:*

$$
Lip \leq nm \max_{i,i,j,j'} (2(C_{i,i'}^X)^2 + 2(C_{j,j'}^Y)^2, 2\lambda)
\tag{75}
$$

*Proof.* Pick $\gamma, \gamma' \in \Gamma_\leq(p,q)$ we have,

$$
\begin{aligned}
&\|\nabla \mathcal{L}_{\tilde{M}}(\gamma) - \nabla \mathcal{L}_{\tilde{M}}(\gamma')\|_F^2 \\
&= \|\tilde{M} \circ \gamma - \tilde{M} \circ \gamma'\|_F^2 \\
&= \|[M - 2\lambda] \circ (\gamma - \gamma')\|_F^2 \\
&= \sum_{i,j} \left( [(M - 2\lambda) \circ (\gamma - \gamma')]_{i,j} \right)^2 \\
&= \sum_{i,j} \left( \sum_{i',j'} (M_{i,j,i',j'} - 2\lambda)(\gamma_{i',j'} - \gamma'_{i',j'}) \right)^2 \\
&\leq \sum_{i,j} \left( \sum_{i',j'} |M_{i,j,i',j'} - 2\lambda| |\gamma_{i',j'} - \gamma'_{i',j'}| \right)^2 \\
&\leq \underbrace{\left( \max_{i,j,i',j'} \{M_{i,j,i',j'} - 2\lambda\} \right)^2}_{A} \cdot \underbrace{\sum_{i,j}^{n,m} \left( \sum_{i',j'}^{n,m} |\gamma_{i',j'} - \gamma'_{i',j'}| \right)^2}_{B}
\end{aligned}
$$

For the first term, we have:

$$
A \leq \max\{2(C^X)^2 + 2(C^Y)^2, 2\lambda\}^2
$$

where

$$
\max\{2(C^X)^2 + 2(C^Y)^2, 2\lambda\} := \max\{\max_{i,i',j,j'} 2(C_{i,i'}^X)^2 + 2(C_{j,j'}^Y)^2, 2\lambda\}
$$

For the second term, we have:

$$
\begin{aligned}
&\sum_{i,j}^{n,m} \left( \sum_{i',j'}^{n,m} |\gamma_{i',j'} - \gamma'_{i',j'}| \right)^2 \\
&\leq \sum_{i,j}^{n,m} \left( nm \sum_{i',j'}^{n,m} |\gamma_{i',j'} - \gamma'_{i',j'}|^2 \right) \\
&\leq n^2 m^2 \|\gamma - \gamma'\|_F^2
\end{aligned}
$$

Thus we obtain

$$\text{Lip} \leq \frac{\max(2(C^X)^2 + 2(C^Y)^2, 2\lambda)nm\|\gamma - \gamma'\|_F}{\|\gamma - \gamma'\|_F} = nm\max(2(C^X)^2 + 2(C^Y)^2, 2\lambda)$$

and we complete the proof. $\qquad\square$

Combined the above two lemmas, we derive the convergence rate of the Frank-Wolf gap (70):

**Proposition K.3.** *When $L(r_1, r_2) = |r_1 - r_2|^2$ in the PGW problem, the Frank-Wolfe gap of algorithm 1, defined in (70) at iteration $k$ satisfies the following:*

$$g_k \leq \frac{\max\left\{2L_1, 4\min(|\mathrm{p}|, |\mathrm{q}|)^2 \cdot nm(\max\{2(C^X)^2 + 2(C^Y)^2, 2\lambda\})\right\}}{\sqrt{k}} \tag{76}$$

*Proof.* The proof directly follows from the upper bounds (72),(75) and the inequality (71). $\qquad\square$

**Remark K.4.** *Note, if the cost function in PGW is defined by $|r_1 - r_2|^p$ for some $p \neq 2$, it is straightforward to verify that the upper bound of $g_k$ is obtained by replacing the term $\max((C^X)^2 + (C^Y)^2, 2\lambda)$ should be replaced by*

$$\max_{i,j,i',j'} \left[ 2^{p-1}((C^X)^p + (C^Y)^p), 2\lambda \right].$$

**Remark K.5.** *It is straightforward to verify that the Frank-Wolf gap for algorithm 2 is also upper bounded by (76).*

**Remark K.6.** *From the proposition (K.3), to achieve an $\epsilon-$accurate solution, the required number of iterations is*

$$\frac{\max\left\{2L_1, 2\min(|\mathrm{p}|, |\mathrm{q}|) \cdot n^2 m^2 \max(\{2(C^X)^2 + 2(C^Y)^2, 2\lambda\})\right\}^2}{\epsilon^2}$$

*In practice, by lemma E.2, when $\lambda$ is large, it is equivalent to set $\lambda = \max((C^X)^2 - (C^Y)^2)$ and thus, the value $\lambda$ will not affect the number of iterations.*

**Remark K.7.** *In this remark, we compare the upper (40) and the upper bounds in previous works, including (Chapel et al., 2020) and cang2024supervised. We observe that:*

*If we ignore the term $\lambda$ (or equivalently, set $\lambda = \max_{i,i',j,j'}(2(C^X)^2 + 2(C^Y)^2)$), the upper bound (76) aligns with the upper bound for the Frank-Wolfe Gap in Mass-constraint PGW (see Chapel et al. (2020, Eq. (10))) and balanced GW (see Cang et al. (2024, p. 18).*

*However, there exist several differences in details between our result and the result in (Chapel et al., 2020).*

- *First, in (Chapel et al., 2020), the term $nm$ is omitted. We suspect this omission is a typo, as the $nm$ term is contained in (Cang et al., 2024).*

- *In (Chapel et al., 2020), "s" plays the role of $\min(|\mathrm{p}|, |\mathrm{q}|)$ in our result. The assumption in Chapel et al. (2020, Lemma 1) is*

$$|\gamma| = |\gamma'| = 1$$

  *If we normalize $\mathrm{p}, \mathrm{q}$ such that $\max(|\mathrm{p}|, |\mathrm{q}|) \leq 1$, we will obtain*

$$2\min(|\mathrm{p}|, |\mathrm{q}|)^2 \leq 2\min(|\mathrm{p}|, |\mathrm{q}|),$$

  *and our result will induce the upper bound in Chapel et al. (2020, Lemma 1).*

## L Related Work: Mass-Constrained Partial Gromov-Wasserstein

Partial Gromov-Wasserstein is first introduced in (Chapel et al., 2020). To distinguish the PGW problem in (Chapel et al., 2020) and the PGW problem in this paper, we call the former one the Mass-Constrained Gromov-Wasserstein problem (MPGW):

$$MPGW_\rho(\mathbb{X}, \mathbb{Y}) := \inf_{\gamma \in \Gamma^\rho_\le(\mu,\nu)} \gamma^{\otimes 2}(L(d_X^q, d_Y^q)), \tag{77}$$

where $\rho \in [0, \min\{|\mu|, |\nu|\}]$, and

$$\Gamma^\rho_\le(\mu, \nu) := \{\gamma \in \mathcal{M}_+(X \times Y) : \gamma_1 \le \mu, \ \gamma_2 \le \nu, \ |\gamma| = \rho\}. \tag{78}$$

Unlike the relationship between Partial OT and OT, it is not rigorous to claim that PGW and MPGW are equivalent. The objective function

$$\gamma \mapsto \int_{(X \times Y)^2} L\big(d_X^2(x, x'), d_Y^2(y, y')\big) d\gamma^{\otimes 2} \tag{79}$$

is non-convex, even if the map $(r_1, r_2) \mapsto L(r_1, r_2)$ itself is convex (Peyré et al., 2019). If the problem were convex, then MPGW, viewed as the 'Lagrangian formulation' of PGW (adding the partial constraint into the functional via Lagrange multipliers), would coincide with PGW. However, since these problems are not convex, we cannot assert their equivalence in principle.

We can still investigate their relationship by the following lemma, based on which we design the wall-clock time experiment in Section Q.

**Proposition L.1.** *For each $\lambda \ge 0$, there exists $\rho \in [0, \min(|\mu|, |\nu|)]$ such that, for each $\gamma \in \Gamma_\le(\mu, \nu)$ with $|\gamma| = \rho$, $\gamma$ is optimal in $PGW_\lambda(\mathbb{X}, \mathbb{Y})$ iff $\gamma$ is optimal in $MPGW_\rho(\mathbb{X}, \mathbb{Y})$. Furthermore,*

$$PGW_\lambda(\mu, \nu) = MPGW_\rho(\mu, \nu) + \lambda(|\mu|^2 + |\nu|^2 - 2\rho^2).$$

*Proof.* Pick $\gamma' \in \Gamma^\rho_\le(\mu, \nu) \subset \Gamma_\le(\mu, \nu)$, since $\gamma$ is optimal in $PGW_\lambda(\mu, \nu)$, we have

$$0 \le C(\gamma; \lambda, \mu, \nu) - C(\gamma'; \lambda, \mu, \nu)$$
$$= \int_{(X \times Y)^2} L(d_X^2(x, x'), d_Y^2(y, y')) d(\gamma^{\otimes 2} - \gamma'^{\otimes 2})$$

Thus, $\gamma$ is optimal in $\Gamma^\rho_\le(\mu, \nu)$ for $MPGW_\rho(\mathbb{X}, \mathbb{Y})$ and we complete the proof. $\square$

**Remark L.2.** *The above proposition clarifies a connection between $PGW$ and $MPGW$. Specifically, it implies that an optimal transportation plan for $PGW$ can be derived from $MPGW$ by appropriately setting $\rho$. **However, it is important to note that $MPGW$ and $PGW$ admit distinct transportation costs. $MPGW$ does not define a metric, whereas $PGW$ does.** Therefore, in experiments that require transportation costs, such as the shape retrieval experiment described in the main text, the results produced by the two methods will differ significantly.*

**Example L.3.** *Consider the following three mm-spaces*

$$\mathbb{X}_1 = \left(\mathbb{R}^3, \|\cdot\|, \sum_{i=1}^{1000} \alpha\delta_{x_i}\right), \quad \mathbb{X}_2 = \left(\mathbb{R}^3, \|\cdot\|, \sum_{i=1}^{800} \alpha\delta_{x_i}\right), \quad \mathbb{X}_3 = \left(\mathbb{R}^3, \|\cdot\|, \sum_{i=1}^{400} \alpha\delta_{x_i}\right),$$

*where $\alpha > 0$ is the mass of each point. For numerical stability reasons, we set $\alpha = 1/1000$. On the one hand, if we compute MPGW, the mass is fixed to be a value $\rho \in [0, 0.4]$, since the total mass in $\mathbb{X}_3$ is 0.4. For our experiment, we set $\rho = 0.4$, and we observe:*

$$MPGW_\rho(\mathbb{X}_1, \mathbb{X}_2; \rho = 0.4) = MPGW_\rho(\mathbb{X}_2, \mathbb{X}_3; \rho = 0.4) = MPGW_\rho(\mathbb{X}_1, \mathbb{X}_3; \rho = 0.4) = 0$$

*On the other hand, if we compute our PGW, considering any $\lambda > 0$ (in particular, we set $\lambda = 10$), we obtain*

$$PGW_\lambda(\mathbb{X}_1, \mathbb{X}_2; \lambda = 10) = 3.6$$
$$PGW_\lambda(\mathbb{X}_2, \mathbb{X}_3; \lambda = 10) = 4.8$$
$$PGW_\lambda(\mathbb{X}_1, \mathbb{X}_3; \lambda = 10) = 8.4$$

*In particular, one can verify the triangular inequality.*

*As a conclusion, in this example, MPGW can not describe the dissimilarity of any two datasets taken from $\{\mathbb{X}_1, \mathbb{X}_2, \mathbb{X}_3\}$. They are three distinct datasets, but MPGW returns zero for each pair. On the contrary, our PGW can measure dissimilarity.*

*In addition, the discrepancy provided by our PGW formulation is consistent with the following intuitive observation: One expects the dissimilarity between $\mathbb{X}_1$ and $\mathbb{X}_3$ to be larger than the difference $\mathbb{X}_1$ and $\mathbb{X}_2$ and than the difference between $\mathbb{X}_1$ and $\mathbb{X}_2$. This is because we are considering discrete measures, with the same mass at each point concentrated on the sets $\{x_1, \ldots, x_{400}\} \subset \{x_1, \ldots, x_{400}, \ldots, x_{800}\} \subset \{x_1, \ldots, x_{400}, \ldots, x_{800}, \ldots, x_{1000}\}$ for the datasets $\mathbb{X}_3, \mathbb{X}_2, \mathbb{X}_1$, respectively.*

**Remark L.4.** *Moreover, there may, in fact, exist instances of the two problems for which the solution sets are not equal for any value of the hyperparameters $\lambda$ and $\rho$; we illustrate this in the following example:*

*Consider the mm-spaces given by*

$$\mathbb{X}_1 = \left(\mathbb{R}^{d_1}, \|\cdot\|, \sum_{i=1}^{n} \delta_{x_i}\right) \quad and \quad \mathbb{X}_2 = \left(\mathbb{R}^{d_2}, \|\cdot\|, \sum_{j=1}^{m} \delta_{y_j}\right).$$

*If we let $\lambda = 0$, then $PGW_0(\mathbb{X}_1, \mathbb{X}_2)$ has $\delta_{(x_i, y_j)}$ as a solution for all $(i, j)$, as well as the zero measure.*

*Now, we observe that for $\rho = 0$, $\delta_{(x_i, y_j)}$ is not a solution of $MPGW_\rho(\mathbb{X}_1, \mathbb{X}_2)$; meanwhile, for any $\rho > 0$, the zero measure is not a solution of $MPGW_\rho(\mathbb{X}_1, \mathbb{X}_2)$.*

*Hence, we see that the set of solutions between $PGW_0(\mathbb{X}_1, \mathbb{X}_2)$ and $MPGW_\rho(\mathbb{X}_1, \mathbb{X}_2)$ are distinct for all $\rho$.*

**Remark L.5.** *The other direction of the "equivalence relation" between PGW and MPGW given by Proposition L.1 may not hold. In particular, there exists a problem $MPGW_\rho(\mathbb{X}_1, \mathbb{X}_2)$ for some $\rho > 0$, such that for all $\lambda > 0$, each solution to $MPGW_\rho(\mathbb{X}_1, \mathbb{X}_2)$ is not a solution to $PGW_\lambda(\mathbb{X}_1, \mathbb{X}_2)$, and each solution to $PGW_\lambda(\mathbb{X}_1, \mathbb{X}_2)$ is not a solution to $MPGW_\rho(\mathbb{X}_1, \mathbb{X}_2)$.*

*As an example, consider $\mathbb{X}_1 = \left(\mathbb{R}^d, \|\cdot\|, \sum_{i=1}^{100} \delta_{x_i}\right)$ and $\mathbb{X}_2 = \left(\mathbb{R}^d, \|\cdot\|, \sum_{i=1}^{200} \delta_{x_i}\right)$, where $x_1, \ldots x_{200}$ are distinct. Then for each $\rho \in (0, 100]$, any solution to $MPGW_\rho(\mathbb{X}_1, \mathbb{X}_2)$ is not a solution for $PGW_\lambda(\mathbb{X}_1, \mathbb{X}_2)$ and vice versa.*

## M  PARTIAL GROMOV-WASSERSTEIN BARYCENTER

We first introduce the classical Gromov-Wasserstein problem (Peyré et al., 2016): Consider finite discrete probability measures $\mu^1, \ldots, \mu^K$, where $\mu^k = \sum_{i=1}^{n_k} p_i^k \delta_{x_i^k}$ and each $x_i^k \in \mathbb{R}^{d_k}$ for some $d_k \in \mathbb{N}$. Let $C^k = [\|x_i^k - x_{i'}^k\|^2]_{i,i' \in [1:n_k]}$ and $\mathrm{p}^k = [p_1^k, \ldots, p_{n_k}^k]^\top$. Given $\mathrm{p} \in \mathbb{R}_+^n$ with $|\mathrm{p}| = 1$ for some $n \in \mathbb{N}$ and $\xi_1, \ldots, \xi_K \geq 0$ with $\sum_{k=1}^{K} \xi_k = 1$, the GW barycenter problem is defined by:

$$\min_{C, \gamma^k} \sum_{k=1}^{K} \xi_k \langle L(C, C^k) \circ \gamma^k, \gamma^k \rangle, \tag{80}$$

where the minimization is over all matrices $C \in \mathbb{R}^{n \times n}, \gamma^k \in \Gamma(\mathrm{p}, \mathrm{p}^k), \forall k \in [1 : K]$.

Similarly, we can extend the above definition into the PGW setting. In particular, we relax the assumptions $|\mathrm{p}| = 1$ and $|\mathrm{p}^k| = 1$ for each $k \in [1:K]$. Given $\lambda_1, \ldots, \lambda_K > 0$, the PGW barycenter is the follow problem:

$$\min_{C, \gamma_k} \sum_k \xi_k \langle M(C, C^k) \circ \gamma^k, \gamma^k \rangle - 2\lambda_k |\gamma^k|^2 \tag{81}$$

where each $\gamma^k \in \Gamma_{\leq}(\mathrm{p}, \mathrm{p}^k)$.

The problem (81) can be solved iterative by two steps:

**Minimization with respect to $C$:** For each $k$, we solve the PGW problem

$$\min_{\gamma^k \in \Gamma_{\leq}(p, p^k)} \langle M(C, C^k) \circ \gamma^k, \gamma^k \rangle - 2\lambda_k |\gamma^k|^2$$

via solver 1 or 2.

**Minimization with respect to $\{\gamma^k\}_k$:**

$$\min_C \sum_k \xi_k \langle M(C, C^k) \circ \gamma^k, \gamma^k \rangle \tag{82}$$

Note, we can ignore the $-2\lambda_k |\gamma^k|^2$ terms as $\gamma^k$ is fixed in this case.

It has closed form solution due to the following lemma and proposition:

**Lemma M.1.** *Given matrices $A \in \mathbb{R}^{n,m}, B \in \mathbb{R}^{m,l}, C \in \mathbb{R}^{n,l}$, let*

$$\mathcal{L} = \langle AB, C \rangle,$$

*then $\frac{d\mathcal{L}}{dA} = CB^\top$.*

*Proof.* For any $i \in [1:n], j \in [1:m]$, we have

$$\begin{aligned}
\frac{d\mathcal{L}}{dA_{ij}} &:= \sum_{i',j'} \frac{d}{dA_{ij}} C_{i',j'} (AB)_{i',j'} \\
&= \sum_{i',j'} C_{i',j'} \frac{d(\sum_k A_{i',k} B_{k,j'})}{dA_{ij}} \\
&= \sum_{j'} C_{i,j'} B_{k,j'} = (CB^\top)_{ij}.
\end{aligned}$$

$\square$

**Proposition M.2.** *If $L$ satisfies (57), and $f_1'/h_1'$ is invertible, then (82) can be solved by*

$$C = \left( \frac{f_1'}{h_1'} \right)^{-1} \left( \frac{\sum_k \xi_k \gamma^k h_2(C^k)(\gamma_k)^\top}{\sum_k \xi_k \gamma_1^k (\gamma_1^k)^\top} \right), \tag{83}$$

*where*

$$\frac{A}{B} = \left[ \frac{A_{ij}}{B_{ij}} \right]_{ij}, \text{with convention } \frac{0}{0} = 0.$$

*Special case: if $|\mathrm{p}| \leq |\mathrm{p}^k|, \forall k$, when $\lambda$ is sufficiently large, (83) and Peyré et al. (2016, Proposition 3) coincide.*

*Proof.* From Proposition H.1, the objective in (82) becomes

$$\begin{aligned}
\mathcal{L} &= \sum_k \xi_k \langle f_1(C) \gamma_1^1 1_{n_k}^\top + 1_n (\gamma_2^k)^\top f_2(C^k) - h_1(C) \gamma^k h_2(C^k)^\top, \gamma^k \rangle \\
&= \sum_k \xi_k \langle f_1(C) \gamma_1^1 1_{n_k}^\top, \gamma^k \rangle + \underbrace{\sum_k \xi_k \langle 1_n (\gamma_2^k)^\top f_2(C^k), \gamma^k \rangle}_{\text{constant}} - \sum_k \xi_k \langle h_1(C) \gamma^k h_2(C^k)^\top, \gamma^k \rangle
\end{aligned}$$

We set $\frac{d\mathcal{L}}{dC} = 0$. From Lemma M.1, we have:

$$
\begin{aligned}
0 &= \frac{d\mathcal{L}}{dC} \\
&= \sum_k \xi_k f_1'(C) \odot \gamma^k 1_{n_k}(\gamma_1^k)^\top - \sum_k \xi_k h_1'(C) \odot \gamma^k h_2(C^k)(\gamma^k)^\top \\
&= f_1'(C) \odot \sum_k \xi_k \gamma^k 1_{n_k}(\gamma_1^k)^\top - h_1'(C) \odot \sum_k \xi_k \gamma^k h_2(C^k)(\gamma^k)^\top \\
&= f_1'(C) \odot \underbrace{\sum_k \xi_k \gamma_1^k(\gamma_1^k)^\top}_{B} - h_1'(C) \odot \underbrace{\sum_k \xi_k \gamma^k h_2(C^k)(\gamma^k)^\top}_{A}.
\end{aligned}
\tag{84}
$$

We claim $\frac{A}{B}$ is well-defined, i.e., if $B_{ij} = 0$, then $A_{ij} = 0$.

For each $i, j \in [1 : n]$, if $B_{ij} = 0$, we have two cases:

Case 1: $\forall k \in [1 : K]$, we have $\gamma_1^k[i] = 0$.

Thus, $\gamma^k[i, :] = 0_{n_k}^\top$. So $A[i, :] = (\gamma^k h_2(C^k)(\gamma^k)^\top)[i, :] = 0_{n_k}^\top$.

Case 2: $\forall k \in [1 : K]$, we have $\gamma_1^k[j] = 0$.

It implies $(\gamma^k)^\perp[:, j] = 0_n$, thus $A[:, j] = (\gamma^k h_2(C^k))(\gamma^k)^\top[:, j] = 0_{n_k}$. Therefore, $A_{ij} = 0$.

Thus $\frac{A}{B}$ is well-defined.

In addition, in these two cases, if we change the value $C_{ij}^k$, $\mathcal{L}$ will not change.

From (84), we have:

$$
\left( \frac{f_1'}{h_1'}(C) \right)_{ij} = \frac{\left( \sum_k \xi_k \gamma^k h_2(C^k)(\gamma^k)^\top \right)_{ij}}{\left( \sum_k \xi_k \gamma_1^k(\gamma_1^k)^\top \right)_{ij}}
$$

if $B_{ij} > 0$. In addition, if $B_{ij} = 0$, there is no constraint for $C_{ij}$.

Combining it with the fact that if $B_{i,j} = 0$, then $C_{i,j}$ does not affect $\mathcal{L}$. Thus,

we have the following is a solution:

$$
C = \left( \frac{f_1'}{h_1'} \right)^{-1} \left( \frac{\sum_k \xi_k \gamma^k h_2(C^k)(\gamma^k)^\top}{\sum_k \xi_k \gamma_1^k(\gamma_1^k)^\top} \right).
$$

In particular case: $|\mathrm{p}| \le |\mathrm{p}^k|, \forall k$, suppose $\lambda > \max\{c^2 : c \in \bigcup_k C^k \cup C\}$, by lemma E.1, we have for each $k$, $|\gamma^k| = \min(|\mathrm{p}|, |\mathrm{p}|^k) = |\mathrm{p}|$, that is $\gamma_1^k = \mathrm{p}$.

Thus,

$$
\sum_k \xi_k \gamma_1^k(\gamma_1^1)^\top = \sum_k \xi_k \gamma_1^k(\gamma_1^k)^\top = \sum_k \xi_k \mathrm{p}\mathrm{p}^\top = \mathrm{p}\mathrm{p}^\top
$$

Thus, $C = \left( \frac{f_1'}{h_1'} \right)^{-1} \left( \frac{\sum_k \xi_k \gamma^k h_2(C^k)(\gamma^k)^\top}{\mathrm{p}\mathrm{p}^\top} \right)$. $\qquad \square$

**Remark M.3.** *In $l^2$ loss case, i.e. $L(r_1, r_2) = |r_1 - r_2|^2$, (83) becomes*

$$
C = \frac{\sum_k \xi_k \gamma^k C^k(\gamma^k)^\top}{\sum_k \xi_k \gamma_1^k(\gamma_1^k)^\top}.
\tag{85}
$$

Since in this case, we can set

$$
f_1(x) = x^2, f_2(y) = y^2, h_1(x) = 2x, h_2(y) = y.
$$

Thus $\frac{f_1'}{h_1'}(x) = \frac{2x}{2} = x$ and $\left( \frac{f_1'}{h_1'} \right)^{-1}(x) = x$. Therefore, (83) becomes (85).

---

**Algorithm 3:** Partial Gromov-Wasserstein Barycenter

---

**Input:** $\{C^k, \mathrm{p}^k, \lambda_k\}_{k=1}^K, \mathrm{p}$
**Output:** $C$
Initialize $C$.
**for** $i = 1, 2, \ldots$ **do**
    compute $\gamma^k \leftarrow \arg\min_{\gamma \in \Gamma_{\leq}(\mathrm{p}, \mathrm{p}^k)} \langle \mathcal{L}(C, C^k) - 2\lambda_k, \gamma \rangle, \forall k \in [1:K]$.
    Update $C$ by (83).
    if convergence, break
**end for**

---

---

**Algorithm 4:** Mass-Constrained Partial Gromov-Wasserstein Barycenter

---

**Input:** $\{C^k, \mathrm{p}^k, \lambda_k\}_{k=1}^K, \mathrm{p}$
**Output:** $C$
Initialize $C$.
**for** $i = 1, 2, \ldots$ **do**
    compute $\gamma^k \leftarrow \arg\min_{\gamma \in \Gamma_{\leq}^{\rho_k}(\mathrm{p}, \mathrm{p}^k)} \langle \mathcal{L}(C, C^k), \gamma \rangle, \forall k \in [1:K]$.
    Update $C$ by (83).
    if convergence, break
**end for**

---

Similarly, we can also extend the above PGW Barycenter into the MPGW setting:

$$\min_{C, \gamma^k} \sum_{k=1}^K \xi_k \langle L(C, C^k) \circ \gamma^k, \gamma^k \rangle,$$

where, for each $k \in [1:K]$, $\rho_k \in [0, \min(|\mathrm{p}|, |\mathrm{p}^k|)]$, and the optimization is over $C \in \mathbb{R}^n$ and $\gamma_k \in \Gamma_{\leq}^{\rho_k}(\mathrm{p}, \mathrm{p}^k)$ for $k \in [1:K]$.

It can be solved by the following algorithm 4.

### M.1 Details of Shape Interpolation Experiment

**Dataset and data processing.** We apply the dataset in (Peyré et al., 2016) with download link. The original data are images, which we convert into a point cloud using the k-mean algorithm, where $k = 1024$ (see the second row of Figure 4).

Suppose $\mathcal{D} \subset \mathbb{R}^2$ is a region that contains these point clouds. Let $\mathcal{R} \subset \mathbb{R}^2$ denote another region. In $\mathcal{R}$, we randomly select and add $n\eta$ noise points to these point clouds. In particular, we consider noise corruption in the following three cases:

Case 1: $\mathcal{R}$ is a rectangle region which is disjoint to $\mathcal{D}$. See the third row in Figure 4.

Case 2: $\mathcal{R} = \mathcal{R}_1 \cup \mathcal{R}_2$, where $\mathcal{R}_1, \mathcal{R}_2$ are rectangles which are disjoint to $\mathcal{D}$. See the fourth row in Figure 4.

Case 3: $\mathcal{R}$ contains $\mathcal{D}$. See the fifth row in Figure 4.

**GW Barycenter and PGW Barycenter methods**. We select $t_1, \ldots, t_K$ with $0 = t_1 < t_2 < \ldots < t_K = 1$. For each $t \in \{t_1, \ldots, t_K\}$, we compute the GW Barycenter

$$\arg\min_{C, \gamma^1, \gamma^2} (1 - t)\langle L(C, C^1) \circ \gamma^1, \gamma^1 \rangle + t\langle L(C, C^2) \circ \gamma^2, \gamma^2 \rangle, \tag{86}$$

where $\gamma_1 \in \Gamma(\mathrm{p}, \mathrm{p}^1), \gamma_2 \in \Gamma(\mathrm{p}, \mathrm{p}^2)$. Apply Smacof-MDS to the minimizer $C$, the resulting embedding, denoted as $X_t \in \mathbb{R}^{n \times 2}$ (where $n = 1024$) is the GW-based interpolation.

Replacing the GW Barycenter with the PGW Barycenter

$$\arg\min_{C, \gamma^1, \gamma^2} (1 - t)(\langle L(C, C^1) \circ \gamma^1, \gamma^1 \rangle + \lambda_1 |\gamma^1|^2) + t(\langle L(C, C^2) \circ \gamma^2, \gamma^2 \rangle + \lambda_2 |\gamma^2|^2), \tag{87}$$

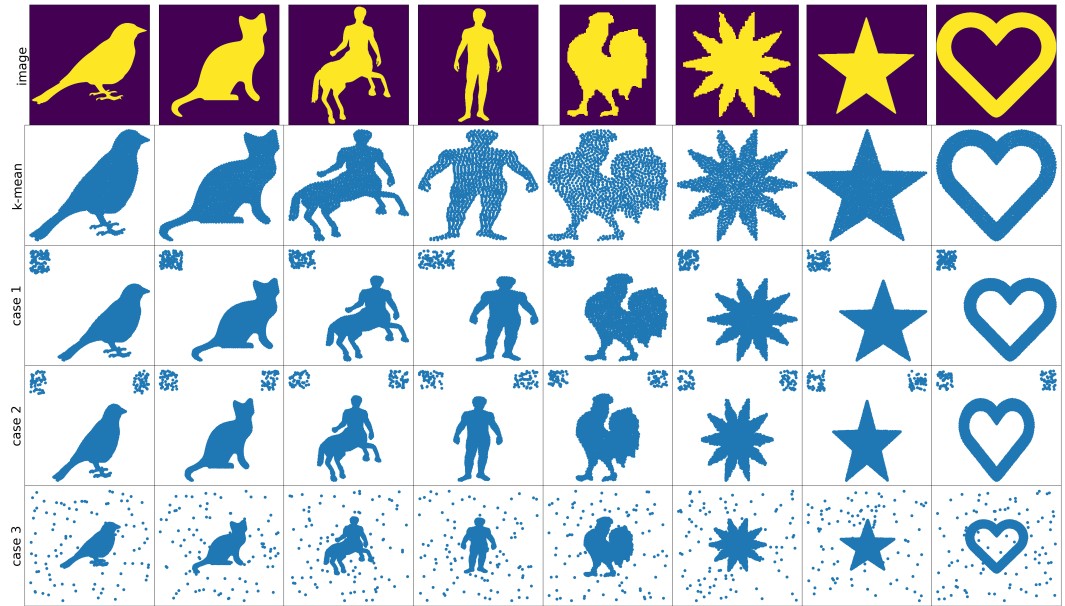

Figure 4: We visualize the dataset in shape interpolation. The first row is the original images in Link. The second row is the point clouds obtained by the k-mean method, where $k = 1024$.

where $\lambda_1, \lambda_2 > 0, \gamma^1 \in \Gamma_{\leq}(\mathrm{p}, \mathrm{p}^1), \gamma^2 \in \Gamma_{\leq}(\mathrm{p}, \mathrm{p}^2)$. Then, we obtain PGW-based interpolation.

**Problem setup**. We select one point cloud from the clean dataset denoted as $X = \{x_i\}_{i=1}^n$ (source point cloud), $n = 1024$.

Next, we select one noise-corrupted point cloud, as described in Case 1, Case 2, and Case 3, respectively. In these three scenarios, we test $\eta = 0.5\%$ and $\eta = 10\%$ where $\eta$ is the noise level. Therefore, we test $3 * 2 = 6$ different interpolation tasks for these two methods. The size of the target point cloud is then $m = n + n\eta$. See Figure 5 for details.

**Numerical details.** In the GW-barycenter method, because of the balanced mass setting, we set

$$\mathrm{p}^1 = \frac{1}{n}\mathbf{1}_n, \mathrm{p}^2 = \frac{1}{m}\mathbf{1}_m, \mathrm{p} = \frac{1}{n}\mathbf{1}_n.$$

In PGW-barycenter, we set

$$\mathrm{p}^1 = \frac{1}{n}\mathbf{1}_n, \mathrm{p}^2 = \frac{1}{n}\mathbf{1}_m, \mathrm{p} = \frac{1}{n}\mathbf{1}_n.$$

In addition, we set $\lambda_1, \lambda_2$ such that $2\lambda_1, 2\lambda_2 \geq \max(\max(C_1)^2, \max(C_2)^2)$. We compute GW/PGW barycenter for $t = 0/7, 1/7, \ldots, 7/7$.

In both GW and PGW barycenter algorithms, we set the largest number of iterations to 100. The threshold for convergence is set to be 1e-5.

**Performance analysis.** Each interpolation task is essentially unbalanced: the source point cloud contains clean data, while the target point cloud contains clean and noise points. We observe that in the first two scenarios, the interpolation derived from GW is clearly disturbed by the noise data points. For example, in rows $1, 3, 5, 7$, columns $t = 1/7, 2/7, 3/7$, we see that the point clouds reconstructed by MDS have significantly different width-height ratios from those of the source and target point clouds.

In contrast, PGW is significantly less disturbed, and the interpolation is more natural. The width-height ratio of the point clouds generated by the PGW barycenter is consistent with that of the source/target point clouds.

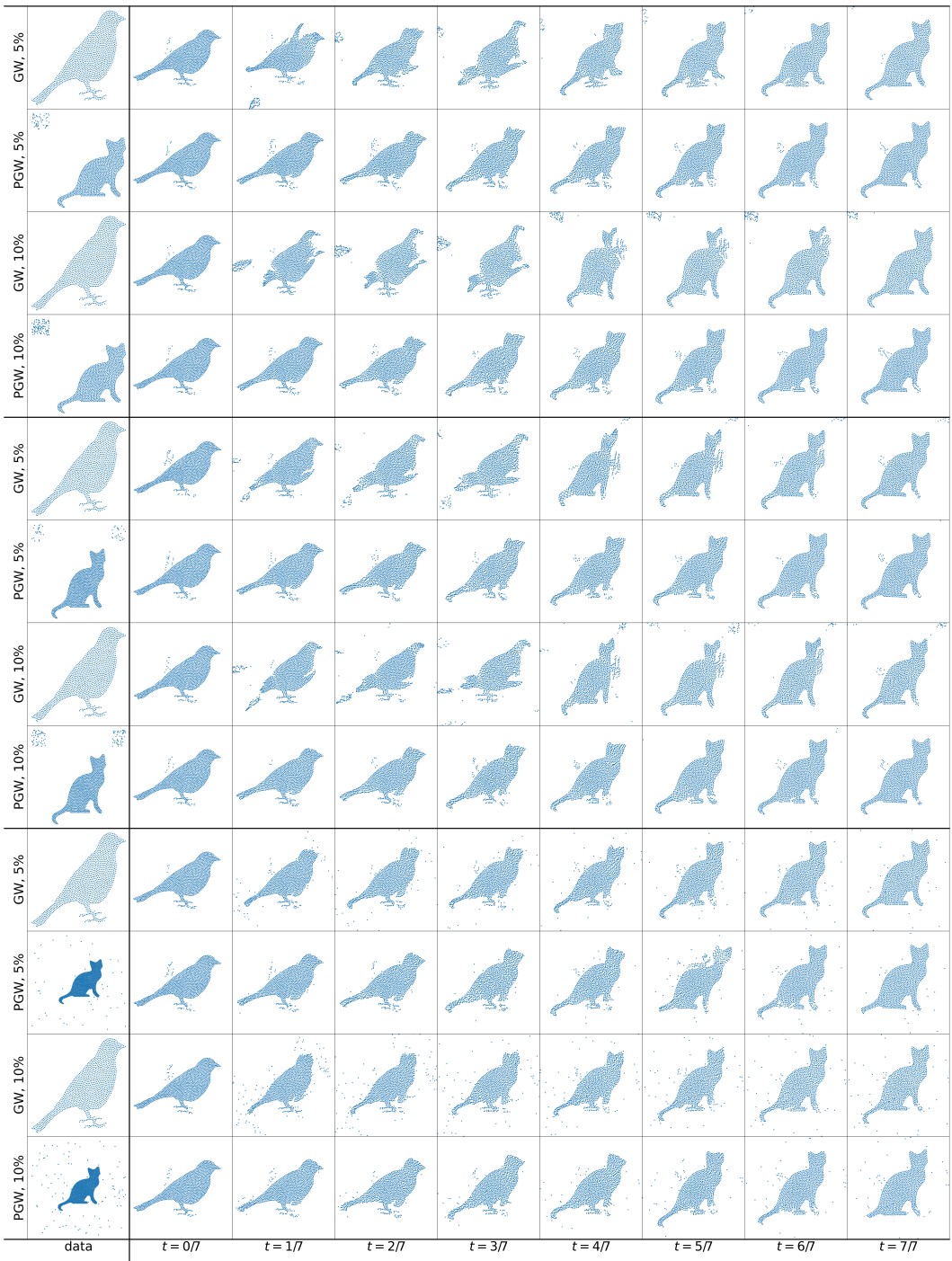

Figure 5: We test interpolation tasks in 3 scenarios: source data is clean, target data is selected from three cases as described in section **dataset and data processing**. In each scenario, we test $\eta = 5\%, 10\%$ respectively. In the first column, we present the source and target point cloud visualization in each task. In columns 2-9, we present GW, PGW barycenter for $t = 0/7, 1/7, \ldots, 7/7$.

In the third scenario, the noise data is uniformly selected from a large region that contains the domain of all clean point clouds. In this case, we observe that the GW and PGW barycenters perform similarly. However, at $t = 1/7, 2/7, 4/7$, GW-barycenters present more noise points than PGW-barycenters in the same truncated region.

**Limitations and future work**. The main issue of the above GW/PGW techniques arises from the MDS method:

Given minimizer $C \in \mathbb{R}^{n \times n}$ of GW/PGW barycenter problem (86) (or (87)), MDS studies the following problem:

$$\min_{X \in \mathbb{R}^{n \times d}} \sum_{i, i'=1}^{n} \left| C_{i,i'}^{1/2} - \|X_i - X_{i'}\| \right|^2 \tag{88}$$

Let $O(n)$ denote the set of all $n \times n$ orthonormal matrices. Suppose $X^*$ is a minimizer, then $RX^*$ is also a minimizer for the above problem for all $R \in O(n)$.

In practice, this means manually setting suitable rotation and flipping matrices for each method at each step, especially for the GW method.

However, we understand that this issue stems from the inherent properties of the GW/PGW method. GW can be seen as a tool that describes the similarity between two graphs, which are rotation-invariant and flipping-invariant. Therefore, the GW/PGW barycenter essentially describes the interpolation between two graphs rather than two point clouds.

M.2 MULTI-SHAPE INTERPOLATION

In this section, we present the interpolation between 4 different shapes. In addition, let $\eta$ be the percentage of outliers; we test $\eta = 0$ and 5%.

**Experiment setup.** We select 4 shapes (bird, cat, human, and rooster) from the 2D clouds dataset. Two of them, i.e., cat and rooster, are embedded into 4D space and are randomly rotated. The goal is to find the interpolation between them.

**Baselines.** We select 4 baselines, optimal transport barycenter (Cuturi & Doucet, 2014), partial optimal transport barycenter (Bonneel & Coeurjolly, 2019), GW barycenter (Peyré et al., 2016) and our PGW barycenter.

Note that OT and partial OT barycenter can not be directly applied to this setting. We first embed the 2D shapes into 4D space via mapping

$$\mathbb{R}^2 \ni x \mapsto [x; 0; 0] \in \mathbb{R}^4,$$

and then compute the OT/POT barycenter. Finally, we apply PCA to project the result back to 2D space for visualization.

**Result analysis.** We present the results in Figures 6 and 7. When $\eta = 0$, from Figure 6, we observe that OT/POT admits good interpolation results between 2D shapes (bird and human). However, the OT/POT interpolation between 2D and 4D is not as good as GW/PGW. The main reason is that the OT/POT barycenter incorporates the absolute coordinates of these shapes for interpolation, which is affected by random rotation. However, GW/PGW is not affected by the absolute coordinates of the shapes.

In the second figure, we demonstrate the result for $\eta = 5\%$. Two shapes are corrupted by the noise points (cat and rooster). We observe that GW's interpolation is affected by the outliers (see, e.g.,a from bird to human or from cat to rooster), while PGW is less affected by these outlier points and admits much better/smoother interpolation.

N DETAILS OF SHAPE MATCHING

**Dataset setup**. In the Moon dataset (see link), we apply $n = 200$ and set Gaussian variance to be 0.2. The outliers are sampled from region $[[-2, -1.5] \times [-3.5, -3]]$.

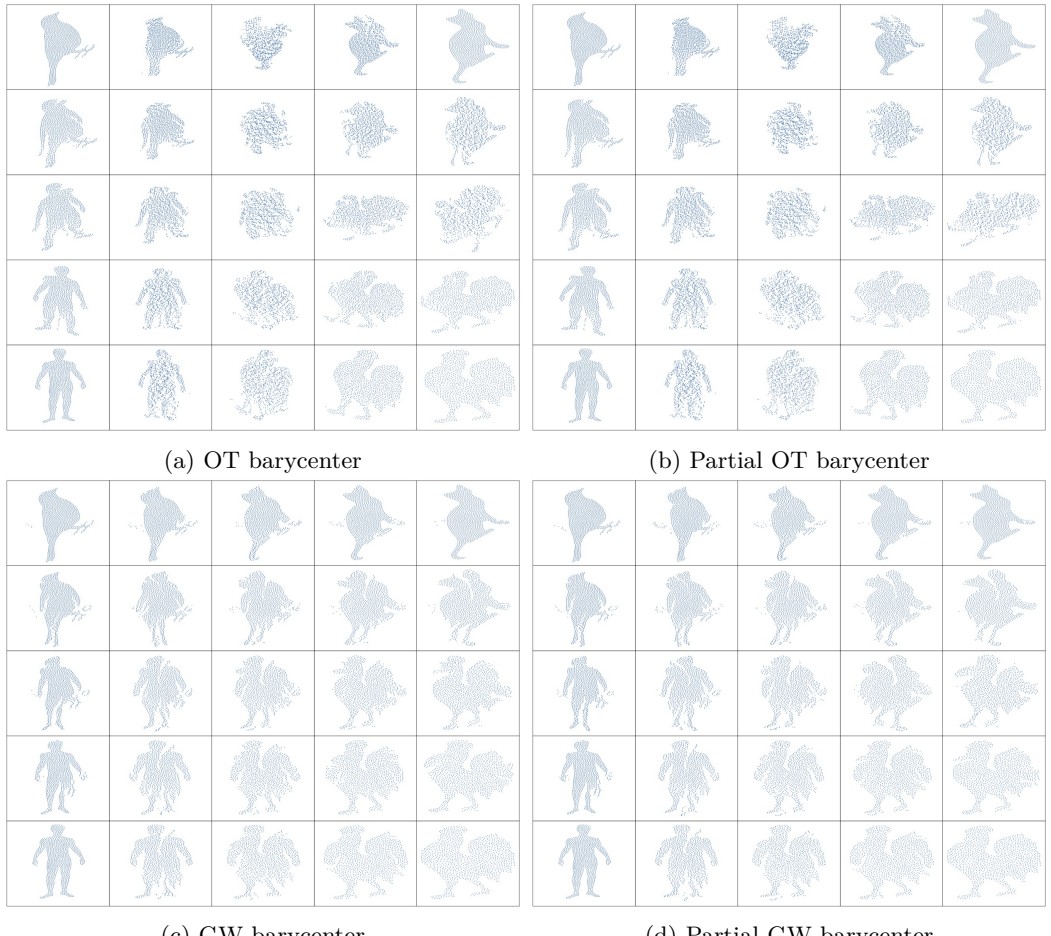

(a) OT barycenter       (b) Partial OT barycenter

(c) GW barycenter       (d) Partial GW barycenter

Figure 6: We visualize multi-shapes interpolation, where the proposition of noise $\eta$ is 0. "bird" and "human" shapes are distributed in 2D space, "cat" and "rooster" are distributed in 4D space.

In the second experiment, the circle data is uniformly sampled from a 2D circle

$$\mathbb{S}^1 = \{s \in \mathbb{R}^2 : \|s\|^2 = 1\}$$

and spherical data is uniformly sampled from 3D sphere

$$\mathbb{S}^2 = \{s + [0, 0, 4] \in \mathbb{R}^2 : \|s\|^2 = 1\},$$

where the shift $[0, 0, 4]$ is applied for visualization.

We set sample size $n = 200$ for both 2D and 3D samples.

In both experiments, the number of outliers is $\eta n = 0.2n = 40$.

**Numerical details.** In GW, we normalize the two-point clouds as

$$\mathbb{X} = (X, d_X, \sum_{i=1}^{n} \frac{1}{n} \delta_{x_i}), \mathbb{Y} = (Y, d_Y, \sum_{j=1}^{n+n\eta} \frac{1}{n + n\eta} \delta_{y_j}).$$

In PGW, MPGW, and UGW, we define the point clouds as

$$\mathbb{X} = (X, d_X, \sum_{i=1}^{n} \frac{1}{n} \delta_{x_i}), \mathbb{Y} = (Y, d_Y, \sum_{j=1}^{n+n\eta} \frac{1}{n} \delta_{y_j}).$$

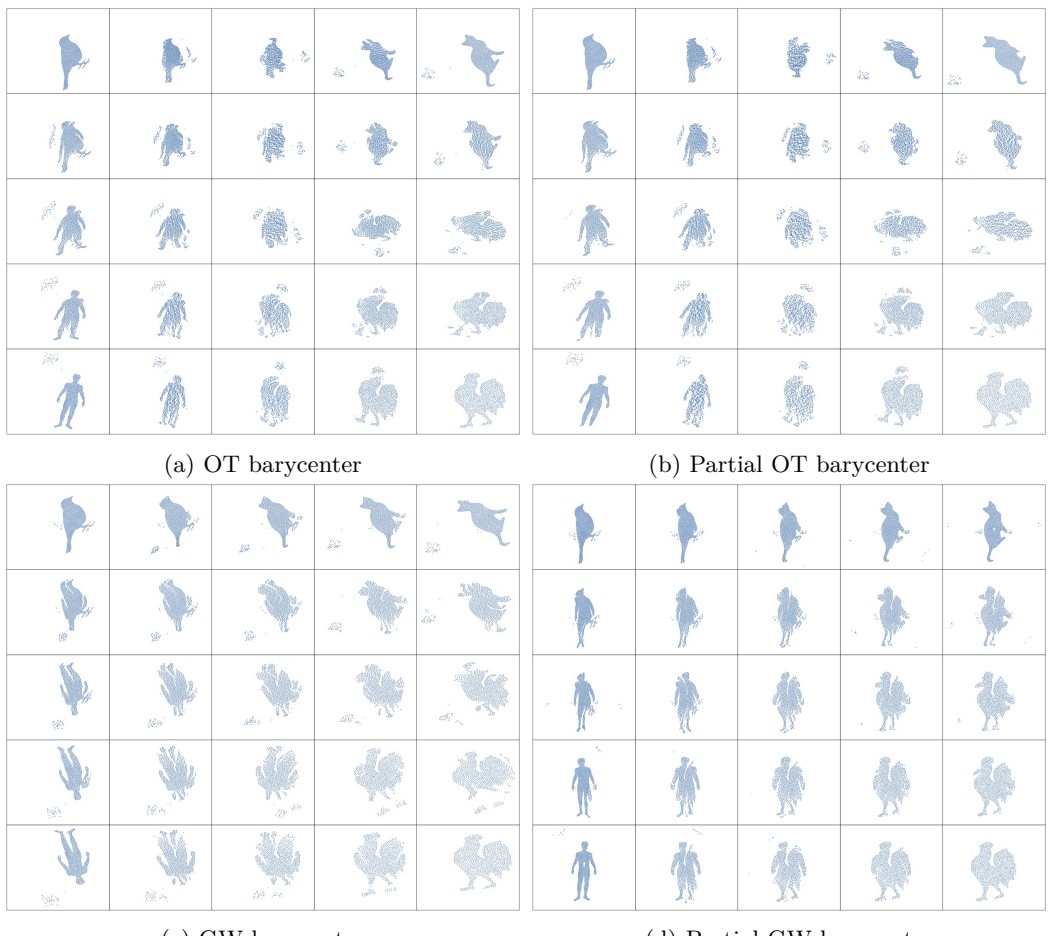

(a) OT barycenter        (b) Partial OT barycenter

(c) GW barycenter        (d) Partial GW barycenter

Figure 7: We visulize of multi-shapes interporlation. "bird","human" shapes are distributed in 2D space, "cat", "rooster" are distributed in 4D space. The noise level, $\eta$, is 5%.

In PGW, we choose $\lambda$ such that $\lambda \geq \max(\max((C^X)^2), \max((C^Y)^2))$, in particular, $\lambda = 10.0$. In addition, we tested $\lambda = 0.01, 0.1, 1.0, 10.0$.

In MPGW, we tested $\rho = 0.3, 0.5, 0.8, 1.0$ and selected $\rho = 1.0$

In UGW, we tested $[1e - 10, 1e - 1, 1.0, 10]$ and selected $\rho_1 = \rho_2 = 1.0$, $\epsilon = 0.05$.

We refer to Figure 8 for the visualization of all tested parameters.

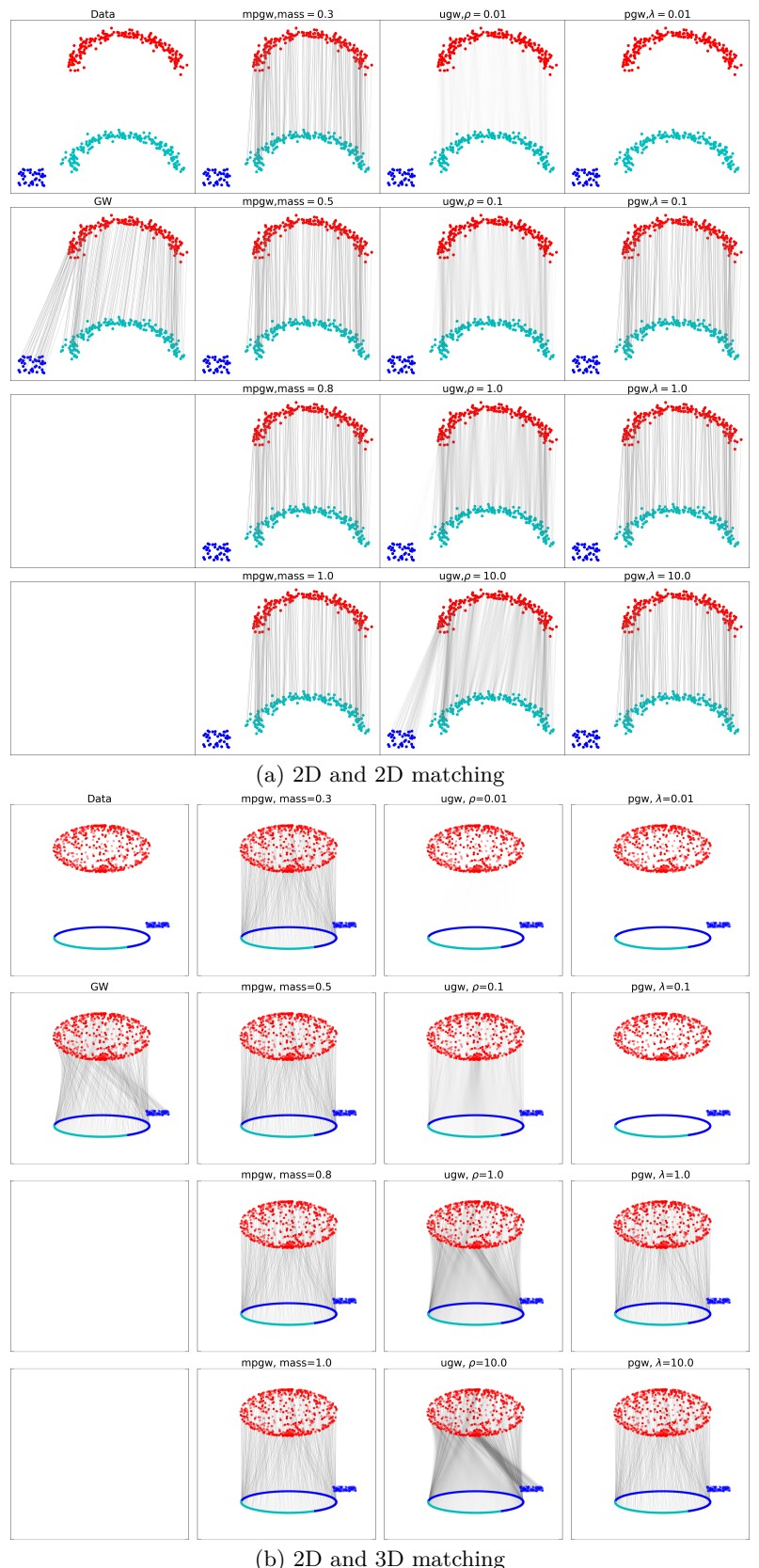

(a) 2D and 2D matching

(b) 2D and 3D matching

Figure 8: We visualize the shape-matching result for all tested parameters.

# O  DETAILS OF SHAPE RETRIEVAL EXPERIMENT

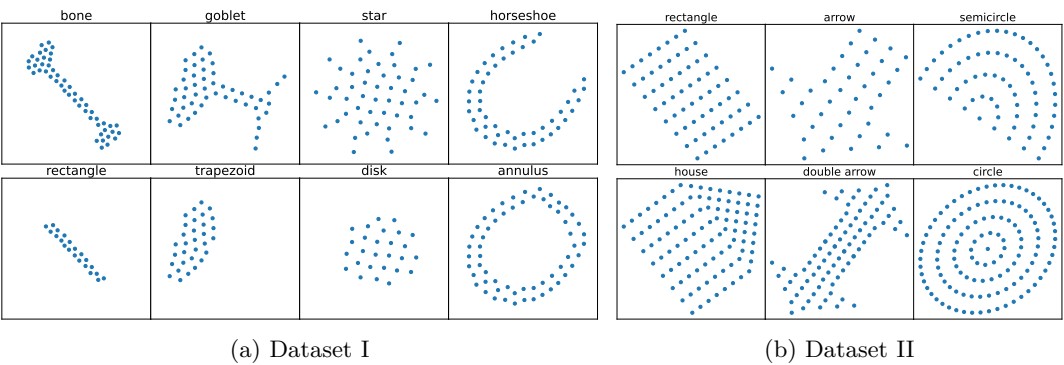

(a) Dataset I                                    (b) Dataset II

Figure 9: Visualization of a representative shape from each class of the two datasets.

**Dataset details.** We test two datasets in this experiment, which we refer to as Dataset I and Dataset II. We visualize Dataset I in Figure 9a and Dataset II in Figure 9b. The complete datasets can be accessed from the supplementary materials.

**Numerical details.** We represent the shapes in each dataset as mm-spaces $\mathbb{X}^i = \left( \mathbb{R}^2, \| \cdot \|_2, \mu^i = \sum_{k=1}^{n^i} \alpha^i \delta_{x_k^i} \right)$. We use $\alpha^i = \frac{1}{n^i}$ to compute the GW distances for the balanced mass constraint setting. For the remaining distances, we set $\alpha = \frac{1}{N}$, where $N$ is the median number of points across all shapes in the dataset. For the SVM experiments, we use $\exp(-\sigma D)$ as the kernel for the SVM model. Here, we normalize the matrix $D$ and choose the best $\sigma \in \{0.001, 0.01, 0.1, 1, 10, 100, 1000, 10000\}$ for each method used in order to facilitate a fair comparison of the resulting performances. We note that the resulting kernel matrix is not necessarily positive semidefinite.

In computing the pairwise distances, for the PGW method, we set $\lambda$ such that $\lambda \leq \lambda_{max} = \max_i (|C^i|^2)$. In particular, we compute $\lambda_{max}$ for each dataset and use $\lambda = \frac{1}{5}\lambda_{max}$ for each experiment. For UGW, we use $\varepsilon = 10^{-1}$ and $\rho_1 = \rho_2 = 1$ for both experiments. Finally, for MPGW, we set the mass-constrained term to be $\rho = \min(|\mu^i|, |\mu^j|)$ when computing the similarity between shape $\mathbb{X}^i$ and $\mathbb{X}^j$.

**Performance analysis.** The pairwise distance matrices are visualized for each dataset in Figure 10, and the confusion matrices computed with each dataset are given in Figure 11. Finally, the classification accuracy with the SVM experiments is reported in Table 1a. The results indicate that the PGW distance is able to obtain high performance across both datasets consistently.

In addition, from Figure 10, we observe that PGW qualitatively admits a more reasonable similarity measure compared to other methods. For example, in Dataset I, class "bone" and "rectangle" should have relatively smaller distances than "bone" and "annulus". Ideally, a reasonable distance should satisfy the following:

$$0 < d(\text{bone}, \text{rectangle}) < d(\text{bone}, \text{anulus}).$$

However, we do not observe this relation in GW and UGW[4], and for the MPGW method, $MPGW(\text{bone}, \text{rectangle}) \approx 0$, which is also undesirable. For PGW, however, we do observe this relation. Additionally, we report the wall-clock time comparison in Table 1b.

## O.1  ADDITIONAL EXPERIMENT WITH OT/UT BASELINES

In this subsection, we incorporate optimal transport distance (Villani, 2003) and unbalanced optimal transport distance (Chizat et al., 2018c) into the shape retrieval experiment.

---

[4]For UGW, this is due to the Sinkhorn regularization term.

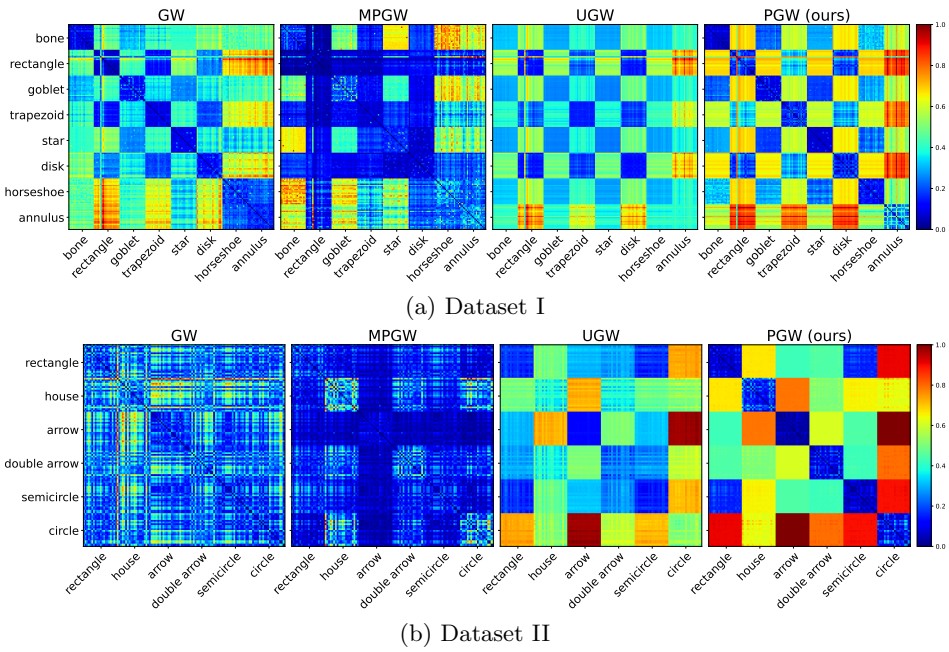

(a) Dataset I

(b) Dataset II

Figure 10: Pairwise distance matrices computed for each dataset.

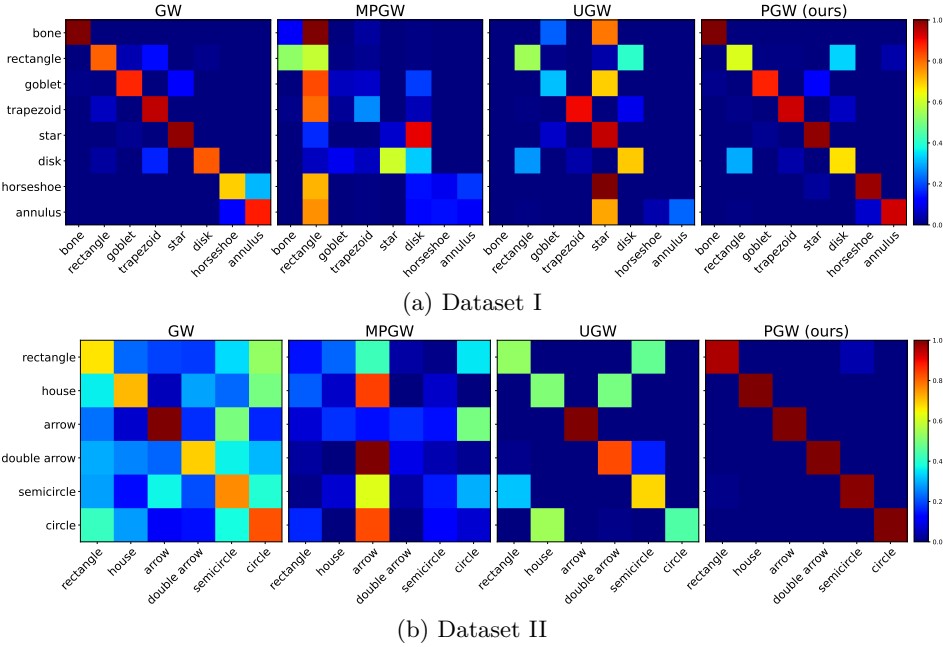

(a) Dataset I

(b) Dataset II

Figure 11: Confusion matrices computed from nearest neighbor classification experiments.

**Dataset setup.** We perform shape retrieval experiments in two scenarios: the original 2D dataset and a 4D dataset. For the 4D scenario, we embed the shapes into 4D space and then apply a random rotation to each shape.

**Baselines.** We evaluate the performance of the following methods: Optimal Transport distance (OT) (Villani, 2003), Unbalanced Optimal Transport (UOT) distance (Chizat et al., 2018c), Gromov-Wasserstein distance (GW) (Gromov, 2001), Mass-constrained Gromov-Wasserstein discrepancy (MPGW) (Chapel et al., 2020), Unbalanced Gromov-Wasserstein

| Method | Dataset I | Dataset II |
|---|---|---|
| OT ($d = 2$) | 95.6% | 55.8% |
| OT ($d = 4$) | 79.3% | 50.8% |
| UOT ($d = 2$) | 73.5% | 73.3% |
| UOT ($d = 4$) | 56.9% | 63.3% |
| GW ($d = 2, 4$) | 98.1% | 80.8% |
| MPGW ($d = 2, 4$) | 23.7% | 25.0% |
| UGW ($d = 2, 4$) | 89.4% | 90.0% |
| PGW (ours, $d = 2, 4$) | 96.2% | 100% |

Table 2: Performance comparison across different methods and datasets.

discrepancy (UGW) (Séjourné et al., 2021), and our Partial Gromov-Wasserstein (PGW) distance.

The parameter settings for GW, MPGW, UGW, and PGW are described in the previous section. For the UOT method, we set the weight of Sinkhorn regularization to 0.1 and the weight of the marginal penalty to 0.2.

**Result.** In Dataset 1, when data is in the 2D space, OT achieves an accuracy of 95.6%, while UOT achieves 73.5%. OT's accuracy is slightly lower than GW/PGW, but it remains a strong classifier in this setting. In Dataset 2, UOT outperforms OT with an accuracy of 73.3% compared to 55.8%.

When the shapes are embedded into 4D space, the accuracy of OT and UOT drops significantly, ranging from 56.9% to 79.3%, far below the performance of GW and PGW. This decline highlights the reliance of OT/UOT on absolute coordinates, which becomes more problematic in higher dimensions. In contrast, GW-based methods (GW, MPGW, UGW, PGW) remain unaffected, as they are invariant to absolute locations. Overall, regardless of whether the data is in 2D or 4D space, OT and UOT consistently perform worse than GW and its variants. The gap in performance becomes even more pronounced in 4D space.

# P    OTHER NUMERICAL IMPLEMENTATIONS

## P.1    INITIAL METHODS

In this experiment, we discuss several different methods to define the initial guess in the Frank-Wolfe algorithm proposed in this paper. Note some of these methods have been applied in FW algorithms/Sinkhorn solvers in classical GW (Mémoli, 2011), Mass-constraint GW (Chapel et al., 2020) and Unbalanced GW (Séjourné et al., 2021)

Given two mm-spaces $\mathbb{X} = (X, d_X, \mu), \mathbb{X} = (Y, d_Y, \nu)$, we consider the two cases:

Case 1: Dimensions of $X$ and $Y$ are the same. Note, in this case, we can define classical OT/partial OT/unbalanced OT between $\mu$ and $\nu$. Thus, (Chapel et al., 2020) proposed the following "**POT initilization**" method:

$$\gamma^{(1)} \leftarrow \arg\min_{\gamma \in \Gamma_{\leq,\pi}(\mathrm{p},\mathrm{q})} \langle L(X, Y), \gamma \rangle_F, \tag{89}$$

where $L(X, Y) \in \mathbb{R}^{n \times m}$, $(L(X, Y))_{ij} = \|x_i - y_j\|^p$ for some fixed $p \geq 1$ and

$$\Gamma_{\leq,\pi}(\mathrm{p},\mathrm{q}) := \{\gamma \in \mathbb{R}_+^{n \times m} : (\gamma^\top 1_n)_j \in \{q_j^Y, 0\}, \forall j; \gamma 1_m \leq \mathrm{p}, |\gamma| = \pi\}. \tag{90}$$

The above problem can be solved by a Lasso ($L^1$ norm) regularized OT solver.

Case 2: Dimensions of $X$ and $Y$ are different. The above technique can not be applied since the problem (89) (in particular $L(X, Y)$) is not well-defined.

In this case, (Séjourné et al., 2021) introduced the **"FLB-UOT"** method:

$$\gamma^{(1)} \leftarrow \arg \min_{\gamma \in \Gamma_{\leq}(\mathrm{p,q})} \int_{X \times Y} |s_{X,p}(x) - s_{Y,p}(y)|^p d\gamma(x,y) + \lambda(D_{KL}(\gamma_1, \mathrm{p}) + D_{KL}(\gamma_2, \mathrm{q})), \tag{91}$$

where $s_{X,p}(x) = \int_X |x - x'|^p d\mu(x)$ and $s_{Y,p}$ is defined similarly. The problem (91) is called Hellinger Kantorovich, which is a classical unbalanced optimal transport problem. The Sinkhorn solver can solve it (Chizat et al., 2018b).

Analog to the above method, we propose the third method, called "**FLB-POT**" (first lower bound-partial optimal transport)

$$\gamma^{(1)} \leftarrow \arg \min_{\gamma \in \Gamma_{\leq}(\mathrm{p,q})} \int_{X \times Y} |s_{X,p}(x) - s_{Y,p}(y)|^2 d\gamma(x,y) + \lambda(|\mathrm{p} - \gamma_1| + |\mathrm{q} - \gamma_2|). \tag{92}$$

The above problem is a partial OT problem and can be solved by classical linear programming (Caffarelli & McCann, 2010).

## P.2 Parameter Setting for PGW

Setting parameter $\lambda$ in PGW is important in the numerical implementation. There are two scenarios to consider:

- Both source and target measures contain outliers:
  The parameter $\lambda$ acts as an upper bound for the transported distance. Specifically, if
  $$\|d_X(x, x') - d_Y(y, y')\|^2 \geq 2\lambda,$$
  then either $(x, y)$ or $(x', y')$ will not be transported. - When the distance between outliers and clean data is large, and the pairwise distances within the clean data are relatively small, $\lambda$ should be set to lie between these two scales.

- Only one measure contains outliers:
  As stated in Lemma E.2, $\lambda$ can be set sufficiently large to satisfy:
  $$2\lambda \geq \max_{x, x' \in X, y, y' \in Y} \|d_X(x, x') - d_Y(y, y')\|^2.$$

  In the interpolation and shape retrieval experiments, we fall under the second scenario, which does not require significant parameter tuning.

## Q    WALL-CLOCK TIME COMPARISON FOR PARTIAL GW SOLVERS

In this section, we present the wall-clock time comparison between our method Algorithms 1, 2, the Frank-Wolf algorithm proposed in (Chapel et al., 2020), and its Sinkhorn version (Peyré et al., 2016; Chapel et al., 2020). Note that these two baselines solve a mass constraint version of the PGW problem, which we refer to as the "MPGW" problem. The proposed PGW formulation in this paper can be regarded as a "Lagrangian formulation" of MPGW[5] formulation to the PGW problem defined in (10). In this paper, we call these two baselines the "MPGW algorithm" and the "Sinkhorn PGW algorithm".

**Numerical details.**   The data is generated as follows:   let $\mu = \text{Unif}([0,2]^2)$ and $\nu = \text{Unif}([0,2]^3)$, we select i.i.d. samples $\{x_i \sim \mu\}_{i=1}^n, \{y_j \sim \nu\}_{j=1}^m$, where $n$ is selected from $[10, 50, 100, 150, ..., 10000]$ and $m = n + 100$, $\text{p} = 1_n/m, \text{q} = 1_m/m$. For each $n$, we set $\lambda = 0.2, 1.0, 10.0$. The mass constraint parameter for the algorithm in (Chapel et al., 2020), and Sinkhorn is computed by the mass of the transportation plan obtained by Algorithm 1 or 2. The runtime results are shown in Figure 12.

Regarding the acceleration technique, for the POT problem in step 1, our algorithms and the MPGW algorithm apply the linear programming solver provided by Python OT package (Flamary et al., 2021), which is written in C++. The Sinkhorn algorithm from Python OT does not have an acceleration technique. Thus, we only test its wall-clock time for $n \leq 2000$. The data type is a 64-bit float number.

From Figure 12, we can observe the Algorithms 1, 2, and MPGW algorithm have a similar order of time complexity. However, using the column/row-reduction technique for the POT computation discussed in previous sections and the fact the convergence behaviors of Algorithms 1 and 2 are similar to the MPGW algorithm, we observe that the proposed algorithms 1, 2 admits a slightly faster speed than MPGW solver.

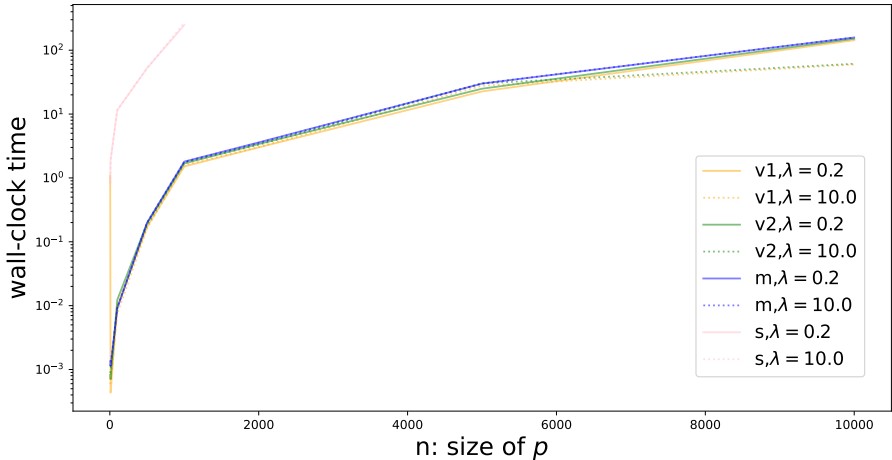

Figure 12: We test the wall-clock time of our Algorithm 1 and Algorithm 2, the MPGW solver (Algorithm 1 in (Chapel et al., 2020)) , and the Sinkhorn algorithm (Peyré et al., 2016). We denote these methods as "v1", "v2", "m" and "s", respectively. The linear programming solver applied in the first three methods is from POT (Flamary et al., 2021), which is written in C++. The maximum number of iterations for all the methods is set to be 1000. The maximum iteration for OT/OPT solvers is set to be $300n$. The maximum Sinkhorn iteration is set to be 1000. The convergence tolerance for the Frank-Wolfe algorithm and the Sinkhorn algorithm are set to be $1e - 5$. To achieve the best performance, we set the number of dummy points to 1 for both MPGW and PGW.

---

[5]Due to the non-convexity of GW, we do not have a strong duality in some of the GW representations. Thus, the Lagrangian form is not a rigorous description.

## R  POSITIVE UNLABELED PROBLEM

### R.1  PROBLEM SETUP

Positive unlabeled (PU) learning (Bekker & Davis, 2020; Elkan & Noto, 2008; Kato et al., 2018) is a semi-supervised binary classification problem for which the training set only contains positive samples. In particular, suppose there exists a fixed unknown overall distribution over triples $(x, o, l)$, where $x$ is data, $l \in \{0, 1\}$ is the label of $x$, $o \in \{0, 1\}$ where $o = 1$, $o = 0$ denote that $l$ is observed or not, respectively. In the PU task, the assumption is that only positive samples' labels can be observed, i.e., $\mathrm{Prob}(o = 1|x, l = 0) = 0$. Consider training labeled data $X^{pu} = \{(x_i^{pu}, l)\}_{i=1}^n \subset \{x : o = 1\}$ and testing data $X^{un} = \{x_j^{un}\}_{j=1}^m \subset \{x : o = 0\}$, where $x_i p_i^X \in \mathbb{R}^{d_1}, x_j^u \in \mathbb{R}^{d_2}$. In the classical PU learning setting, $d_2 = d_1$. However, in (Séjourné et al., 2021), this assumption is relaxed. The goal is to leverage $X^p$ to design a classifier $\hat{l} : x^u \to \{0, 1\}$ to predict $l(x^u)$ for all $x^u \in X^u$.[6]

Following (Elkan & Noto, 2008; Chapel et al., 2020; Séjourné et al., 2021), in this experiment, we assume that the "select completely at random" (SCAR) assumption holds: $\mathrm{Prob}(o = 1|x, l = 1) = \mathrm{Prob}(o = 1|l = 1)$. In addition, we use $\pi = \mathrm{Prob}(l = 1) \in [0, 1]$ to denote the ratio of positive samples in testing set[7]. Following the PU learning setting in (Kato et al., 2018; Hsieh et al., 2019; Chapel et al., 2020; Séjourné et al., 2021), we assume $\pi$ is known. In all the PU learning experiments, we fix $\pi = 0.2$.

### R.2  METHODOLOGY

Similar to (Chapel et al., 2020) our method is designed as follows: We set $\mathrm{p} \in \mathbb{R}^n, \mathrm{q} \in \mathbb{R}^m$ as $p_i^X = \frac{\pi}{n}, i \in [1 : n]$;  $q_j^Y = \frac{1}{m}, j \in [1 : m]$. Let $\mathbb{X}^p = (X^p, \|\cdot\|_{d_1}, \sum_{i=1}^n p_i^X \delta_{x_i}), \mathbb{X}^u = (X^u, \|\cdot\|_{d_2}, \sum_{j=1}^n q_j^Y \delta_{y_j})$. We solve the partial GW problem $PGW_\lambda(\mathbb{X}^p, \mathbb{X}^u)$ and suppose $\gamma$ is a solution. Let $\gamma_2 = \gamma^\top 1_n$. The classifier $\hat{l}$ is defined by the indicator function

$$\hat{l}_\gamma(x^u) = \mathbb{1}_{\{x^u : \gamma_2(x^u) \geq \text{quantile}\}}, \tag{93}$$

where quantile is the quantile value of $\gamma_2$ according to $1 - \pi$.

Regarding the initial guess $\gamma^{(1)}$, (Chapel et al., 2020) proposed a POT-based approach when $X$ and $Y$ are sampled from the same domain, i.e., $d_1 = d_2$, which we refer to as "POT initialization."

When $X, Y$ are sampled from different spaces, that is, $d_1 \neq d_2$, the above technique (89) is not well-defined. Inspired by (Mémoli, 2011; Séjourné et al., 2021), we propose the following "first lower bound-partial OT" (FLB-POT) initialization:

$$\gamma^{(1)} = \arg \min_{\gamma \in \Gamma_{\leq}(\mathrm{p}, \mathrm{q})} \int_{X \times Y} |s_{X,2}(x) - s_{Y,2}(y)|^2 d\gamma(x, y) + \lambda(|\mathrm{p} - \gamma_1| + |\mathrm{q} - \gamma_2|),$$

where $s_{X,2}(x) = \int_X |x - x'|^2 d\mu(x)$ and $s_{Y,2}$ is defined similarly. The above formula is analog to Eq. (7) in (Séjourné et al., 2021), which is designed for the unbalanced GW setting. To distinguish them, in this paper, we call Eq. (7) in (Séjourné et al., 2021) as "FLB-UOT initialization".

### R.3  DATASETS

The datasets include MNIST, EMNIST, and the following three domains of Caltech Office: Amazon (A), Webcam (W), and DSLR (D) (Saenko et al., 2010). For each domain, we select the SURF features (Saenko et al., 2010) and DECAF features (Donahue et al., 2014). For MNIST and EMNIST, we train an auto-encoder, respectively, and the embedding space dimension is 4 and 6, respectively. See Figure 13 for the TSNE visualization of these datasets.

---

[6]In the classical setting, the goal is to learn a classifier for all $x$. In this experiment, we follow the setting in (Séjourné et al., 2021).

[7]In the classical setting, the prior distribution $\pi$ is the ratio of positive samples of the original dataset. For convenience, we ignore the difference between this ratio in the original dataset and the test dataset.

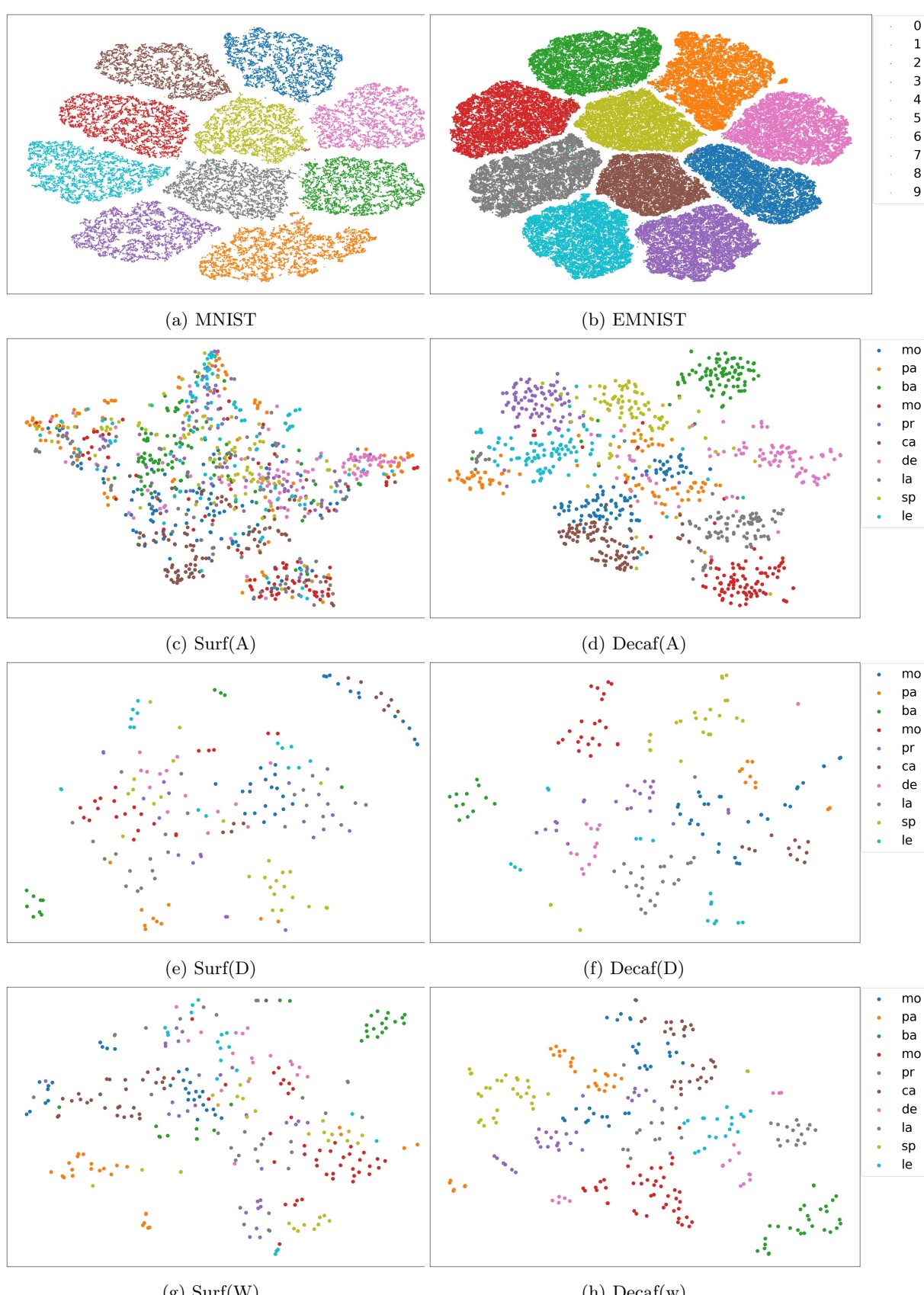

Figure 13: TSNE visulization for datasets MNIST,EMNIST,Caltech Office.

R.4   NUMERICAL DETAILS AND PERFORMANCE

**Accuracy Comparison.** In Table 3 and Table 5, we present the accuracy results for the MPGW, UGW, and the proposed PGW methods when using three different initialization methods: POT, FLB-UOT, and FLB-POT.

Following (Chapel et al., 2020), in the MPGW and PGW methods, we incorporate the prior knowledge $\pi$ into the definition of $p$ and $q$. Thus, it is sufficient to set $mass = \pi$ for MPGW and choose a sufficiently large value for $\lambda$ in the PGW method. This configuration ensures that the mass matched in the target domain $\mathcal{Y}$ is exactly equal to $\pi$. However, in the UGW method (Séjourné et al., 2021), the setting is $p = \frac{1}{n}1_n$ and $q = \frac{1}{m}1_m$.

Overall, all methods show improved performance in MNIST and EMNIST datasets. One possible reason for this could be the better separability of the embeddings in MNIST and EMNIST, as illustrated in Figure 13. Additionally, since MPGW and PGW incorporate information from $r$ into their formulations, they exhibit slightly better accuracy in many experiments.

**Numerical details.** In this experiment, to prevent unexpected convergence to local minima in the Frank-Wolf algorithms, we manually set $\alpha = 1$ during the line search step for both MPGW and PGW methods.

For the convergence criteria, we set the tolerance term for Frank-Wolfe convergence and the main loop in the UGW algorithm to be $1e - 5$. Additionally, the tolerance for Sinkhorn convergence in UGW was set to $1e - 6$. The maximum number of iterations for the POT solver in PGW and MPGW was set to $500n$. In addition, for MPGW, we set mass = 0.2 and for PGW method, based on lemma E.2, we set $\lambda$ to be constant such that $2\lambda \geq (\max(|C^X|)^2 + \max(|C^Y|)^2)$. For UGW, as we directly apply the numerical method in (Séjourné et al., 2023), where the prior knowledge $\pi = 0.2$ is not Incorporated in the setting of $p$ and $q$. Thus, in each experiment, we test different parameters $(\rho, \rho_2, \epsilon)$ and select the ones that result in transported mass close to $\pi$.

Regarding data types, we used 64-bit floating-point numbers for MPGW and PGW and 32-bit floating-point numbers for UGW.

For the MNIST and EMNIST datasets, we set $n = 1000$ and $m = 5000$. In the Surf(A) and Decaf(A) datasets, each class contained an average of 100 samples. To ensure the SCAR assumption, we set $n = 1/2*100 = 50$ and $m = 250$. Similarly, for the Surf(D) and Decaf(D) datasets, we set $n = 15$ and $m = 75$. Finally, for Surf(W) and Decaf(W), we used $n = 20$ and $m = 100$.

**Wall-clock time** In Table 4, we provide a comparison of wall-clock times for the MNIST and EMNIST datasets.

| DATASET | INIT METHOD | INIT ACCURACY | MPGW | UGW | PGW (OURS) |
|---------|-------------|---------------|------|-----|------------|
| M → M | POT | 100% | 100% | 95% | 100% |
| M → M | FLB-U | 75% | 96% | 95% | 96% |
| M → M | FLB-P | 75% | 99% | 95% | 99% |
| M → EM | FLB-U | 78% | 94% | 95% | 94% |
| M → EM | FLB-P | 78% | 94% | 95% | 94% |
| EM → M | FLB-U | 75% | 97% | 96% | 97% |
| EM → M | FLB-P | 75% | 97% | 96% | 97% |
| EM → EM | POT | 100% | 100% | 95% | 100% |
| EM → EM | FLB-U | 78% | 94% | 95% | 94% |
| EM → EM | FLB-P | 78% | 95% | 95% | 95% |

Table 3: Accuracy comparison of the MPGW, UGW, and the proposed PGW method on PU learning. Here, 'M' denotes MNIST, and 'EM' denotes EMNIST.

| Source | Target | Init Method | Init Time | MPGW | UGW | PGW (ours) |
|--------|--------|-------------|-----------|------|-----|------------|
| M(1000) | M(5000) | POT | 0.5 | **7.2** | 152.0 | 7.4 |
| M(1000) | M(5000) | FLB-U | 0.02 | 30.5 | 152.6 | **27.8** |
| M(1000) | M(5000) | FLB-P | 0.5 | 27.8 | 144.9 | **26.9** |
| EM(1000) | EM(5000) | POT | 0.5 | **7.3** | 157.3 | 7.5 |
| EM(1000) | EM(5000) | FLB-U | 0.02 | 30.0 | 181.8 | **29.9** |
| EM(1000) | EM(5000) | FLB-P | 0.5 | **22.2** | 155.1 | 22.3 |
| M(1000) | EM(5000) | FLB-U | 0.02 | **34.0** | 157.9 | 34.4 |
| M(1000) | EM(5000) | FLB-P | 0.5 | **34.9** | 155.5 | 35.0 |
| EM(1000) | M(5000) | FLB-U | 0.02 | 24.3 | 139.3 | **22.2** |
| EM(1000) | M(5000) | FLB-P | 0.5 | 32.0 | 162.7 | **29.9** |
| M(2000) | M(10000) | POT | 1.7 | **31.1** | 1384.8 | 32.1 |
| M(2000) | M(10000) | FLB-U | 0.1 | 209.0 | 1525.8 | **192.5** |
| M(2000) | M(10000) | FLB-P | 1.7 | 208.0 | 1418.4 | **192.1** |
| M(2000) | EM(10000) | FLB-U | 0.1 | 165.1 | 1606.1 | **164.2** |
| M(2000) | EM(10000) | FLB-P | 1.7 | 224.1 | 1420.7 | **223.7** |
| EM(2000) | M(10000) | FLB-U | 0.1 | 149.1 | 1426.5 | **138.1** |
| EM(2000) | M(10000) | FLB-P | 1.7 | 113.9 | 1407.6 | **103.9** |
| EM(2000) | EM(10000) | POT | 1.6 | **32.4** | 1445.9 | 33.4 |
| EM(2000) | EM(10000) | FLB-U | 0.1 | **233.0** | 1586.3 | 233.9 |
| EM(2000) | EM(10000) | FLB-P | 1.8 | **142.1** | 1620.6 | **142.1** |

Table 4: In this table, we present the wall-clock time for the MPGW, UGW, and the proposed PGW method, as well as three different initialization methods (POT, FLB-UOT, FLB-POT). In the "Source" (or "Target") column, M (or EM) denotes the MNIST (or EMNIST) dataset, and the value 1000 (or 5000) denotes the sample size of $X$ (or $Y$). The units of all reported wall-clock times are seconds.

## S  Compute Resources

All experiments presented in this paper are conducted on a computational machine with an AMD EPYC 7713 64-Core Processor, $8 \times 32$GB DIMM DDR4, 3200 MHz, and an NVIDIA RTX A6000 GPU.

| Dataset | Init Method | Init Accuracy | MPGW | UGW | PGW (ours) |
|---|---|---|---|---|---|
| SURF(A) → SURF(A) | POT | 81.2% | 74.7% | 66.5% | 74.7% |
| SURF(A) → SURF(A) | FLB-U | 64.9% | 65.7% | 66.5% | 65.7% |
| SURF(A) → SURF(A) | FLB-P | 63.3% | 66.5% | 66.5% | 66.5% |
| DECAF(A) → DECAF(A) | POT | 95.1% | 95.1% | 60.8% | 95.1% |
| DECAF(A) → DECAF(A) | FLB-U | 78.0% | 67.4% | 83.7% | 67.4% |
| DECAF(A) → DECAF(A) | FLB-P | 78.0% | 74.7% | 88.6% | 74.7% |
| SURF(D) → SURF(D) | POT | 100% | 100% | 89.3% | 100% |
| SURF(D) → SURF(D) | FLB-U | 62.7% | 73.3% | 84.0% | 73.3% |
| SURF(D) → SURF(D) | FLB-P | 60.0% | 60.0% | 78.7% | 60.0% |
| DECAF(D) → DECAF(D) | POT | 100% | 100% | 100% | 100% |
| DECAF(D) → DECAF(D) | FLB-U | 76.0% | 68.0% | 70.7% | 68.0% |
| DECAF(D) → DECAF(D) | FLB-P | 73.3% | 73.3% | 86.7% | 73.3% |
| SURF(W) → SURF(W) | POT | 100.0% | 100.0% | 81.3% | 100.0% |
| SURF(W) → SURF(W) | FLB-U | 76.0% | 70.7% | 81.3% | 70.7% |
| SURF(W) → SURF(W) | FLB-P | 73.3% | 68.0% | 78.7% | 68.0% |
| DECAF(W) → DECAF(W) | POT | 100% | 100% | 100% | 100% |
| DECAF(W) → DECAF(W) | FLB-U | 73.3% | 68.0% | 62.7% | 68.0% |
| DECAF(W) → DECAF(W) | FLB-P | 70.7% | 70.7% | 73.3% | 70.7% |
| SURF(A) → DECAF(A) | FLB-U | 73.9% | 83.7% | 91.8% | 83.7% |
| SURF(A) → DECAF(A) | FLB-P | 73.9% | 83.7% | 87.8% | 83.7% |
| DECAF(A) → SURF(A) | FLB-U | 67.3% | 67.3% | 69.0% | 67.3% |
| DECAF(A) → SURF(A) | FLB-P | 67.3% | 68.2% | 71.4% | 68.2% |
| SURF(D) → DECAF(D) | FLB-U | 76.0% | 76.0% | 65.3% | 76.0% |
| SURF(D) → DECAF(D) | FLB-P | 76.0% | 76.0% | 65.3% | 76.0% |
| DECAF(D) → SURF(D) | FLB-U | 73.3% | 62.7% | 73.3% | 62.7% |
| DECAF(D) → SURF(D) | FLB-P | 73.3% | 73.3% | 73.3% | 73.3% |
| SURF(W) → DECAF(W) | FLB-U | 70.7% | 70.7% | 76.0% | 70.7% |
| SURF(W) → DECAF(W) | FLB-P | 70.7% | 70.7% | 76.0% | 70.7% |
| DECAF(W) → SURF(W) | FLB-U | 68.0% | 68.0% | 65.3% | 68.0% |
| DECAF(W) → SURF(W) | FLB-P | 68.0% | 68.0% | 70.7% | 68.0% |

Table 5: In this table, we present the accuracy comparison of the MPGW, UGW, and the proposed PGW method. We first describe the initialization method and report its accuracy, followed by the accuracy of MPGW, UGW, and PGW. The prior distribution $\pi = p(l = 1)$ is set to be 0.2 in all experiments. To guarantee the SCAR assumption, for Surf(A) and Decaf(A), we set $n = 50$, which is half of the total number of data in one single class. $m$ is set to be 250. Similarly, we set suitable $n, m$ for Surf(D), Decaf(D), Surf(W), Decaf(W).

| Dataset | Init Method | Init Time | MPGW | UGW | PGW (ours) |
|---|---|---|---|---|---|
| SURF(A) → SURF(A) | POT | 1.4E-3 | 1.9E-2 | 3.8 | 2.0E-2 |
| SURF(A) → SURF(A) | FLB-U | 2.2E-3 | 1.8E-2 | 3.6 | 1.9E-2 |
| SURF(A) → SURF(A) | FLB-P | 1.7E-3 | 1.8E-2 | 3.8 | 1.5E-2 |
| DECAF(A) → DECAF(A) | POT | 1.7E-3 | 1.9E-2 | 7.3 | 1.9E-2 |
| DECAF(A) → DECAF(A) | FLB-U | 9.6E-3 | 1.8E-2 | 6.8 | 1.5E-2 |
| DECAF(A) → DECAF(A) | FLB-P | 2.0E-3 | 1.8E-2 | 6.7 | 1.6E-2 |
| SURF(D) → SURF(D) | POT | 2.9E-4 | 5.8E-4 | 3.1 | 3.8E-4 |
| SURF(D) → SURF(D) | FLB-U | 1.4E-3 | 3.0E-3 | 5.4 | 2.2E-3 |
| SURF(D) → SURF(D) | FLB-P | 3.1E-4 | 2.9E-3 | 5.4 | 2.1E-3 |
| DECAF(D) → DECAF(D) | POT | 3.1E-4 | 6.0E-4 | 3.3 | 3.6E-4 |
| DECAF(D) → DECAF(D) | FLB-U | 1.4E-3 | 2.9E-3 | 5.8 | 2.1E-3 |
| DECAF(D) → DECAF(D) | FLB-P | 3.4E-4 | 2.8E-3 | 5.3 | 2.0E-3 |
| SURF(W) → SURF(W) | POT | 3.0E-4 | 6.0E-4 | 5.2 | 3.6E-4 |
| SURF(W) → SURF(W) | FLB-U | 1.3E-3 | 2.9E-3 | 5.1 | 2.1E-3 |
| SURF(W) → SURF(W) | FLB-P | 3.3E-4 | 2.9E-3 | 5.1 | 2.1E-3 |
| DECAF(W) → DECAF(W) | POT | 3.3E-4 | 6.2E-4 | 3.3 | 3.4E-4 |
| DECAF(W) → DECAF(W) | FLB-U | 1.2E-3 | 2.9E-3 | 5.8 | 2.1E-3 |
| DECAF(W) → DECAF(W) | FLB-P | 3.3E-4 | 2.8E-3 | 5.4 | 2.0E-3 |
| SURF(A) → DECAF(A) | FLB-U | 1.1E-1 | 2.8E-2 | 6.7 | 2.6E-2 |
| SURF(A) → DECAF(A) | FLB-P | 1.9E-3 | 2.2E-2 | 0.2 | 2.1E-2 |
| DECAF(A) → SURF(A) | FLB-U | 0.1 | 5E-2 | 6.7 | 4E-2 |
| DECAF(A) → SURF(A) | FLB-P | 2E-3 | 1.8 | 6.8 | 1.5 |
| SURF(D) → DECAF(D) | FLB-U | 1.8E-3 | 5.3E-3 | 6.0 | 2.3E-3 |
| SURF(D) → DECAF(D) | FLB-P | 3.5E-4 | 3.9E-4 | 5.9 | 3.8E-4 |
| DECAF(D) → SURF(D) | FLB-U | 1.8E-3 | 0.296 | 5.6 | 0.165 |
| DECAF(D) → SURF(D) | FLB-P | 3.3E-4 | 0.218 | 5.6 | 0.170 |
| SURF(W) → DECAF(W) | FLB-U | 1.8E-3 | 5.3E-3 | 5.0 | 2.3E-3 |
| SURF(W) → DECAF(W) | FLB-P | 3.4E-4 | 4.1E-4 | 5.0 | 3.9E-4 |
| DECAF(W) → SURF(W) | FLB-U | 1.8E-3 | 5.1E-3 | 5.8 | 2.1E-3 |
| DECAF(W) → SURF(W) | FLB-P | 3.4E-4 | 2.9E-3 | 5.6 | 2.2E-3 |

Table 6: In this table, we present the wall-clock time comparison of the MPGW, UGW, and the proposed PGW method. We report the initialization method and its wall-clock time, followed by the wall-clock time of each of the methods MPGW, UGW, and PGW. The units of all reported wall-clock times are seconds. The prior distribution $\pi = p(l = 1)$ is set to be 0.2 in all experiments. To guarantee the SCAR assumption, for Surf(A) and Decaf(A), we set $n = 50$, which is half of the total number of data in one single class. $m$ is set to be 250. Similarly, we set suitable $n, m$ for Surf(D), Decaf(D), Surf(W), Decaf(W).

