# OpenReview forum: "Partial Gromov-Wasserstein Metric"
_ICLR.cc/2025/Conference — ICLR 2025 Poster_

### Official Review · Reviewer_2CuZ · 2024-10-29

**Soundness:** 3
**Presentation:** 3
**Contribution:** 2
**Rating:** 8
**Confidence:** 3

**Summary:**

This paper introduces Partial Gromov-Wasserstein (PGW), a particular case of UGW, and prove that it is a metric between mm-spaces. The authors propose Frank-Wolfe-based algorithms with convergence guarantees for solving the PGW problem and extend the concept of barycenters to mm-spaces. PGW’s effectiveness is demonstrated through applications like shape matching, retrieval, and interpolation, showing improved performance over existing baselines.

**Strengths:**

1. This paper is well-written and easy to follow.

2. The paper rigorously proves that PGW, a specific case of UGW with the TV norm as the chosen $f$-divergence, defines a valid metric between mm-spaces, contributing to the theoretical understanding of these transport frameworks.

3. The authors propose using the Frank-Wolfe algorithm to solve the PGW problem and provide a convergence guarantee, enhancing the computational tractability of the method.

**Weaknesses:**

1. Limited Experimental Scope: The experimental evaluation in this paper could be expanded. It might be helpful to include experiments on PU learning (positive-unlabeled learning), similar to those conducted in the UGW paper, for a more comprehensive comparison.

2. Hyperparameter Tuning in Baselines: For the toy examples and point cloud matching experiments, the baselines' performance might improve with more thorough hyperparameter tuning since authors use fixed values mentioned in Appendix N. Given that UGW has been shown to be robust to outliers both theoretically and empirically (as demonstrated in the original UGW paper and also [A]), better-tuned baselines could potentially mitigate the impact of outliers more effectively in these experiments.

3. Minor Issues: Several broken references and citations are present in the paper, such as those on lines 165-166, 894, and 1025. Addressing these would improve the paper's readability and professionalism.


[A] Tran, Quang Huy, et al. "Unbalanced co-optimal transport." *Proceedings of the AAAI Conference on Artificial Intelligence*. Vol. 37. No. 8. 2023.

**Questions:**

1. Could the authors incorporate other experiments, such as PU learning (positive-unlabeled learning), to provide a broader evaluation?

2. Could the authors report the results of varying hyperparameters in the baselines to examine their impact on performance?

3. While the KL divergence in UGW is not a rigorous metric, it has desirable properties such as robustness to outliers. Could the authors explain why the TV norm used in PGW results in better performance in the numerical experiments compared to UGW? Alternatively, is the improved performance due to the use of the Frank-Wolfe (FW) algorithm instead of Sinkhorn?

---

> ### Author Response · Authors · 2024-11-23
> **Weaknesses 1,2,3. Questions 1,2,3**
>
> We would like to express our gratitude for the thorough and insightful feedback provided in the review. In what follows, we offer our explanations to the reviewer's questions and concerns. We welcome follow-up inquiries and would be happy to engage in more in-depth discussions.
>
> ---
>
> **W1 and Q1: Experimental Scope.** Since the main focus of this paper is the "Partial Gromov-Wasserstein Metric," experiments prioritizing the transportation cost from GW/PGW/MPGW/UGW are emphasized. The PU learning experiment, however, requires only the transportation plan, which is why this experiment was not included in the initial submission.
>
> We have now added the PU learning results. Please see Section R and Tables 3-7 in the updated PDF.
>
> In our tests, PGW and MPGW show similar accuracy. This is because, in this setting, full matching from positive samples to the test dataset is desired. Based on our Proposition L.1, these two methods can yield the same transportation plan. However, their transportation costs — though different — are not relevant to this particular experiment.
>
> ---
>
> **W2 and Q2: Hyperparameter Tuning.**
>
> - **Clarification of Potential Misunderstanding.** The sentence "$\rho=1.0$" in Appendix N will be updated to the following:
>
>   *"We perform a grid search to set $\rho$ and test $\rho = 0.01, 0.1, 1.0, 10.0$. The value $\rho=1.0$ yielded the best visualization and was selected in the paper."*
>
>   Additionally, we tested different values of $\epsilon$ in UGW and selected the smallest value that avoids NaN errors, which is $\epsilon=0.05$. **Therefore, UGW’s performance in Figure 1 is based on sufficient hyperparameter tuning.**
>
> - **Reproducibility.** Numerical reproduction details can be found in the code repository at [repo](https://anonymous.4open.science/r/PGW_Metric-7CFE/README.md) under `shape_matching/shape_matching.ipynb`.
>   Results with different parameters are presented for [2D](https://anonymous.4open.science/r/PGW_Metric-7CFE/shape_matching/results/shape_match_full_2d.png) and [3D](https://anonymous.4open.science/r/PGW_Metric-7CFE/shape_matching/results/shape_match_full_3d.png).
>
> - **Experiment-Specific Parameter Setting.** In the shape matching experiment, one shape (red shape) does not contain outliers, while the other shape (blue + green shape) does. Full mass matching is preferred across all methods. In MPGW/PGW/UGW, the corresponding parameter ($mass, \lambda, \rho$) should be set sufficiently large.
>
>   <!-- - Theoretically, parameter $\rho$ in UGW should also be set to be sufficiently large. However, we observe that when $\rho=10.0$, matching between clean data and outliers is still observed. We suspect this is due to the effect of the Sinkhorn entropic term. -->
>
> - **UGW's Visualization Performance.** UGW does not yield better visualizations in the shape matching experiments, primarily due to its reliance on the Sinkhorn algorithm, while PGW/MPGW use linear programming in each FW iteration. The negative entropy term in the Sinkhorn algorithm induces a mass-splitting transportation plan, as observed in Figure 1.
>
> - **Thresholding in UGW Paper Visualization.** In Figure 1 of the [UGW paper](https://arxiv.org/pdf/2009.04266), the visualization threshold is likely increased, causing many matches to be omitted, which reduces the appearance of mass splitting in the figure.
>
> ---
>
> **W3: Minor Issues.** The authors apologize for these compile errors, which have now been fixed.
>
> ---
>
> **Q3: PGW vs UGW.** We refer to the explanation in our official comment, in the section titled "PGW vs UGW". The primary reason for the performance difference between UGW and PGW/MPGW lies in their numerical solvers:
>
> - UGW applies the Sinkhorn solver, which introduces a negative entropy regularization term. This term promotes mass-splitting transportation plans, as they yield higher entropy. Consequently, UGW sometimes matches clean data to outliers.
> - PGW/MPGW do not include the Sinkhorn regularization term and instead use linear programming in each FW iteration, avoiding this issue.
>
> > Alternatively, is the improved performance due to the use of the Frank-Wolfe (FW) algorithm instead of Sinkhorn?
>
> The improved performance is primarily due to the use of linear programming in each FW algorithm step, rather than the Sinkhorn algorithm.

---

> > ### Comment · Reviewer_2CuZ · 2024-11-24
> >
> > I appreciate the authors' detailed response to my questions. I have raised my score from 6 to 8.

---

> > > ### Author Response · Authors · 2024-11-25
> > > **comment**
> > >
> > > The authors thank the reviewer for their valuable comments and insightful discussion. A detailed comparison between UGW and PGW/MPGW in the shape matching experiment will be included in the revised paper.

---

### Official Review · Reviewer_htBd · 2024-11-02

**Soundness:** 4
**Presentation:** 3
**Contribution:** 3
**Rating:** 8
**Confidence:** 4

**Summary:**

The Gromov-Wasserstein (GW) distance is a metric on probability measures supported on different metric spaces. This paper extends this notion to more general Radon measures, not necessarily probability measures, termed partial Gromov-Wasserstein (PGW) distance. The authors show that PGW is indeed a metric. Two Frank-Wolfe (FW) algorithms are proposed to compute PGW, and three examples are provided to illustrate the performance of PGW.

**Strengths:**

1, The main contribution is to establish the metric property of partial GW.

2, Simulations in subsection 6.2 clearly demonstrate improved performance of PGW compared to MPGW.

2, The simulation examples are interesting and illustrative.

**Weaknesses:**

1, The current manuscript has many typos. For example, the citations on lines 165 and 177 are not displaying properly.

**Questions:**

1, Please read the paper thoroughly and correct any other typos.

2, Is the proposed Algorithm 1 the same as Algorithm 1 in the paper "Partial Optimal Transport with Applications on Positive-Unlabeled Learning"? What are the novelties in the optimization algorithms?

---

> ### Author Response · Authors · 2024-11-23
> **Novelty of Algorithm 1, typos.**
>
> We appreciate the reviewer’s time and insightful feedback.
>
> ---
>
> **W1 and Q1: Typos.** The authors apologize for the compile errors in these lines. These errors have been fixed.
>
> ---
>
> **Q2: Algorithm 1.** Please refer to the section titled "Novelty of Algorithm 1" in our official comment.

---

### Official Review · Reviewer_pJHH · 2024-11-03

**Soundness:** 3
**Presentation:** 3
**Contribution:** 2
**Rating:** 6
**Confidence:** 4

**Summary:**

This paper introduces the Partial Gromov-Wasserstein (PGW) metric, a novel adaptation of the Gromov-Wasserstein (GW) distance tailored for metric measure spaces. Unlike traditional GW, which requires equal mass in spaces for meaningful comparison, PGW accommodates unbalanced settings through the incorporation of a total variation penalty. The authors present theoretical results, including proof of PGW as a metric, and develop two Frank-Wolfe solvers for computational efficiency. Experiments on shape matching, retrieval, and interpolation demonstrate PGW's robustness to some extent.

**Strengths:**

- The paper rigorously proves that PGW qualifies as a metric.
- The two Frank-Wolfe solvers for PGW demonstrate effective performance, making the method feasible for practical applications.
- Experimental results indicate that PGW outperforms traditional GW in handling unbalanced data with outliers, enhancing its utility in real-world tasks.

**Weaknesses:**

- Comparisons with alternative unbalanced OT metrics are somewhat limited, which leaves the practical performance of PGW against state-of-the-art methods less explored.
- The robustness parameter $\lambda$ requires careful tuning, which could pose a challenge for practical applications.
- The computational efficiency of PGW may still be restrictive in high-dimensional settings or for very large datasets.

**Questions:**

- The baseline selection of the experiment is too limited to demonstrate the effectiveness of your methods. Why don't you compare the methods [1,2,3] based on robust OT? Please add corresponding experiments.
- - [1] Nietert S, Goldfeld Z, Cummings R. Outlier-robust optimal transport: Duality, structure, and statistical analysis[C]//International Conference on Artificial Intelligence and Statistics. PMLR, 2022: 11691-11719.
- - [2] Wang X, Huang J, Yang Q, et al. On Robust Wasserstein Barycenter: The Model and Algorithm[C]//Proceedings of the 2024 SIAM International Conference on Data Mining (SDM). Society for Industrial and Applied Mathematics, 2024: 235-243.
- - [3] Le K, Nguyen H, Nguyen Q M, et al. On robust optimal transport: Computational complexity and barycenter computation[J]. Advances in Neural Information Processing Systems, 2021, 34: 21947-21959.
- In Wasserstein barycenter's experiment for robustness, why only show the interpolation between two shapes? Can you show me how the algorithm computes the average shape of several shapes?
- It seems that all the experiments you provide can be replaced by the equivalent of robust OT, which cannot truly reflect the advantages of robust Gromov-Wasserstein barycenter.
- Is the parameter $\lambda$ easy to search? Different proportions of noise seem to require different $\lambda$s. Does this hinder the practicality of this method? It seems that the specific $\lambda$ search process and the influence of $\lambda$ on the experimental effect are not given in the experiment.

---

> ### Author Response · Authors · 2024-11-23
> **Weaknesses 1,2,3, Questions 1,2,3,4**
>
> We would like to thank the reviewer for their feedback. We hope the following adequately addresses the reviewer's concerns and would be more than happy to engage in further discussions.
>
> ---
>
> **W1 and Q1: Comparisons with OT/UOT/Robust OT.** We have added OT and UOT as baselines in our experiments. For further details, please refer to the official comment section:
>
> - **Shape Interpolation Experiment:** "Free-support OT barycenter" and "free-support partial OT barycenter" have been included as additional baselines.
> - **Shape Retrieval Experiment:** "OT" and "UOT" distances have been added as additional baselines.
>
> > Why don't you compare the methods [1,2,3] based on robust OT?
>
> We appreciate the suggestion to include additional baselines ([1], [2], [3]), but their inclusion is not feasible for the following reasons:
>
> - **Outlier-Robust OT ([1]):**
>    - For interpolation, a barycenter algorithm for "outlier-robust OT" is unavailable in the public [GitHub repository](https://github.com/sbnietert/robust-OT).
>    - For shape retrieval, functions for computing transportation plans and costs are not provided. The implementation of "outlier-robust OT" in this repository is implicitly described via dual variables (two neural networks) and is incorporated into a GAN framework.
>    - Since [1] is a special case of UOT, the inclusion of UOT in our experiments adequately represents [1].
>
> - **OT Barycenter ([2], [3]):**
>    - The method [2] focus on (partial) OT barycenter algorithms and lack public implementations.
>    - The method [3] focus on **fixed-support barycenter problem** and we need **free-support barycenter method**. In addition, [3] lack public implementations.
>    - [2] and [3] propose computationally efficient Sinkhorn variants for OT barycenters. Since we already include OT and partial OT barycenters, these methods are adequately represented.
>
> ---
>
> **W2 and Q4: Parameter Tuning.** We refer to the "Parameter Setting" section in the official comment for details. In particular, we note that in the interpolation and shape retrieval experiments, we do not require signficant tuning of the parameter $\lambda$.
>
>
>
>
> **W3: PGW Efficiency.** As detailed in Section K, the computational complexity of the proposed FW algorithm matches that of classical FW algorithms for GW and mass-constrained PGW. We acknowledge that handling large datasets is a challenge, but this limitation is not unique to PGW—it also applies to GW and UGW.
>
> However, the computational complexity of PGW, like classical OT, UOT, outlier-robust OT, GW, and UGW, is independent of the data dimension.
>
> ---
>
> **Q2: Multi-Shape Interpolation.** We have added an experiment with interpolation among four shapes. The baselines include OT barycenter, partial OT barycenter, GW barycenter, and PGW barycenter. Please refer to the official comment section for experimental details and results.
>
> ---
>
> **Q3: Experimental Setups.** Multi-shape interpolation and shape retrieval experiments have been conducted with additional baselines, including OT and UOT. For details, refer to the official comment section. The superiority of GW/PGW over OT/UOT is explained in detail in the results analysis.

---

> > ### Comment · Reviewer_pJHH · 2024-11-25
> >
> > Regarding some algorithms in Optimal Transport (OT), even if there is no public implementation, implementing them oneself should not be particularly challenging. If someone is already familiar with OT, implementing Partial Gromov-Wasserstein (PGW) or an OT-based version should be quite straightforward. This is simply a matter of implementing a basic algorithm, and it doesn't involve a significant amount of code. Claiming that the lack of a public implementation is the reason seems unconvincing. Based on my experience and intuition, I strongly suspect that the experimental results might not outperform the OT version, which is why adding such experiments has been consistently opposed.

---

> ### Author Response · Authors · 2024-11-25
> **Follow-up to Reviewer pJHH**
>
> We would like to clarify a potential misunderstanding. **We do not oppose the suggested addition of baseline methods.** In fact, we have already conducted experiments with baselines equivalent to [[1]](https://proceedings.mlr.press/v151/nietert22a/nietert22a.pdf), [[2]](https://arxiv.org/pdf/2312.15762), and [[3]](https://arxiv.org/pdf/2102.06857), and our proposed PGW consistently outperforms these additional baselines. For further details, please refer to the official comment, which highlights cases where OT/UOT-based methods fail. These results are also available in the updated manuscript.
>
> ### Summary
> - **[[1]](https://proceedings.mlr.press/v151/nietert22a/nietert22a.pdf):** This (see Eq. 1) is a special case of the [Unbalanced OT problem](https://arxiv.org/abs/1607.05816) (see Eq. 2.4). We have already incorporated the [UOT distance](https://arxiv.org/abs/1607.05816) into the shape retrieval experiment. Additionally, it is important to note that "outlier-robust OT" is **not a metric**. Specifically, when one shape is a subset of another, outlier-robust OT produces a zero cost, even though the shapes are distinct. However, in the shape experiment, the metric property is critical as it provides a meaningful measure to distinguish shapes across different classes.
>
> - **[[2]](https://arxiv.org/pdf/2312.15762):** This corresponds to the "free-support partial OT barycenter" (see Eq. 3.10) and is equivalent to the Free-support OT barycenter when there are no outliers in the data. We have included the [Free-support OT barycenter](https://proceedings.mlr.press/v32/cuturi14.html) (see Section 4.4) in the interpolation experiment. Additionally, we added the [partial OT barycenter](https://hal.science/hal-02111220/file/spot.pdf) (see Section 6.3).
>   The main distinction between [[2]](https://arxiv.org/pdf/2312.15762) and the "Free-support (partial) OT barycenter" is that [[2]](https://arxiv.org/pdf/2312.15762) achieves similar results while offering improved computational efficiency.
>
> - **[[3]](https://arxiv.org/pdf/2102.06857):** This focuses on the **fixed-support** Unbalanced OT barycenter (see Eq. 10). However, for the shape interpolation experiment, we require **free-support** barycenter methods. Therefore, **the baseline in [[3]](https://arxiv.org/pdf/2102.06857) is not applicable** to our experiment. Instead, we have included the [partial OT barycenter](https://hal.science/hal-02111220/file/spot.pdf) (see Section 6.3).
>
> ---
>
> > "I strongly suspect that the experimental results might not outperform the OT version, which is why adding such experiments has been consistently opposed."
>
> We respectfully disagree with this statement. The experimental results clearly demonstrate that PGW outperforms OT in our scenarios. **Once again, we emphasize that we do not oppose the inclusion of these methods.** As stated in our initial rebuttal, we have already incorporated the OT versions into our experiments. The reasons for their limitations are discussed in the official comments.
>
> For a concise comparison of GW and OT, please refer to the section on GW vs OT. In brief, GW/PGW/UGW enables matching between shapes in different spaces, while OT/UOT requires both shapes to reside in the same space.
>
> ---
>
> Finally, we emphasize the following experimental results to explain why the "outlier-robust OT" [1] and the "robust OT barycenter" [2]  do not achieve better performance. In summary, in this setting, there are no "outliers," and these methods are equivalent to the standard OT/OT barycenter.
>
> - **Shape Retrieval Experiment (Dataset I):**
>   - In this dataset, Unbalanced OT (or the outlier-robust OT in [[1]](https://proceedings.mlr.press/v151/nietert22a/nietert22a.pdf)) offers no advantage over OT because there are no outliers. This is reflected in the results: for both cases $d=2$ and $d=4$, OT performs better than UOT.
>   - However, when $d=4$, the performance of both OT/UOT significantly deteriorates compared to GW/PGW. The theoretical explanation for this behavior is provided in the official comments.
>
> - **Multi-Shape Interpolation Experiment (First Dataset, Noise Level $\eta=0$):**
>   - In this case, there are **no outliers** among the four shapes. As a result, OT and UOT **free-support** barycenter methods yield similar results because UOT reduces to OT in this special setting. The results align with this expectation.
>   - While OT/UOT methods perform well for interpolation between 2D shapes, their performance deteriorates for interpolation between 2D and 4D shapes, as these methods are not designed for such cases.

---

> > ### Comment · Reviewer_pJHH · 2024-11-25
> >
> > Thank you for your detailed answer. I will slightly raised my score.

---

> > > ### Author Response · Authors · 2024-11-25
> > >
> > > The authors thank the reviewer for the valuable comments and discussion.

---

### Official Review · Reviewer_t4aR · 2024-11-04

**Soundness:** 2
**Presentation:** 2
**Contribution:** 2
**Rating:** 6
**Confidence:** 4

**Summary:**

The Gromov-Wasserstein (GW) distance has drawn a lot of attention in recent years for the comparison of measures in different metric spaces. This paper focuses on the Unbalanced Gromov-Wasserstein (UGW) problem, which they called Partial Gromov-Wasserstein (PGW). They showed PGW is a well-defined metric between metric measuring spaces and discussed some theoretical properties. They designed a Frank-Wolfe algorithm-based method to solve PGW. Convergence analysis is provided. Numerically, they tested it on some datasets with existing baselines.

**Strengths:**

Pros:

They propose PGW, a total variation version of the UGW problem with theoretical properties analysis.

The experiment is showing their good performance in PGW.

**Weaknesses:**

Cons:

1. The main concern is the novelty and significance since the problem itself is not well-generalizable to other problems. Meanwhile, some techniques are adopted from existing works. For example:

a. What is the unique technique of your method design compared to the Frank-Wolfe solver presented in (Chapel et al., 2020) except their problem setting is UGW?

b. How is the convergence analysis partially harder than the existing method?

c. Since the algorithm only guarantees stationary points (this is fine, the reviewer does not require a global optimality), how can we convinced it works well in more real scenarios? Is there any more evidence?



2. For the FW algorithm in the optimization problem in line 331, it’s the main subroutine in each iteration, how is the complexity of solving it? This is important in the analysis since FW usually could be dramatically slowed down by the subroutine.


3. Minor issue. There are some citation errors in lines 165 and 177.

**Questions:**

Please address the question in the weakness section.

---

> ### Author Response · Authors · 2024-11-23
> **Weaknesses 1,2,3**
>
> We are grateful for the thorough and insightful feedback. In what follows, we offer our explanations to the reviewer's questions and concerns. We welcome follow-up inquiries and would be happy to engage in more in-depth discussions.
>
> ---
>
> **W1: Novelty/Significance**
>
> > What is the unique technique of your method design compared to the Frank-Wolfe solver presented in (Chapel et al., 2020) except their problem setting is UGW?
>
> We refer to the official comment, "Novelty of Algorithm 1" section, for a detailed explanation.
>
> In summary, the main difference between the proposed Algorithm 1 and the FW algorithm in [Chapel et al., 2020](https://arxiv.org/abs/2002.08276) stems from the fact that these algorithms are designed for different problems.
>
> We've also explained the main difference between PGW and MPGW in Section L and refer the reviewer to the official comment for an intuitive example.
>
> > How is the convergence analysis partially harder than the existing method?
>
> The convergence analysis of the proposed algorithms and the theoretical convergence rate is presented in Appendix K. Based on Proposition K.3, the convergence rate of the proposed algorithm is the same as the convergence rate of the Frank-Wolfe (FW) algorithm in MPGW and classical GW. We would like to request further clarification from the reviewer on what they mean by "partially harder than the existing method."
>
> > Since the algorithm only guarantees stationary points (this is fine, the reviewer does not require a global optimality), how can we be convinced it works well in more real scenarios? Is there any more evidence?
>
> Since GW/UGW/PGW/MPGW problems are non-convex, all existing solvers can only guarantee convergence to stationary points rather than the global minimum. However, in practice, several initialization methods can be applied to improve their performance in real-world scenarios. We refer the reviewer to the official comment, section "Initialization Methods," for details.
>
> > Is there any more evidence?
>
> The Frank-Wolfe PGW solver has been applied in our experiments on shape retrieval and point cloud interpolation. Additionally, the Frank-Wolfe GW algorithm is widely used in related applications, such as PU learning ([Chapel et al., 2020](https://arxiv.org/abs/2002.08276)), shape interpolation/clustering ([Peyré et al., 2016](https://proceedings.mlr.press/v48/peyre16.pdf)), and graph matching ([Xu et al., 2019](https://arxiv.org/abs/1901.06003)).
>
> ---
>
> **W2: FW Complexity.** The complexity of each FW iteration is primarily determined by the partial OT solving step (line 331), which has a complexity of $\mathcal{O}(n^3)$ when using a linear programming solver, where $n$ is the average size of $\mu$ and $\nu$. If the Sinkhorn algorithm is used instead, the complexity for this step reduces to $\mathcal{O}\left(\frac{1}{\epsilon^2}n^2\ln(n)\right)$, where $\epsilon$ represents the weight of the entropic regularization term. The performance (in terms of accuracy and computational efficiency) depends heavily on the choice of $\epsilon$.
>
> All other steps in each FW iteration (lines 330, 332, and 333) require $\mathcal{O}(n^2)$ complexity, which is negligible compared to the computation in line 331.
>
> ---
>
> **W3: Citation Issues.** We thank the reviewers for pointing out the citation issues. These LaTeX compile errors have now been fixed.

---

> > ### Comment · Reviewer_t4aR · 2024-11-24
> > **Response to the author**
> >
> > The reviewer thanks the author for the explanation. I slightly raised my score.

---

> > > ### Author Response · Authors · 2024-11-25
> > >
> > > The authors sincerely thank the reviewer for their valuable comments and thoughtful discussion. The citation issue highlighted has been addressed and corrected.

---

### Official Review · Reviewer_q52X · 2024-11-04

**Soundness:** 3
**Presentation:** 2
**Contribution:** 3
**Rating:** 6
**Confidence:** 4

**Summary:**

This paper improves unbalanced approaches to the optimal transport (OT) problem.
The objective is to compare probability measures between different metric spaces (GW distances).
To overcome the non-convexity of the  formulation, which usually relies on the Frank-Wolfe (FW)
algorithm.

The authors propose a new metric rooted in the so called Partial Gromov-Wasserstein (PGW) problem,
and the main novelty is to use a total variation penalty as well as to provide efficient solvers.

The paper commences by introducing OT and its partial extension POT.  Then, it comes out the regularization
term which in principle is the Kullback-Leibler divergence which is replaced by Total Variation (TV).
Then, the FW algorithm is adopted.

The toy experiment:
"TOY EXAMPLE: SHAPE MATCHING WITH OUTLIERS
We use the moon dataset and synthetic 2D/3D spherical data in this experiment."

should be explained earlier in the paper to motivate the reader. Formal language should not be
an enemy of intuition.

Pros:
- Nice formal language.

Cons:
- Not intuitive for general readers.
- The contribution is relevant (avoid strong outliers)

Questions:
- Is there any formalism for Bregman divergences?

**Strengths:**

Nice formalization.

**Weaknesses:**

Very hard to read even for an experienced reviewer.

**Questions:**

* Adequacy to Bregman divergences?
* Put synthetic examples in the beigning as a motivation and smooth the formal language.

---

> ### Author Response · Authors · 2024-11-23
> **Weakness 1, Questions 1,2**
>
> We truly thank the reviewer for their time and insightful feedback. Below we provide our responses. We hope they adequately address the reviewer's concerns and would be happy to engage in further discussions.
>
> ---
>
> **W1 and Q2: Organization/Writing.**  The following figure will be added into the main text as introduction of OT/GW and motivation for the formulation of PGW: [image](https://anonymous.4open.science/r/PGW_Metric-394D/toy_example/example.png).
>
> Essentially, OT/GW and their variant problems study the following problem: Given two point clouds/shapes, the goal to find a "reasonable matching" between them. This matching can be used in, e.g., color transfer, shape registration, domain adaptation, etc.
>
> For example, in this figure, shape 1 is the union of the circle and the vertical line (the red + orange points), whereas shape 2 is the other straight line (blue points). Classical OT and its unbalanced version can produce this matching based on the absolute locations of the individual points. Shape 1 contains more points than shape 2, so if we aim to match part of shape 1 to shape 2, unbalanced OT can provide such a matching.
>
> However, suppose the aboslute location of the shapes is not important, and we need the matching to reflect the shape's inherent structure. In this example, we would like the straight line part of shape 1 to match the straight line in shape 2. GW/PGW can provide such a matching. However, PGW allows partial matching while GW can only provide full matching. Thus, we can observe that PGW is the only one which provides the ideal matching (straight line in shape 1 matches the straight line in shape 2).
>
> ---
>
> **Q1: Bregman Divergences.** To the best of our knowledge, Bregman divergences and the OT/GW distances are distinct approaches for evaluating the similarity between two datasets or distributions. Bregman divergence applies a strictly convex function to measure the density or probability mass function (PMF) of the two distributions, whereas OT and GW quantify similarity by optimizing a transportation cost function.
>
> However, there is some relationship between OT/GW and Bregman divergence. For example, in unbalanced OT, partial OT, unbalanced GW, and partial GW, the penalty term for mass creation and destruction is modeled using Bregman divergence. Additionally, when the Sinkhorn algorithm is applied in OT/UOT/GW/UGW, the negative entropy regularization term is, in essence, a Bregman divergence term.

---

> > ### Comment · Reviewer_q52X · 2024-11-26
> > **Comment**
> >
> > Thank you for your explanations. I keep my score.

---

### Author Response · Authors · 2024-11-23
**answers to common questions**

## Novelty of Algorithm 1
The proposed Algorithm 1, the algorithm in [Mass Constrained Partial GW](https://arxiv.org/abs/2002.08276), and the algorithm in [Balanced GW](https://proceedings.mlr.press/v48/peyre16.pdf) all belong to the Frank-Wolfe (conditional gradient) algorithm family. From this perspective, the main structure and fundamental idea of these methods are the same, centered around gradient descent.

The differences arise from the specific problems these algorithms aim to address. In particular, variations occur in the gradient computation step (line 330), the optimal direction search step (line 331), and the line search (optimal step size search) step (line 332).

Our algorithm is specifically designed to solve the proposed PGW, which differs from Frank-Wolfe algorithms designed for other GW variants.

## Motivation of Introducing PGW

The motivation for studying PGW lies in our goal to develop a metric that leverages the advantages of GW while accommodating unbalanced settings. MPGW, however, cannot serve as a proper metric.

For instance, consider $K$ different mm-spaces (e.g., shapes or datasets). To use $MPGW$ to measure their similarity, the transported mass $\rho$ must be set to less than the total mass for each shape. This requirement can undermine performance, especially if one shape has an extremely small mass. In contrast, the proposed PGW formulation enables the transport of varying masses for different pairs while maintaining a fixed parameter $\lambda > 0$.

Numerically, the shape retrieval experiment demonstrates that $MPGW$ is unsuitable as a metric, as evidenced by its low accuracy (~20%).

## Main Difference Between GW Methods and OT Methods

OT and unbalanced OT focus on matching two measures (datasets/point clouds) within the same space, whereas GW and its variants (unbalanced/partial GW) address the matching of measures across different spaces.

Even when two datasets reside in the same space, OT and its unbalanced variants (UOT, outlier-robust OT, partial OT) depend on the absolute coordinates of points within each shape. If the goal is to emphasize the intrinsic structure of the shapes while disregarding absolute positions, OT and unbalanced OT are unsuitable. In contrast, GW and its variants are specifically designed for structure-aware matching.

To illustrate this distinction, the following [image](https://anonymous.4open.science/r/PGW_Metric-394D/toy_example/example.png) will be included in the paper. The source shape (red + orange points) consists of a circle (the orange outliers) and a straight line (red), while the target shape (blue points) forms a straight line. If we aim to match the straight line in the source shape with the straight line in the target shape, OT and its variants fail due to their reliance on absolute positions. On the other hand, GW and PGW successfully achieve the desired matching.

---

> ### Author Response · Authors · 2024-11-23
> **Answers to common questions**
>
> ## Addition of OT/UOT Baselines and More Experiments
>
> We have added OT/UOT baselines in the multi-shape interpolation and shape retrieval experiments. These results will be included in the paper.
>
> ### Multi-Shape Interpolation
>
> - **Setup:**
>   Two shapes (bird, human) are in 2D space, while two others (cat, rooster) are in 4D space. The "effective dimension" of each shape is 2, but this information is unknown in the experiment.
>
>   We run one set of experiments without the addition of noise, and then repeat the experiments with the addition of $\eta = 5\%$ noise to the human shape (in 2D) and to the cat shape (in 4D).
> - **Baseline Methods:**
>   OT barycenter, unbalanced OT barycenter, GW barycenter, and PGW barycenter are used as baselines.
>
>   Free-support barycenter algorithms are required. OT barycenter is implemented using [PythonOT](https://pythonot.github.io/). For unbalanced OT, no publicly available code exists for free-support UOT barycenters, so we adapted the partial OT barycenter from [Bonneel et al., 2019](https://perso.liris.cnrs.fr/nbonneel/spot/).
>
>   OT/UOT cannot directly handle interpolation between 2D and 4D shapes. To address this, we embed the 2D shapes into 4D space before applying OT interpolation, followed by PCA for visualization.
>
>   Note that the most rigorous way to apply OT/UOT for interpolation between 2D and 4D data is to find the optimal embedding from 2D to 4D by solving the following optimization problem:
>   $$\min_{\phi:\mathbb{R}^2\to\mathbb{R}^4} OT(\phi (\mu),\nu).$$
>   This formulation aligns with the [Wasserstein GAN](https://arxiv.org/abs/1701.07875) approach. However, this introduces a two-layer optimization problem, and the computational cost is significantly higher than that of GW/PGW.
>
> - **Results:**
>   Comparisons of OT barycenter, UOT barycenter, GW barycenter, and PGW barycenter are available in the [repo](https://anonymous.4open.science/r/PGW_Metric-394D/) in the `interpolation/result` directory.
>
>   We refer to section M.2 in the updated pdf for additional detail.
>
> - **Analysis:**
> When the data is not corrupted by noise, for 2D shapes (e.g., bird and human), OT/UOT interpolation performs comparably to GW/PGW. However, for interpolation between 2D and 4D shapes, OT/UOT barycenters fail, producing results resembling random noise. In contrast, GW and PGW successfully capture the structure even in higher dimensions. When outlier is added into some shapes (i.e. cat and human), we observe PGW admits a smooth interporlation result and GW's interporlation is affected by the noise data.
>
>
> ### Shape Retrieval Experiment
> We refer section O.1. in the update pdf for details.
> - **Dataset Setup:**
>   Data is tested both in the original 2D space and after embedding into a 4D space ($d=4$). In addition, the shapes in 4D space are randomly rotated.
>
> - **Baselines:**
>   OT, unbalanced OT, GW, MPGW, UGW, PGW (ours).
>
> - **Results (Accuracy):**
>
> | Method             | Dataset I  | Dataset II  |
> |--------------------|---------------------|---------------------|
> | OT ($d=2$)       | 95.6%              | 55.8%              |
> | OT ($d=4$)       | 79.3%              | 50.8%              |
> | UOT ($d=2$)      | 73.5%              | 73.3%              |
> | UOT ($d=4$)      | 56.9%              | 63.3%              |
> | GW ($d=2,4$)     | 98.1%              | 80.8%              |
> | MPGW ($d=2,4$)   | 23.7%              | 25.0%              |
> | UGW ($d=2,4$)    | 89.4%              | 90.0%              |
> | PGW (ours, $d=2,4$) | 96.2%              | 100%               |
>
> - **Analysis:**
>     In Dataset 1, when data is in the 2D space, OT achieves an accuracy of 95.6%, while UOT achieves 73.5%. OT's accuracy is slightly lower than GW/PGW, but it remains a strong classifier in this setting. In Dataset 2, UOT outperforms OT with an accuracy of 73.3% compared to 55.8%.
>
>     When the shapes are embedded into 4D space, the accuracy of OT and UOT drops significantly, ranging from 56.9% to 79.3%, far below the performance of GW and PGW. This decline highlights the reliance of OT/UOT on absolute coordinates, which becomes more problematic in higher dimensions. In contrast, GW-based methods (GW, MPGW, UGW, PGW) remain unaffected, as they are invariant to absolute locations.
> Overall, regardless of whether the data is in 2D or 4D space, OT and UOT consistently perform worse than GW and its variants. The gap in performance becomes even more pronounced in 4D space.

---

> ### Author Response · Authors · 2024-11-23
> **Answers to common questions**
>
> ## Initial Guess for GW/UGW/PGW Algorithm
> Since GW, UGW, and PGW are non-convex problems, their Frank-Wolfe or Sinkhorn algorithms can only guarantee convergence to a local minimum rather than a global minimum. Therefore, it is crucial to set a suitable initial guess for these algorithms. The following section has been added to the paper (please refer to Section P in the updated PDF for details):
>
> Suppose we have two mm-spaces $(X, d_X, \mu)$ and $(Y, d_Y, \nu)$:
>
> - **Situation 1:** If the two measures reside in spaces of the same dimension, we can use the solution from unbalanced OT as the initial guess:
>   $$\gamma^0 = UOT(\mu, \nu).$$
>   This approach was proposed by [Chapel et al., 2020](https://arxiv.org/abs/2002.08276).
>   <!-- Note: The above $UOT$ problem can also be replaced with OT or other unbalanced OT formulations. -->
>
> - **Situation 2:** If the two measures reside in spaces of different dimensions, we solve the following unbalanced OT problem:
>   $$\gamma^{(1)} \gets \arg\min_{\gamma \in \mathcal{M}(X \times Y)} \int_{X \times Y} |s_{X,p}(x) - s_{Y,p}(y)|^2 d\gamma(x, y) + \lambda \big(D_\phi(\gamma_1 \parallel \mu) + D_\phi(\gamma_2 \parallel \nu)\big).$$
>   Here, $s_{X,p}(x) = \int_X d_X(x, y) d\mu(y)$, and $s_{Y,p}(y)$ is defined similarly.
>
> This formulation was first introduced by [Mémoli et al., 2011](https://link.springer.com/article/10.1007/s10208-011-9093-5) and is referred to as the First Lower Bound (FLB).
>
>
> ## PGW vs UGW
> In the shape matching experiment, we observe that both MPGW and PGW produce a one-to-one correspondence between the 2D shapes and the 3D shapes. However, the UGW method exhibits significant mass-splitting. To our understanding, this is **not due to the total variation penalty being inherently better than the KL divergence**. Instead, the primary reason lies in their numerical implementation.
>
> The UOT solver used in UGW relies on the Sinkhorn algorithm, whereas the OT/POT solver in MPGW/PGW employs linear programming. Due to the negative entropy regularization in the Sinkhorn algorithm, it tends to produce a mass-splitting transportation plan. For this reason, we believe UGW's performance in shape retrieval is also inferior to that of PGW.
>
>
> ## Parameter Settings
> The following section on parameter settings will be added to the paper.
>
> There are two scenarios to consider:
>
> 1. **Both source and target measures contain outliers:**
>    - The parameter $\lambda$ acts as an upper bound for the transported distance. Specifically, if
>      $$
>      \|d_X(x,x') - d_Y(y,y')\|^2 \geq 2\lambda,
>      $$
>      then either $(x, y)$ or $(x', y')$ will not be transported.
>    - When the distance between outliers and clean data is large, and the pairwise distances within the clean data are relatively small, $\lambda$ should be set to lie between these two scales.
>
> 2. **Only one measure contains outliers:**
>    - As stated in Lemma E.2, $\lambda$ can be set sufficiently large to satisfy:
>      $$
>      2\lambda \geq \max_{x,x' \in X, y,y' \in Y} \|d_X(x,x') - d_Y(y,y')\|^2.
>      $$
>
> In the interpolation and shape retrieval experiments, we fall under the second scenario, which does not require signficant parameter tuning.

---

### Meta-Review · Area_Chair_jv1N · 2024-12-17

**Metareview:**

The paper is a nice contribution, introducing the Partial Gromov-Wasserstein (PGW) metric, which addresses limitations in existing Gromov-Wasserstein formulations for unbalanced settings. The authors provide rigorous theoretical guarantees, such as proving that PGW is a well-defined metric, and propose two effective solvers with convergence guarantees. Some reviewers raised concerns about the experimental assumptions, particularly regarding the equivalence of UOT-based approaches and PGW in practice, and the effect of entropy regularization. The authors adequately addressed these concerns by clarifying the role of regularization and demonstrating PGW’s superior robustness through additional experiments. Overall, the paper’s theoretical depth, algorithmic design, and experimental validation make it a valuable addition to the field, justifying acceptance.

**Additional Comments On Reviewer Discussion:**

During the reviewer discussion, key points included the practical equivalence between PGW and UOT, the impact of entropy regularization on results, and the need for additional comparisons with robust OT methods. pJHH questioned the experimental assumptions, particularly regarding UOT’s performance, while others raised concerns about computational efficiency and hyperparameter tuning. The authors addressed these by clarifying the limitations of entropy regularization, adding further comparisons, and improving experimental details. The rebuttal was thorough and resolved most concerns, demonstrating PGW’s robustness and theoretical contributions, which were decisive in recommending acceptance.

---

### Decision · Program_Chairs · 2025-01-22

Accept (Poster)